# Improving 3-day deterministic air pollution forecasts using machine learning algorithms

Zhiguo Zhang[1], Christer Johansson[2,3], Magnuz Engardt[3], Massimo Stafoggia[4], Xiaoliang Ma[1]

[1] KTH Royal Institute of Technology, Dept. of Civil and Architectural Engineering, Stockholm, Sweden
[2] Department of Environmental Science, Stockholm University, Stockholm, Sweden
[3] Environment and health administration, SLB-analys, Stockholm, Sweden
[4] Department of Epidemiology, Lazio Region Health Service, Rome, Italy

*Correspondence to*: Christer Johansson (christer.johansson@aces.su.se) and Xiaoliang Ma (liang@kth.se)

**Abstract.** As air pollution is regarded as the single largest environmental health risk in Europe it is important that communication to the public is up-to-date and accurate and provides means to avoid exposure to high air pollution levels. Long- as well as short-term exposure to outdoor air pollution is associated with increased risks of mortality and morbidity.
Up-to-date information on present and coming days' air quality helps people avoid exposure during episodes with high levels of air pollution. Air quality forecasts can be based on deterministic dispersion modelling, but to be accurate this requires detailed information on future emissions, meteorological conditions and process-oriented dispersion modelling. In this paper, we apply different machine learning (ML) algorithms – Random forest (RF), Extreme Gradient Boosting (XGB) and Long-Short Term Memory (LSTM) – to improve 1-, 2- and 3-day deterministic forecasts of $PM_{10}$, $NO_x$, and $O_3$ at different sites in
Greater Stockholm, Sweden.

It is shown that the deterministic forecasts can be significantly improved using the ML models but that the degree of improvement of the deterministic forecasts depends more on pollutant and site than on what ML algorithm is applied. Also, four feature importance methods, namely Mean Decrease in Impurity (MDI), Permutation, Gradient-based method, and Shapley Additive exPlanations (SHAP), are utilized to identify significant features that are common and robust across all
models and methods for a pollutant. Deterministic forecasts of $PM_{10}$ are improved by the ML models through the input of lagged measurements and Julian Day partly reflecting seasonal variations not properly parameterised in the deterministic forecasts. A systematic discrepancy by the deterministic forecasts in the diurnal cycle of $NO_x$ is removed by the ML models considering lagged measurements and calendar data like hour and weekday reflecting the influence of local traffic emissions. For $O_3$ at the urban background site, the local photochemistry is not properly accounted for by the relatively coarse Copernicus
Atmosphere Monitoring Service ensemble model (CAMS) used here for forecasting $O_3$ but is compensated for using the ML models by taking lagged measurements into account.

Through multiple repetitions of the training process, the resulting ML models achieved improvements for all sites and pollutants. For $NO_X$ at street canyon sites, MSE decreased by up to 60%, and seven metrics, such as $R^2$ and MAPE, exhibited consistent results. The prediction of $PM_{10}$ is improved significantly at the urban background site, whereas the ML models at street sites have difficulty capturing more information. The prediction accuracy of $O_3$ also modestly increased, with differences between metrics.

Further work is needed to reduce deviations between model results and measurements for short periods with relatively high concentrations(peaks) at the street canyon sites. Such peaks can be due to a combination of non-typical emissions and unfavourable meteorological conditions, which are rather difficult to forecast. Furthermore, we show that general models trained using data from selected street sites can improve the deterministic forecasts of NOx at the station not involved in model training. For $PM_{10}$ this was only possible using more complex LSTM models. An important aspect to consider when choosing ML algorithms is the computational requirements for training the models in the deployment of the system. Tree-based models (RF and XGB) require less computational resources and yield comparable performance in comparison to LSTM. Therefore, tree-based models are now implemented operationally in the forecasts of air pollution and health risks in Stockholm. Nevertheless, there is big potential to develop generic models using advanced ML to take into account not only local temporal variation but also spatial variation at different stations.

All datasets and code in this paper are publicly accessible on Zenodo (https://zenodo.org/records/8433033 ).

**Keywords: Dispersion modelling, Machine Learning, LSTM, $PM_{10}$, $O_3$, $NO_x$, GAM**

# 1    Introduction

According to the World Health Organisation (WHO) air pollution is one of the leading causes of mortality worldwide and is regarded as the single largest environmental health risk (Fuller et al., 2022). Acute effects of air pollution are due to short-term (e.g. daily) exposures that can lead to reduced lung function, respiratory infections and aggravated asthma (Lee et al., 2021). According to the European air quality directive, information on air quality should be made available to the public. Public information regarding the expected health risks associated with current or the next few days' concentrations of pollutants can be very important for sensitive persons when planning their outdoor activities.

There are different approaches to obtaining information on the spatio-temporal variation of air pollutant concentrations – from complex process-oriented models to different types of statistical models. Gaussian plume models are widely used in urban areas for estimating impacts on atmospheric concentrations from different emission sources and for health risk assessments (Munir et al., 2020; Johansson et al., 2009; Orru et al., 2015; Johansson et al., 2017). Eulerian chemical transport models that describe emission, transport, mixing, and chemical transformation of trace gases and aerosols such as e.g. CHIMERE, EMEP and MATCH are part of the Copernicus Atmosphere Monitoring Service (CAMS, atmosphere.copernicus.eu/) to predict air pollution over Europe (Horàlek et al., 2019). The uncertainties in the output of the deterministic models include uncertainties in the input, such as emissions, model algorithms and parameterisations.

In urban areas, detailed knowledge of the dedicated emission source is often crucial. For example, road traffic, as a main emission source, can be modelled by various levels of emission models (André and Rapone, 2009; Ma et al., 2012; Keller et al., 2017). To assess the concentration of contaminants,  it is often required to combine the models of emission and dispersion process e.g. (Ma et al., 2014).  An alternative approach may derive spatio-temporal distribution of air pollutants without modelling emission process. For example, land use regression model is a popular method to explain spatial contrasts of air pollution concentrations e.g. (Hoek et al., 2008).

Data-driven models using machine learning (ML) have become increasingly popular in predicting outdoor air quality (Rybarczyk and Zalakeviciute, 2018; Iskandaryan et al., 2020). Previous studies predict both hourly and daily average concentrations of particulate matter (PM) as well as gaseous air pollutants using meteorological and traffic data (e.g. Quadeer et al., 2020; Di et al., 2019; Thongthammachart et al., 2021; Kamińska, 2019; Chuluunsaikhan et al., 2021; Doreswamy et al., 2020; Castelli et al., 2020; Stafoggia et al., 2019; Stafoggia et al., 2020). In addition, a combination of ML, LUR, dispersion modelling, ground-based and satellite measurements have been used to obtain temporally and spatially distributed concentrations (Shtein et al., 2020; Staffogia et al., 2019; Brokamp et al., 2017; Di et al., 2019). Recently, Kleinert et al. (2022) conducted a study to forecast $O_3$ concentrations in a longer-term horizon, and meanwhile, deterministic model was also combined with ML in the study of Hong et al. (2022) to forecast the $PM_{2.5}$ concentration.

This paper aims to demonstrate how ML can improve the one-, two- and three-day deterministic forecasts of several critical urban air pollutants: particulate matter ($PM_{10}$, particles with an aerodynamic diameter less than 10 µm), nitrogen oxides ($NO_x$) and ozone ($O_3$). The study covers both urban background and street canyon sites in Stockholm, Sweden. Three ML algorithms

were adopted, two based on decision trees (RF and XGB) and one deep neural network model (LSTM). These models were compared to investigate if there are systematic differences in their prediction performance depending on different pollutants and measurement sites, which can be used to improve current applications in Stockholm. Meanwhile, four methods for feature importance ranking were applied to analyse the effects of different features on the model prediction results.

## 2    Background

### 2.1    The Stockholm air quality forecast system

Stockholm city has launched an air quality forecast system since 2021. Three different dispersion models are used to forecast concentrations considering emissions and dispersion at the European, Urban and Street scales described by Figure 1. The CAMS ensemble model, part of the Copernicus program was used to obtain forecasts of long-range transported air pollution
from outside of the Greater Stockholm area. Previous assessments have found the ensemble model to be more accurate than any individual model part of CAMS (Meteo-France, 2017; Marècal et al., 2015). CAMS regional ensemble forecasts are published once a day and each forecast covers 96 hours (4 days).

The contributions to concentrations due to local emissions in the metropolitan area were performed on a 100 m resolution using a Gaussian dispersion model part of the Airviro system (https://www.airviro.com/airviro/). In this modelling domain
(Greater Stockholm, 35 by 35 km) individual buildings and street canyons are not resolved but treated using a roughness parameter (Gidhagen et al., 2005). The Gaussian model is fed with meteorological forecasts from the Swedish Meteorological and Hydrological Institute (SMHI). A diagnostic wind model is used to account for influences of variations in topography and land-use on the dispersion parameters input to the Gaussian model. For details regarding uncertainties and validation of local modelling see (Johansson et al. 2017).
Finally, the Operational Street Pollution Model (OSPM), developed by Berkowicz (2000) and driven by forecasted meteorology from SMHI, is applied to the street canyon sites. It has been applied earlier at Hornsgatan in Stockholm in a number of modelling studies (e.g. Krecl et al., 2021; Ottosen et al., 2015). $NO_x$ and $PM_{10}$ are modelled on all scales, whereas $O_3$ is only forecasted by the CAMS ensemble model.

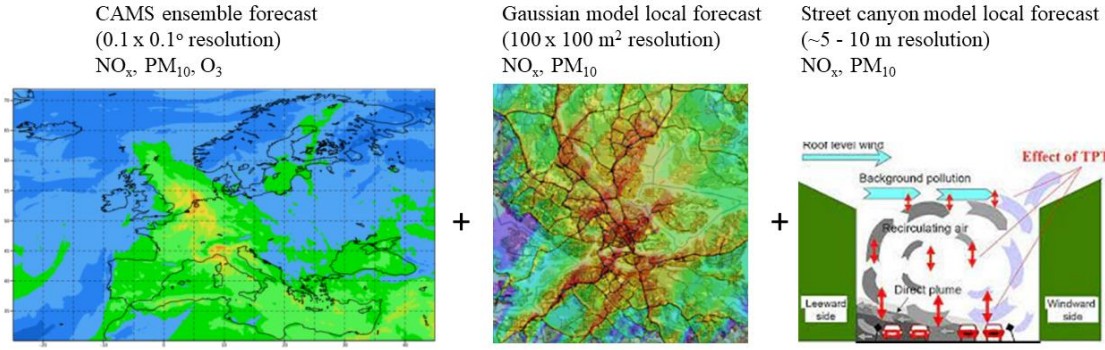

**Figure 1. Illustration of the deterministic modelling from European scale at a resolution of 0.1° by 0.1° (ca 11 km × 6 km), via urban scale (100 m resolution over an area of 35 by 35 km) down to the street canyon sites. The CAMS ensemble forecast map example is**

taken from https://atmosphere.copernicus.eu/ (accessed 1 Feb 2023). The map with the Gaussian model local forecast example is output from the Airviro system (https://www.airviro.com/airviro/, accessed 1 Feb 2023) used in Stockholm. The illustration of a street canyon site is taken from https://www.wikiwand.com/en/Operational_Street_Pollution_Model (accessed 1 Feb 2023).

For the urban scale model domain, a detailed emission database is used as input for the local dispersion modelling. The database and its applications and comparisons between modelling and measurements are described in (SLB, 2022). The total emissions from road traffic are based on emission factors for different vehicle types including passenger cars, buses, light- and heavy-duty trucks. Exhaust emission factors of $NO_x$ and particles are based on HBEFA version 3.3 (Keller et al., 2017) depending on vehicle Euro class. The emission factors per vehicle category were weighted according to the national Swedish Transport Administration vehicle registry, but the vehicle composition taken from national vehicle registry has been shown to be similar to the local fleet using real-world number plate recognition measurements at Hornsgatan (Burman and Johansson, 2010) (Burman et al., 2019). Non-exhaust emissions of PM due to wear of brakes, tyres and roads are calculated using the NORTRIP model (Denby et al., 2013) forced by the forecasted meteorology from SMHI. Information on shares of studded winter tyres is obtained from manual counting every week during the winter at different locations in the city centre and along highways outside of the city. Road traffic emissions are calculated for all roads with more than 3,000 vehicles per day. Other emission sources included in the local emissions database include shipping, private and municipal heating (including burning of waste). More information about the Stockholm air quality forecast system is provided in (Engardt et al. 2021).

## 2.2    Meteorological forecasts

As an integral part of the Stockholm air quality forecast system, meteorological forecasts for a point in central Stockholm are downloaded every morning from the websites of SMHI (https://www.smhi.se/data/oppna-data) and MET Norway (https://docs.api.met.no/doc/). The meteorological forecasts extend over 10 days and are a combination of output from a number of regional and global numerical weather prediction models. The combination is based on statistical adjustments as well as manual edits. Initial models of weather-dependent PM emissions and urban and street canyon air quality modeling are driven by meteorology. The forecasted meteorological data are, finally, also used as predictors for the models in this study.

## 3    Methods

### 3.1    Data and Pre-processing

The data used in this study was collected from four monitoring stations in central Stockholm, including one urban background site (Torkel Knutssonsgatan, hereafter called UB or urban) and 3 street canyon sites (Hornsgatan HO, Folkungagatan FO, and Sveavägen SV). They are all located in central Stockholm (see Figure 2). Detailed descriptions of measurement methods and sites are provided in Appendix A.

Data from the UB site covers approx. 1000 days (10 April 2019 through 31 December 2021). As the OSPM model became operational at a later date, the street canyon data extends over 500 days (5 August 2020 through 31 December 2021). Pollutant

concentration measurements from monitoring stations, pollutant forecasts and meteorological forecasts from the Stockholm air quality forecast system were aggregated into the following four datasets.

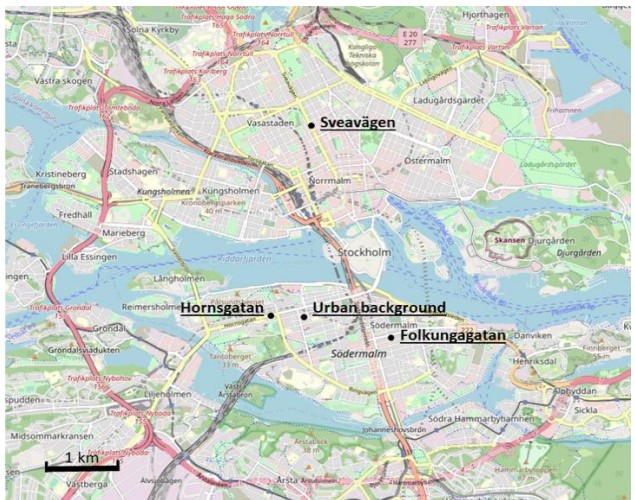

**Figure 2. Map of central Stockholm showing locations of the urban background site and the street canyons traffic sites. Base map credits: © OpenStreetMap contributors, licensed under the Open Data Commons Open Database License (ODbL) v1.0.**

All the data above was collected at 1-hour intervals, with details illustrated in Table 1. It should be noted that there are several studies show the impact of the COVID-19 pandemic on pollutant emissions as a result of some restrictive regulations (Sokhi et al., 2021; Torkmahalleh et al., 2021). The COVID-19 pandemic in Sweden commenced in January 2020 and continued until February 2022, so the majority of the data was collected during this pandemic period.

**Table 1. Description of the dataset.**

| Name | Time Range | Pollutants | Amount | Features |
|------|-----------|-----------|--------|----------|
| Urban Background UB | 04/10/2019 - 12/31/2021 | $NO_X$, $PM_{10}$, $O_3$ | 23927 | Pollutant measurements |
| Folkungagatan FO | 08/05/2020 - 12/31/2021 | $NO_X$, $PM_{10}$ | 12335 | Pollutant forecasts |
| Hornsgatan HO | 08/05/2020 - 12/31/2021 | $NO_X$, $PM_{10}$ | 12335 | |
| Sveavägen SV | 08/05/2020 - 12/31/2021 | $NO_X$, $PM_{10}$ | 12335 | Meteorological forecasts |

The pollutant measurements and forecasts from deterministic model exhibit a missing rate of less than 5%, with a few inaccurate samples, including outliers and negative values. Appendix B shows the missing status of $O_3$ in the UB dataset. To accurately represent the extreme values in the real world, outliers were deliberately included in the data because their occurrence is hard to justify. But negative pollutant samples were eliminated, and missing data was manually interpolated using historical average interpolation (Willmott et al., 1995).

Frequently employed approaches of interpolating time series data comprise constant interpolation, nearest neighbor interpolation, and linear interpolation. To keep the temporal relationship, the historical average interpolation is applied based on the periodicity pattern in the data. The periodicity of each feature, denoted by *p*, is determined by the analysis of the

autocorrelation function (ACF) and partial autocorrelation function (PACF) of the data. Subsequently, the missing value $\tilde{p}(t)$ at time $t$ is substituted by the average of the available data from the two preceding periods as well as their adjacent values:

$$\tilde{p}(t) = \frac{1}{n}\sum (\tilde{p}(t-p), \tilde{p}(t-p \pm 1), \tilde{p}(t-2p), \tilde{p}(t-2p \pm 1)) \tag{1}$$

where $n$ is the number of samples used in Equation 2. An example result of interpolation is shown in Appendix B.

## 3.2 Prediction scheme

This study is to forecast hourly concentrations for the coming one, two and three days based on historical pollutant measurements and other available information as inputs, which is a time series prediction for multiple time steps, for example, 72 time steps for three days prediction. Instead of more complex network structure, multiple single-output ML models are chosen for forecasting different air pollutants for $k$=1 day, 2 day and 3 day intervals, as shown in Equation 2.

$$\hat{\rho}_{i,j}(d,t) = ML\_model\left(\tilde{\rho}_{i,j}(d-k,t), \bar{\rho}_{i,j}^{S}(d-k,t), \check{\rho}_{i,j}(d,t), W(d,t), C(d,t)\right) \tag{2}$$

where $\hat{\rho}_{i,j}(d,t)$ is the forecast of the pollutant $j$ for day $d$ and time $t$ at the location $i$, and $\tilde{\rho}_{i,j}(d,t)$ is the corresponding real measurement; $\bar{\rho}_{i,j}^{S}(d,t)$ uses a set S to represent several statistical measures, including maximum, minimum, 25% quantile and 75% quantile of the measured concentration data during the past 24 hours until $t$, and the measurement dataset can be represented by a set, i.e. $\{\tilde{\rho}_{i,j}(d,t), \tilde{\rho}_{i,j}(d,t-1), \tilde{\rho}_{i,j}(d,t-2)....\}$. $\check{\rho}_{i,j}(d,t)$ is the predicted concentration using deterministic model. $W(d,t)$ represents the weather condition predicted for day $d$ and time $t$.

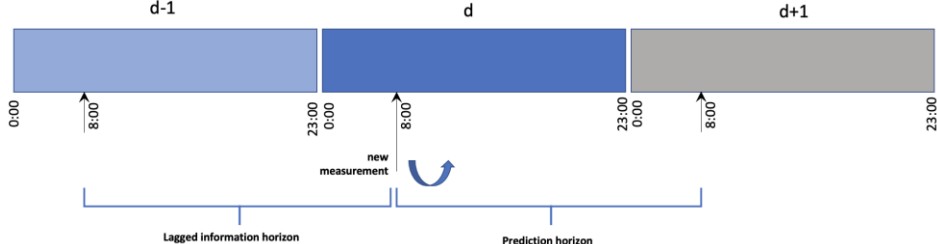

**Figure 3. Illustration of the machine learning modelling scheme for 1-day prediction based on available datasets.**

Figure 3 demonstrates the prediction horizon and lagged information horizon for the case of one day prediction. To build consistent statistical ML models with a fixed rolling horizon, a new measurement point at the current time (d, t) will lead to an additional prediction for one day ahead, i.e. the predicted value at (d+1,t). In this case, the measurement statistics $\bar{\rho}_{i,j}^{S}(d,t)$ will be based on one day preceding measurement data of (d, t), resulting in a lagged rolling horizon described by Figure 3.

## 3.3 Machine learning models

As already mentioned before, two tree-based ML models, RF and XGB, and one deep learning model, LSTM are applied to implement the prediction scheme. In addition, an ensemble learning approach based on a General Additive Model (GAM), aggregating the selected three learning models, is also applied to further optimise the results.

### 3.3.1 Framework

Figure 4 summarises the framework of ML models and associated computational experiments for air pollution prediction. The input includes the deterministic forecasts of $PM_{10}$, $NO_x$ and $O_3$, to evaluate how much the deterministic forecasts can be improved by the ML algorithms. In the computational experiments, data-driven forecasting models are trained for one urban background site and three street canyon sites separately. Different ML models are trained and tested separately for predicting various air pollution concentrations in future periods, i.e. 1-day (0 – 24 h), 2-day (25 – 48 h) and 3-day (48 – 72 h).

To make a fair comparison with all models, a vanilla LSTM model in this case is set up to take the same type of input as the other two models. In addition to the measured air pollution time series data itself, the forecasted meteorological conditions for the prediction day $d$ (or $d+1$ or $d+2$) and calendar information such as weekday, hour, etc. are also applied as input features. Moreover, the air pollutant concentrations predicted by the deterministic models are also used as inputs to the ML models.

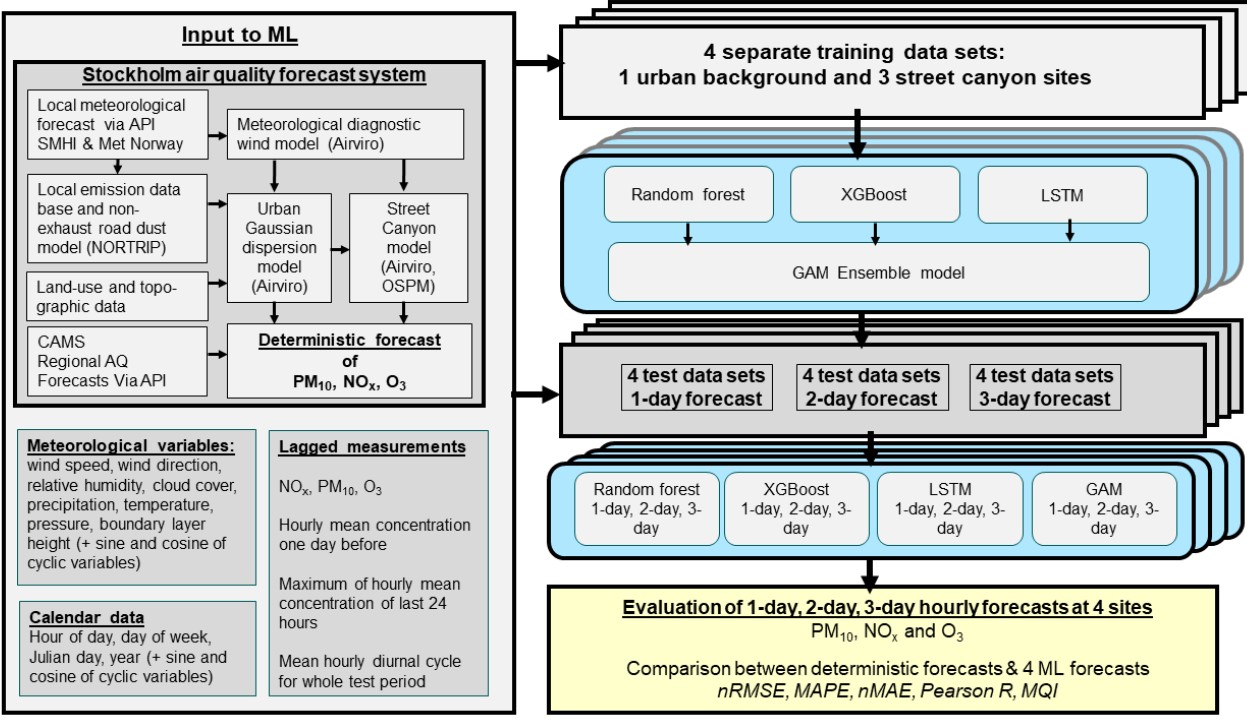

**Figure 4. Illustration summarising input data for modelling 1-, 2- and 3-day forecasts of $PM_{10}$, $NO_x$ and $O_3$ using the 4 models.**

Table 2 presents a detailed explanation of the essential input features that are applied in the computational experiments. During feature engineering, new features are constructed through statistical analysis to expand the feature space and facilitate context extraction. At the same time, temporal attributes are decomposed and encoded to the dataset to reflect the temporal dependence of each sample.

**Table 2. Measured and forecasted air pollutant concentrations used as input data (features) in the ML modelling of pollutant concentrations at the urban background site (UB) and at the street canyon sites (SC). For periodic input data, using sine and cosine values can remove discontinuities and create consistent distance measures, thereby improving model accuracy.**

| Category | Short names | Description |
|---|---|---|
| Deterministic features | $NO_x\_nday\_local$ $PM_{10}\_nday\_local$ **n=1, 2, 3** | Deterministic 1-day, 2-day and 3-day forecast of contributions from local emissions based on urban scale Gaussian modelling |
| | $NO_x\_nday\_regional$ $PM_{10}\_nday\_regional$ $O_3\_nd\_regional$ **n=1, 2, 3** | Deterministic 1-day, 2-day and 3-day forecast of contributions based from non-local emissions based on CAMS ensemble model (regional background) |
| Autocorrelation features | $NO_x\_lagXX$ $PM_{10}\_lagXX$ $O_3\_lagXX$ **XX = 24, 48, 72** | XX hour lagged air pollutant concentrations based on autocorrelation and prediction time span. |
| Statistical features | $NO_x\_Sta\_dXX$ $PM_{10}\_Sta\_dXX$ $O_3\_Sta\_dXX$ **Sta=avg., median, min, max, Q1, Q3** **XX = 24, 48, 72** | Average, median, minimum, maximum, quantiles 1 and quantiles 3 of lagged air pollutant concentrations in rolling XX hour periods. |
| Time features | **Time; Time_sin; Time_cos** **Time=year, julianday, month, weekday, day, hour** | Julian day of the year (1, 2, 3, … 365), sine and cosine of 2\*pi\*day/365. Day of the week (1, 2, 3, … 7), sine and cosine of 2\*pi\*day/7. Hour of the day (0, 1, 2, … 23), sine and cosine of 2\*pi\*hour/24. Year, Month Day |
| Meteorological features | wind_direction wind_direction_cos wind_direction_sin | Wind direction[0, 360) at 10 m in central Stockholm, sine and cosine of (2\*pi/360)\*wind direction |
| | pressure; temperature; precipitation; cloudiness | Pressure (10 m); Temperature (10 m) |
| | wind_speed | Wind speed (10 m) |
| | relative_humidity | Relative humidity |
| | boundary_layer_height | Boundary layer height for central Stockholm |

### 3.3.2 Model setups

5 All ML models are implemented in *python* using existing libraries including "*scikit-learn*"(Bisong et al., 2019) and "*pytorch*" (Paszke et al., 2019) for conventional ML models and deep learning models respectively. The detailed implementation can be referred to the open-source code provided in (Zhang & Ma, 2023).

The following configurations are applied as the initial models:

- The initial parameters of the two tree-based models (XGB and RF) are the default parameters of "*scikit learn*", and
10      the tuned parameters are presented in Appendix C.

- The LSTM model architecture consists of two layers of LSTM with 100 neurons and a fully connected layer before the output. The activation function was "*tanh*".

- The LSTM model was trained by *Adam* optimizer. The initial learning rate is 0.01 and is dynamically changed using "*ReduceLROnPlateau*" algorithm, with the parameter patience of 10, which means that the algorithm will monitor

the performance (e.g., validation loss) for 10 consecutive epochs. If there is no improvement, the learning rate will be reduced according to the specified reduction strategy. Also, the initial batch size is set as 72.

The data is split along the time axis with a ratio of 16:4:5 to achieve non-overlapping among training, validation, and test data. Due to the autocorrelation of the air pollutant data, the assumption of independent and identically distributed classical cross-validation is not satisfied. Therefore, to preserve the time-dependent property, the function "*TimeSeriesSplit*" in "*scikit-learn*" was chosen as the cross-validation method. In the $k_{th}$ split, the data of the first k folds are set as the training data whereas the data of the $(k+1)_{th}$ fold is the test set. Empirically, the value of k is set to be 5.

Given the inherent uncertainty of the ML models, they are trained by setting different random seeds. Therefore, the final results are presented in terms of statistical means and their confidence intervals, which provide a consistent way to evaluate the robustness of the prediction models. The number of repeated training processes in our experiment is set to 10 for each model.

## 3.4 Hyperparameter optimisation

The grid and greedy search approaches are combined in the hyperparameter tuning process to balance the model optimality and computational cost (Liashchynskyi et al., 2019). The grid search allows for a systematic investigation of different combinations of hyperparameters, whereas the greedy approach searches local optimum for a certain variable iteratively.

Table 3 depicts the strategies of parameter optimization when training the ML models. For each model, a tuning strategy is represented by a combination of grid search (the searching dimensions are described in {}) and greedy search (the search sequence is presented by →). The parameter search space and optimal parameter combinations are presented in Appendix C. For XGB and RF, the most influencing parameters are the number of evaluators (*n_estimators*), the number of input features (*max_features*), and the learning rate. So, a grid search is first applied to identify an optimal combination of those parameters. Appendix C shows the results of grid search for *n_estimators* and *learning_rate*. The search spaces for *n_estimators* and *learin_rate* are set to 9 and 12 respectively, resulting in a total of 108 grid points. The optimal model performance is achieved in (60, 0.03). Subsequently, the greedy search strategy is applied sequentially to find the suboptimal combination of the parameters. The model performance is evaluated according to the mean squared error (MSE) on the validation set. For LSTM model, only greedy search strategy is applied to optimise the parameters sequentially due to the large search space and computational cost for training LSTM model.

**Table 3. Hyperparameter tuning method and process.**

| Models | Hypterparameter tuning strategy[*] |
|---|---|
| XGBoost | {n_estimators, learning_rate} → max_depth → subsample → colsample_bytree → min_child_weight |
| RandomForest | {n_estimators, max_features} → max_depth → min_samples_split → min_samples_leaf |
| LSTM | batch_size → n_steps_in → hidden_size → learning rate |

[*] {} represents the dimension of grid search and → represents greedy search sequence

### 3.5 Feature importance ranking

ML models used in our study are black-box models, and feature importance analysis plays a key role in understanding the model behavior and improvement. Feature analysis is carried out by calculating an importance score for each individual feature to quantitatively evaluate how much a feature may contribute to the forecasts.

For tree-based models, three methods, namely Mean Decrease in Impurity (MDI), Permutation method, and Shapley Additive exPlanations (SHAP), are used for feature ranking. For LSTM models, the gradient-based method, Permutation, and SHAP were frequently employed. Below is a simple explanation of the feature ranking methods for the ML models:

     1)   Mean Decrease in Impurity

Mean Impurity Decrease (MDI) is a popular feature importance analysis for tree-based models, such as RF. The

implementation of the method is integrated into "*scikit-learn*". It calculates the average reduction of impurities by the inclusion of a particular feature as the importance score of this feature. However, the computation of impurity-based importance is based on the training data, so it does not accurately reflect the performance of the features for the test set (Bisong et al., 2019).

     2)   Permutation

The permutation method is defined as the decrease of a model performance when a single feature value is randomly shuffled

(Breiman, 2001). For the data used in this study, it can be applied to tree-based models but also to neural networks like LSTM. The computation of feature scores allows for the consideration of the impacts of various features on the model prediction capacity. The method has benefit of circumventing the concerns about the tendency of MDI to favor high cardinality features.

     3)   Gradient-based method

Gradient-based method explains the local relationship between inputs and outputs by harnessing the gradients of the model

prediction with respect to input features as an importance score (Baehrens et al., 2010). It should be noted that the gradients of neural networks depend on both input and output data, and the feature importance for the LSTM model was computed as the average of feature gradient obtained from all samples in test data.

     4)   SHAP

Shapley Additive exPlanations (SHAP) is a general explanatory framework, in which SHAP values represent the average

marginal contribution of each feature towards the difference between the model's prediction and a reference prediction. The greatest strength of SHAP is its ability to reflect the influence of each feature on each sample, which is interpreted as a positive or negative influence. The SHAP is an interpretation scheme for almost all ML models. This study uses the *Python* library *shap* to evaluate tree-based models and LSTM respectively (Lundberg et al., 2017; Shrikumar et al., 2017).

### 3.6 Statistical performance indicators

Several performance metrics have been selected for comparing the prediction results of different ML models including R-squared ($R^2$), mean square error(MSE) and normalized error measures: mean average error (MAE), mean absolute percentage error (MAPE), root mean squared error (RMSE) and Pearson correlation(Pearson). These measures have also

been recommended for air quality model benchmarking in the context of the Air Quality Directive 2008/50/EC (AQD) by Janssen and Thunis (2022).

**Table 4. Performance indicators.**

| Indicators | Formula | Indicators | Formula |
|---|---|---|---|
| **R²** | $R^2(y,\hat{y}) = 1 - \dfrac{\sum_{i=1}^{n}(y_i - \hat{y}_i)^2}{\sum_{i=1}^{n}(y_i - \bar{y})^2}$ | **Mean Square Error** | $MSE(y,\hat{y}) = \dfrac{1}{n}\sum_{i=1}^{n}(y_i - \hat{y}_i)^2$ |
| **Mean absolute percentage error** | $MAPE(y,\hat{y}) = \dfrac{1}{n}\sum_{i=1}^{n}\dfrac{|y_i - \hat{y}_i|}{|y_i|}$ | **Root Mean Square Error** | $RMSE(y,\hat{y}) = \sqrt{\dfrac{1}{n}\sum_{i=1}^{n}(y_i - \hat{y}_i)^2}$ |
| **Pearson correlation** | $Pearson(y,\hat{y}) = \dfrac{\sum_{i=1}^{n}(y_i - \bar{y_i})(\hat{y}_i - \bar{\hat{y}_i})}{\sqrt{\sum_{i=1}^{n}(y_i - \bar{y_i})^2}\sqrt{\sum_{i=1}^{n}(\hat{y}_i - \bar{\hat{y}_i})^2}}$ | | |

Note: $\hat{y}_i$ is the predicted value of the $i$-th sample, $y_i$ is the corresponding true value, and $\bar{y}$ is the mean value of all $n$ samples. MAE and RMSE were normalized by diving by the mean of the measured concentrations, hereafter called nMAE and nRMSE.

In addition, to properly assess model quality, it is necessary to consider measurement uncertainty. In the Forum for Air Quality Modeling, the modelling quality indicator (MQI) is used to assess if a model fulfils certain objectives (Janssen and Thunis, 2022). It is defined as the ratio between the model bias at a fixed time (i), quantified by the RMSE, and a quantity proportional to the measurement uncertainty as:

$$MQI(i) = \frac{\sqrt{\dfrac{1}{n}\sum_{i=1}^{n}(y_i - \hat{y}_i)^2}}{\beta\sqrt{\dfrac{1}{n}\sum_{i=1}^{n}U(y_i)^2}} = \frac{RMSE}{\beta RMS_U}$$

Where $U(y_i)$ is the expanded 95th percentile measurement uncertainty and $\beta$ is a coefficient of proportionality (Janssen and Thunis, 2022). The value of $\beta$ determines the stringency of the MQI and is set equal to 2, allowing thus deviation between modelled and measured concentrations as twice the measurement uncertainty. The uncertainty of the measurements ($RMS_U$) was calculated for the mean of the measurement concentrations as:

$$U(y_i) = U_r(RV)\sqrt{(1 - \alpha^2)y_i^2 + \alpha^2 RV^2}$$

Where $U_r(RV)$ and $\alpha$ are parameters that depend on pollutant and RV is a reference value, here taken to be 200, 50 and 120 µg m$^{-3}$, corresponding $U_r(RV)$ was 0.24, 0.28 and 0.18 and $\alpha$ was 0.20, 0.25, 0.79 for NO$_2$, PM$_{10}$ and O$_3$ respectively (Janssen and Thunis, 2022). In our case we have calculated NO$_x$, not NO$_2$, but we used the same settings of the parameters for NO$_x$ as recommended for NO$_2$.

## 4 Computational Results

The focus of this paper is to compare the deterministic forecasts of $NO_x$, $PM_{10}$ and $O_3$ with the forecasts based on the different machine learners which also include the deterministic forecasts as input variables (features). As described above we have made deterministic and ML forecasts for hourly mean concentrations for the coming 72 hours, based on 1-day, 2-day and 3-day meteorological forecasts for one urban background site ($NO_x$, $PM_{10}$ and $O_3$) and three street canyon sites ($NO_x$ and $PM_{10}$). We also compare results separately for the urban background site and the street canyon sites.

### 4.1 Urban background

#### 4.1.1 Comparison between deterministic forecasts and ML models - urban background

As illustrated in Table 5 and Figure 5, all statistical performance measures of the deterministic forecasts are improved by the ML models for the pollutants: NOx, $PM_{10}$ and $O_3$. The statistical mean and 95% confidence intervals are estimated from 10 repeated computational experiments using 10 different random seeds.

Table 5. summarises the prediction performance of both deterministic and ML models in terms of five selected metrics. For NOx, the $R^2$ value increases, from a range between 0.12 and 0.22 for the deterministic forecasts to a range between 0.33 and 0.42 achieved by ML models. The other four metrics, including MAPE, nRMSE, nMAE and MSE, decrease for all forecasting days. The LSTM model achieves superior performance for almost all the metrics, and XGBoost performs closely in this case. For $PM_{10}$, $R^2$ increases, from the range of 0.08-0.21 in the deterministic forecasts, to higher values between 0.28 and 0.55 using ML models. Again, there are big reductions on the other four performance measures, among which MSE is decreased by 45% compared to deterministic forecasts. XGB and RF models are the winners with comparable performance.

For $O_3$ there is about a 40% drop in MSE for tree-based models, with slight improvements on other metrics for all forecasting days. LSTM also performs equally well and achieves remarkable performance for the 3-day prediction. While the errors of deterministic CAMS modelling for $O_3$ are quite small when compared to the prediction of NOx and $PM_{10}$, MLs demonstrate their capacities to further refine the pollutant prediction.

The width of the confidence interval indicates the reliability of the model prediction results. The two tree-based models (XGB and RF) produce a very small variance, less than 1%, whereas the LSTM model exhibits a higher variance but less than 5%. The higher variance of LSTM model may be due to the random initialization of the weights, which affects the subsequent gradient descent trajectory and model results.

**Table 5.** Comparison of 1-, 2-, 3-day deterministic and ML forecasts for NOx, $PM_{10}$ and $O_3$ for the urban background site. $R^2$ = R-Squared, MAPE = mean absolute percentage error, nRMSE = normalised root mean square error, nMAE = normalised mean absolute error and MSE=mean square error. The average performances with their 95% confidence interval were computed on the test set from 10 experimental repetitions conducted with different random seeds, and the best performances are bold.

**$NO_x$**

| | $R^2$ | | | MAPE | | | nRMSE | | | nMAE | | | MSE | | |
|---|---|---|---|---|---|---|---|---|---|---|---|---|---|---|---|
| | 1-day | 2-day | 3-day | 1-day | 2-day | 3-day | 1-day | 2-day | 3-day | 1-day | 2-day | 3-day | 1-day | 2-day | 3-day |
| Det | 0.13 | 0.22 | 0.12 | 0.72 | 0.73 | 0.88 | 1.27 | 1.20 | 1.28 | 0.61 | 0.60 | 0.69 | 229.77 | 205.21 | 233.25 |
| XGB | 0.30 ± 0.01 | 0.30 ± 0.01 | 0.30 ± 0.00 | **0.37 ± 0.00** | **0.39 ± 0.00** | **0.39 ± 0.00** | 1.14 ± 0.00 | 1.14 ± 0.00 | 1.14 ± 0.00 | **0.40 ± 0.00** | **0.41 ± 0.00** | **0.41 ± 0.00** | 184.19 ± 1.58 | 185.93 ± 1.46 | 185.91 ± 1.18 |
| RF | 0.27 ± 0.00 | 0.27 ± 0.00 | 0.27 ± 0.00 | 0.48 ± 0.00 | 0.50 ± 0.00 | 0.50 ± 0.00 | 1.17 ± 0.00 | 1.17 ± 0.00 | 1.17 ± 0.00 | 0.44 ± 0.00 | 0.45 ± 0.00 | 0.45 ± 0.00 | 192.87 ± 0.53 | 194.5 ± 0.71 | 194.66 ± 0.89 |
| LSTM | **0.33 ± 0.05** | **0.41 ± 0.03** | **0.42 ± 0.02** | 0.44 ± 0.06 | 0.41 ± 0.03 | 0.41 ± 0.03 | **1.12 ± 0.04** | **1.04 ± 0.03** | **1.04 ± 0.02** | 0.44 ± 0.02 | 0.43 ± 0.02 | 0.42 ± 0.02 | 178.32 ± 12.89 | **155.12 ± 8.43** | **153.57 ± 6.19** |
| GAM | **0.33 ± 0.01** | 0.30 ± 0.01 | 0.34 ± 0.01 | 0.43 ± 0.01 | 0.45 ± 0.01 | 0.45 ± 0.00 | **1.12 ± 0.01** | 1.14 ± 0.01 | 1.11 ± 0.01 | 0.43 ± 0.00 | 0.44 ± 0.00 | 0.44 ± 0.00 | **176.48 ± 2.52** | 184.91 ± 2.60 | 176.21 ± 2.82 |

**$PM_{10}$**

| | $R^2$ | | | MAPE | | | nRMSE | | | nMAE | | | MSE | | |
|---|---|---|---|---|---|---|---|---|---|---|---|---|---|---|---|
| | 1-day | 2-day | 3-day | 1-day | 2-day | 3-day | 1-day | 2-day | 3-day | 1-day | 2-day | 3-day | 1-day | 2-day | 3-day |
| Det | 0.21 | 0.13 | 0.08 | 0.54 | 0.56 | 0.63 | 0.67 | 0.70 | 0.72 | 0.46 | 0.48 | 0.51 | 41.67 | 45.78 | 48.65 |
| XGB | **0.55 ± 0.00** | **0.49 ± 0.01** | **0.41 ± 0.01** | 0.47 ± 0.01 | 0.53 ± 0.01 | 0.57 ± 0.01 | **0.50 ± 0.00** | **0.53 ± 0.00** | **0.58 ± 0.00** | **0.35 ± 0.00** | **0.38 ± 0.00** | **0.40 ± 0.00** | **23.7 ± 0.26** | **26.75 ± 0.31** | **31.25 ± 0.28** |
| RF | **0.55 ± 0.00** | 0.42 ± 0.00 | 0.37 ± 0.00 | 0.44 ± 0.00 | 0.48 ± 0.00 | 0.52 ± 0.00 | 0.51 ± 0.00 | 0.57 ± 0.00 | 0.60 ± 0.00 | **0.35 ± 0.00** | 0.39 ± 0.00 | **0.40 ± 0.00** | 24.07 ± 0.07 | 30.63 ± 0.16 | 33.59 ± 0.12 |
| LSTM | 0.37 ± 0.04 | 0.39 ± 0.04 | 0.28 ± 0.04 | 0.57 ± 0.10 | 0.52 ± 0.04 | 0.55 ± 0.05 | 0.60 ± 0.02 | 0.59 ± 0.02 | 0.64 ± 0.02 | 0.43 ± 0.02 | 0.41 ± 0.01 | 0.44 ± 0.01 | 33.41 ± 1.99 | 32.13 ± 2.34 | 38.38 ± 1.88 |
| GAM | 0.53 ± 0.01 | 0.36 ± 0.01 | 0.33 ± 0.01 | **0.43 ± 0.01** | **0.47 ± 0.00** | **0.50 ± 0.01** | 0.52 ± 0.00 | 0.60 ± 0.00 | 0.62 ± 0.00 | 0.36 ± 0.00 | 0.41 ± 0.00 | 0.42 ± 0.00 | 24.97 ± 0.37 | 33.72 ± 0.43 | 35.41 ± 0.34 |

**$O_3$**

| | $R^2$ | | | MAPE | | | nRMSE | | | nMAE | | | MSE | | |
|---|---|---|---|---|---|---|---|---|---|---|---|---|---|---|---|
| | 1-day | 2-day | 3-day | 1-day | 2-day | 3-day | 1-day | 2-day | 3-day | 1-day | 2-day | 3-day | 1-day | 2-day | 3-day |
| Det | 0.38 | 0.32 | 0.19 | 0.48 | 0.54 | 0.59 | 0.31 | 0.32 | 0.35 | 0.24 | 0.25 | 0.28 | 210.07 | 231.27 | 276.85 |
| XGB | **0.65 ± 0.00** | **0.58 ± 0.00** | 0.54 ± 0.00 | 0.40 ± 0.00 | 0.44 ± 0.00 | 0.46 ± 0.00 | **0.23 ± 0.00** | **0.25 ± 0.00** | 0.27 ± 0.00 | **0.18 ± 0.00** | **0.20 ± 0.00** | 0.21 ± 0.00 | **121.14 ± 0.56** | **143.44 ± 1.55** | 157.89 ± 1.35 |
| RF | 0.62 ± 0.00 | 0.52 ± 0.00 | 0.50 ± 0.00 | 0.42 ± 0.00 | 0.48 ± 0.00 | 0.47 ± 0.00 | 0.24 ± 0.00 | 0.27 ± 0.00 | 0.28 ± 0.00 | 0.19 ± 0.00 | 0.21 ± 0.00 | 0.21 ± 0.00 | 129.52 ± 0.41 | 164.16 ± 0.38 | 169.71 ± 0.34 |
| LSTM | 0.62 ± 0.01 | 0.57 ± 0.02 | **0.59 ± 0.01** | **0.39 ± 0.01** | **0.42 ± 0.01** | **0.40 ± 0.01** | 0.24 ± 0.00 | 0.26 ± 0.01 | **0.25 ± 0.00** | 0.19 ± 0.00 | **0.20 ± 0.00** | **0.20 ± 0.00** | 131.05 ± 3.84 | 146.17 ± 6.2 | **139.51 ± 3.09** |
| GAM | 0.56 ± 0.00 | 0.43 ± 0.00 | 0.42 ± 0.00 | 0.42 ± 0.00 | 0.50 ± 0.00 | 0.49 ± 0.00 | 0.26 ± 0.00 | 0.30 ± 0.00 | 0.30 ± 0.00 | 0.20 ± 0.00 | 0.23 ± 0.00 | 0.23 ± 0.00 | 151.39 ± 0.69 | 195.4 ± 0.85 | 196.78 ± 1.19 |

Figure 5 presents statistical mean of 1-day, 2-day and 3-day forecasts by ML and deterministic models. Overall, all the performance metrics, including MQI and Pearson correlation, are consistently improved by ML models for three pollutants, $NO_X$, $PM_{10}$ and $O_3$. The difference in performance metrics achieved by different ML models is less than 30%.

All MQI results are below 100%, indicating that the deviation between model results and measurements is smaller than the estimated uncertainties of the measurements. XGBoost seems more efficient in reducing MQI, from 66% to 52% for $PM_{10}$. The LSTM model shows a reduction of around 10% on MQI for both NOx and $O_3$. The Pearson correlation reveals similar behavior to the $R^2$ but represents a more pronounced enhancement on improvement.

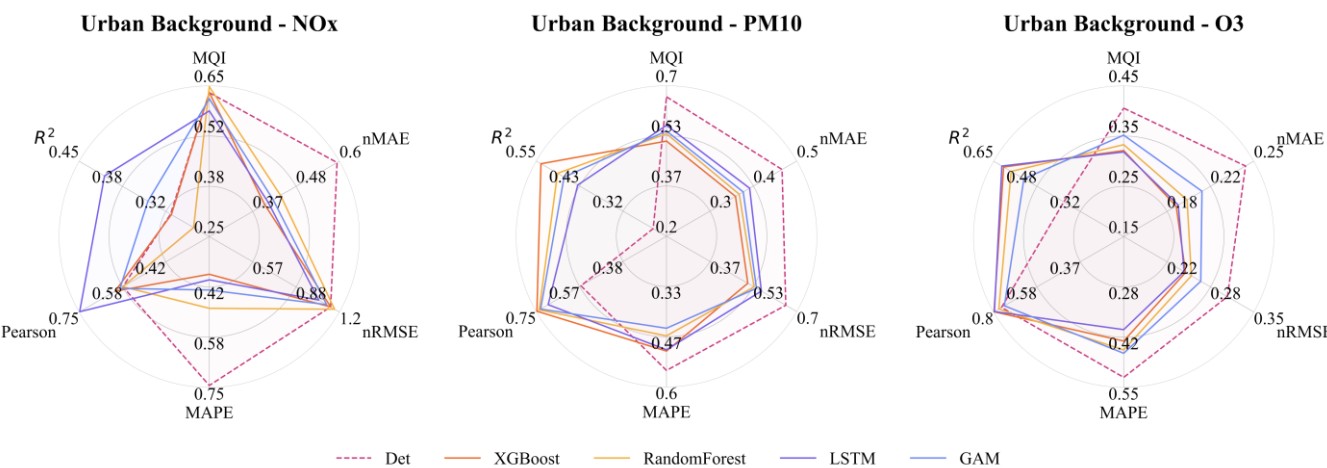

**Figure 5. Statistical performances for ML models and the deterministic hourly forecasts for the urban site. Mean of 1-day, 2-day and 3-day forecasts. Note that the ranges are different for different metrics.**

10  Figure 6(a) shows an example time series plot of the forecasts by the GAM and deterministic models during September 2021. Similar plots are also demonstrated for other models in Appendix D According to the figures, the ML models show better performance in capturing the trends and variation of measured pollutant concentrations, compared to the deterministic forecasts, although they still have obvious deviations from the real measurement. None of the models performs well in capturing the peaks of $PM_{10}$, e.g. on 30th September. Figure 6(b) demonstrates an example time series plot of the difference

15  between the forecasted concentrations of three pollutants, NOx, $PM_{10}$ and $O_3$, predicted by both deterministic and ML models, and the real observation. The graphs illustrate that during some hours all models systematically show large absolute deviations from the observed mean concentrations. Sometimes the hours with large deviations for NOx coincide with deviations for $PM_{10}$ indicating some specific meteorological situation or common source that caused this deviation.

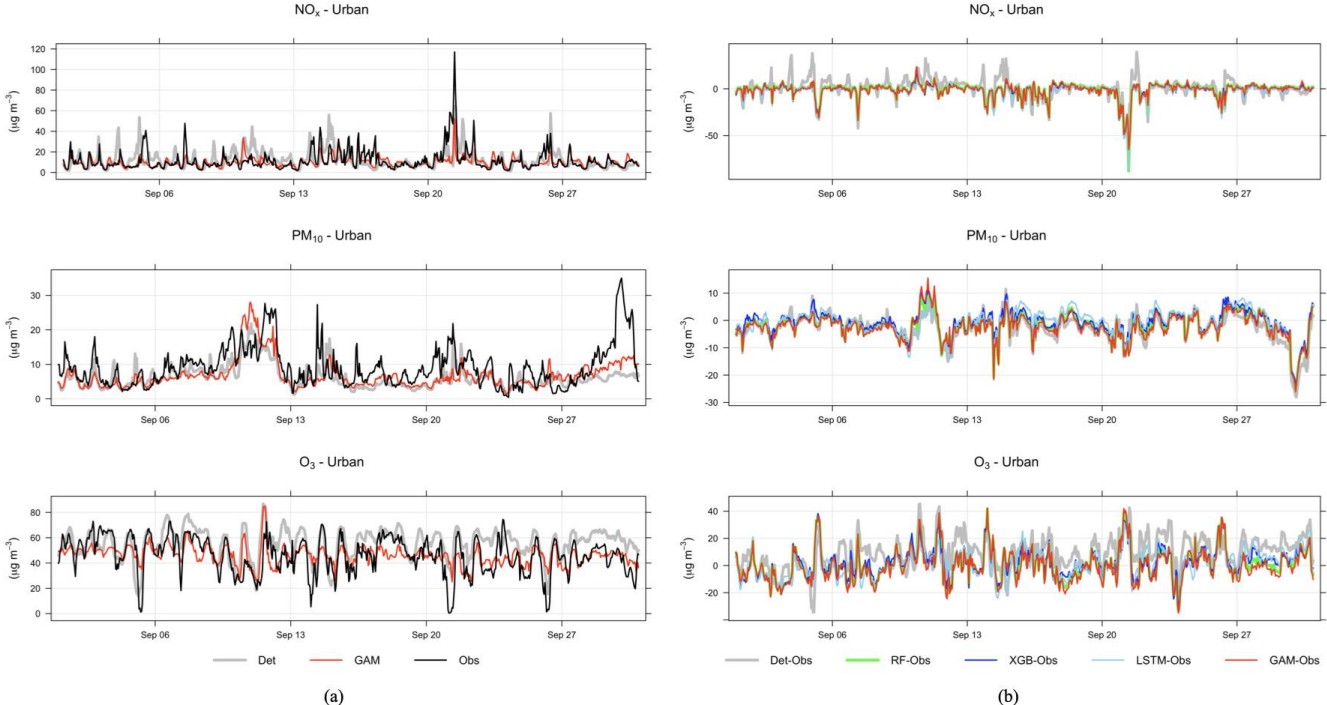

**Figure 6. (a) Temporal variations of hourly mean concentrations of NO$_x$, PM$_{10}$ and O$_3$ at the urban background site during September 2021 based on mean of 1-, 2- and 3-day forecasts for observations, deterministic forecasts and GAM. (b) Absolute deviations of forecasted NO$_x$, PM$_{10}$ and O$_3$ concentrations from observed (Obs) concentrations based on mean of 1-, 2- and 3-day forecasts for September 2021. All data are hourly mean concentrations.**

Systematic deviations between the observed mean diurnal variations and the deterministic forecast are shown in Appendix D. The deterministic forcasts are significantly improved using the ML models, especially for NO$_x$ and O$_3$. For O$_3$ the deterministic forecast systematically overestimates the concentrations which is mainly due to the fact that the chemical destruction of O$_3$ in the city centre is not properly accounted for by the regional CAMS model. For NO$_x$, the concentrations calculated by the deterministic model are systematically shifted one hour compared to the observed concentration and this is likely associated with errors in parameterisation of traffic emissions, which is the most important source of NO$_x$ in Stockholm. For PM$_{10}$ concentrations modelled by the deterministic model are too low during the night compared to observations, but this is corrected using RF and XGB, but not using GAM.

For the general public, it is important to receive information on future pollution episodes with high concentrations. The plots in Appendix E show that statistical performances for all models are worse when concentrations are higher than when the mean value is analysed. R$^2$ is somewhat higher for O3, while NO$_X$ and PM$_{10}$ decreased significantly, with the LSTM model having a relatively higher among all models for NO$_X$ and the XGBoost for PM10. nRMSE showed a similar trend to R$^2$.

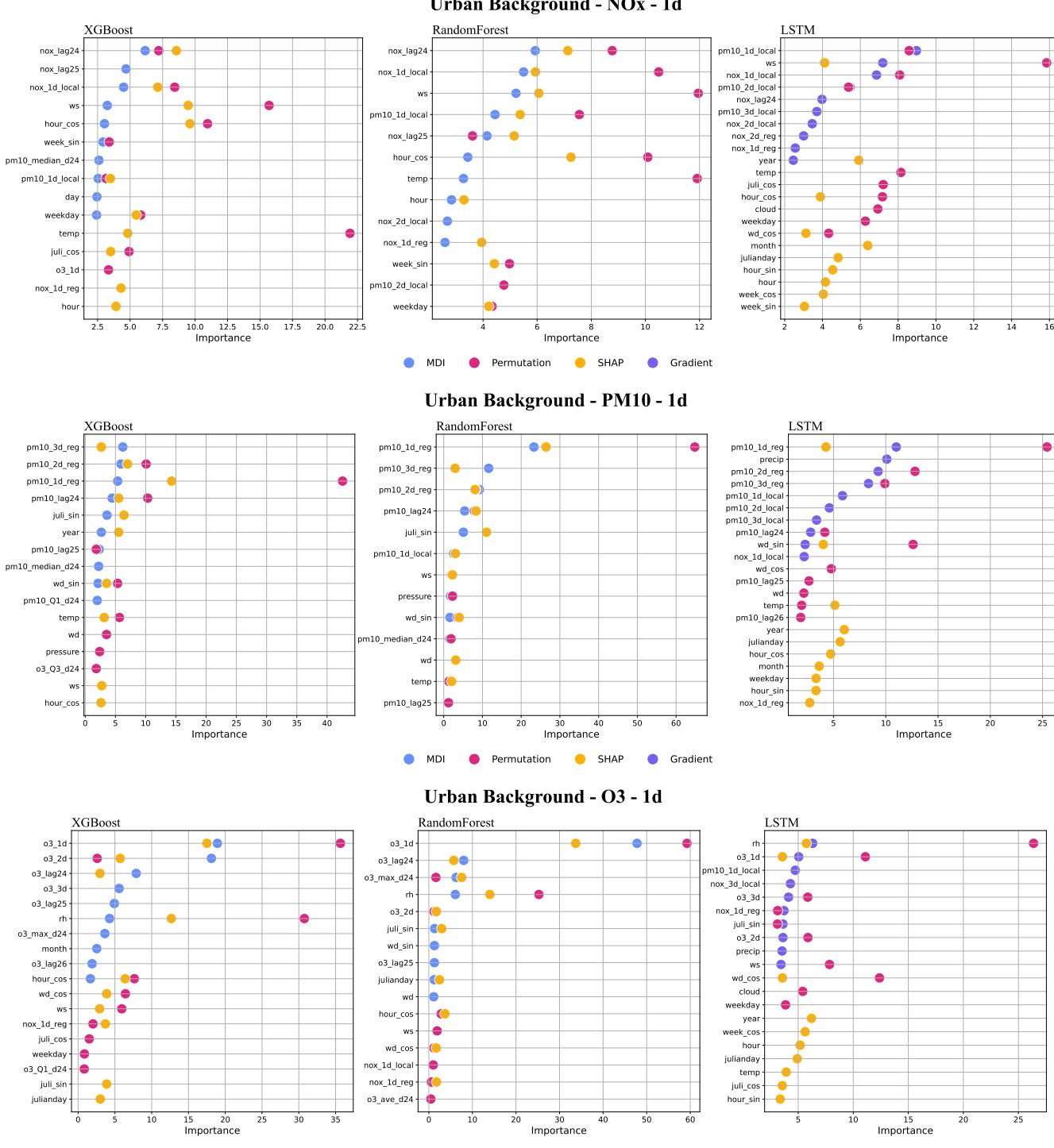

**Figure 7. Top 10 important features (%) of all 1-day forecasting models, XGB, RF and LSTM, for the urban site. All data are hourly mean concentrations.**

#### 4.1.2 Importance of features - urban background

Figure 7 presents the top 10 features obtained by the four feature ranking methods i.e., MDI, Gradient-based, permutation and SHAP. More detailed plots of feature importance ranking are shown in Appendix F, including the results of all models (RF, XGB and LSTM), for all three pollutants ($PM_{10}$, $NO_x$, $O_3$) and for all three forecasting periods (1-day, 2-day and 3-day). It should be noted that the local deterministic models, both Gaussian and OSPM models, use the same meteorological data to forecast hourly pollutant concentrations. So, when the meteorological variables are important features for the ML models, it indicates that the deterministic models don't capture all hidden processes related to those factors. Regarding feature importance ranking for urban background model, we have the following findings:

1) For $NO_x$ model, the factors, including temperature, wind speed, calendar data, lagged 24-hour mean concentrations, and local deterministic forecasts, are among the top 10 important variables, but the deterministic forecast is not the most important feature for any model. Among the calendar features, hour is the most important factor, indicating the importance of regular, diurnal variations of traffic emissions. Since both XGB and RF are decision tree-based algorithms, the top 10 features selected by the three feature ranking methods are basically the same, however, for LSTM, different features are extracted. Among all models, only the permutation model raises the importance of the deterministic forecasts of $O_3$ and $PM_{10}$, which reflect the fact that $O_3$ production is dependent on the status of $NO_X$ (Hagenbjörk et al., 2017) and compensate for the results of other methods of feature importance.

2) Regarding $PM_{10}$, the regional deterministic forecast is the most important feature of all models. Among the meteorological factors, both wind direction and pressure show their importance for prediction. The seasonal variation is reflected in the importance of the Julian day. For LSTM, precipitation shows their high importance, indicating the dependence of suspension of dust on surface wetness not being captured by the deterministic forecasts. For redundant features such as hour_sin and hour_cos, the permutation method may calculate lower importance values for both features due to multicollinearity although they are important in reality. In this case, MDI and SHAP can capture those features.

3) For $O_3$, all models result in similar feature importance rankings. The deterministic forecasts are the dominant features for the models of various forecasting horizons. Also, the lagged maximum concentration, *O3_max_d24*, demonstrates its higher importance for tree-based models. The high importance of relative humidity (RH) reflects the potential fact that $O_3$ concentrations may be higher during dry, clear sky conditions, not completely captured by the deterministic forecasts.

## 4.2    Street Canyon sites

### 4.2.1    Comparison between deterministic forecasts and ML models - street canyon sites

For all street sites, the forecasts of $NO_x$ are improved by the ML models, which are illustrated in detail for different pollutants in Figure 8 and Table 6. The improvements in terms of MQI, R-squared ($R^2$), Pearson correlation, MAPE, nRMSE, nMAE, and MSE show similar patterns for the ML models but differ between street sites.

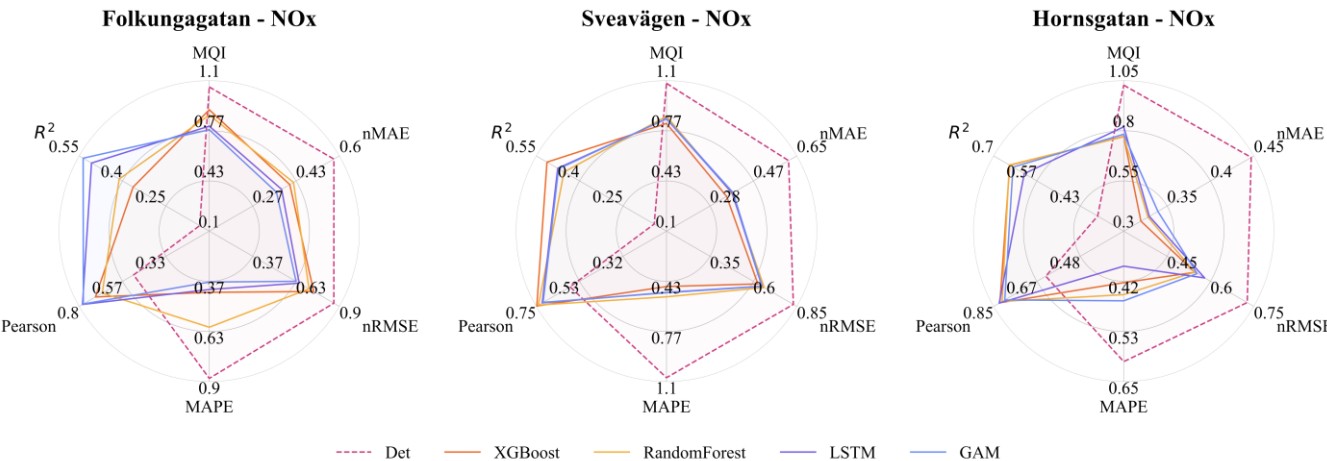

**Figure 8. Statistical performances for ML models and the deterministic hourly forecasts of $NO_X$ for the street site. Mean of 1-day, 2-day and 3-day forecasts. Note that the ranges are different for different metrics.**

Figure 8 summarises the improvements, in terms of different statistical performance metrics, for $NO_x$ prediction at all street canyon sites and for different ML models. The error, represented by MAPE, nRMSE, nMAE and MSE, is reduced by 30% to 60%, and the R-squared coefficients are increased by 30% to 50%. Similar to Urban Background, the variation of Pearson correlation is similar to that of $R^2$, but Pearson correlation tends to be much larger than $R^2$ for the same model. Also, relative uncertainties decrease using the ML models compared to the deterministic forecast.

It should be noted that the $R^2$ of some deterministic forecasts is negative in Table 6, which implies that the deterministic forecasts are sometimes worse than simply using the mean of pollutant concentration as the predictor. For Folkungagatan, the GAM model shows a good integration of results from the tree-based model and LSTM, resulting in further improvement on the prediction performance. MSE of the XGBoost model drops by more than 40% in Sveavägen. Forecasts for Hornsgatan show higher $R^2$ and lower relative errors compared to the other streets. In addition, LSTM models exhibit greater variability compared to the tree model due to its training process being more susceptible to random influences.

**Table 6. Comparison of 1-, 2-, 3-day deterministic and ML forecasts for NO$_x$ for the street canyon sites. All data are based on hourly mean values. The average performances with their 95% confidence interval were computed on the test set from 10 experimental repetitions conducted with different random seeds, and the best performances are bold.**

| Folkungagatan FO | | | | | | | | | | | | | | | |
|---|---|---|---|---|---|---|---|---|---|---|---|---|---|---|---|
| | $R^2$ | | | MAPE | | | nRMSE | | | nMAE | | | MSE | | |
| | 1-day | 2-day | 3-day | 1-day | 2-day | 3-day | 1-day | 2-day | 3-day | 1-day | 2-day | 3-day | 1-day | 2-day | 3-day |
| Det | -0.08 | 0.02 | 0.09 | 0.84 | 0.96 | 0.96 | 0.97 | 0.92 | 0.89 | 0.62 | 0.61 | 0.60 | 1337.64 | 1209.23 | 1125.66 |
| XGB | 0.35 ± 0.00 | 0.34 ± 0.01 | 0.36 ± 0.01 | 0.43 ± 0.00 | 0.44 ± 0.00 | 0.44 ± 0.01 | 0.75 ± 0.00 | 0.75 ± 0.00 | 0.74 ± 0.00 | 0.41 ± 0.00 | 0.42 ± 0.00 | 0.41 ± 0.00 | 799.15 ± 6.08 | 813.34 ± 8.94 | 791.93 ± 8.65 |
| RF | 0.41 ± 0.00 | 0.40 ± 0.00 | 0.40 ± 0.00 | 0.59 ± 0.00 | 0.63 ± 0.00 | 0.64 ± 0.00 | 0.72 ± 0.00 | 0.72 ± 0.00 | 0.72 ± 0.00 | 0.42 ± 0.00 | 0.43 ± 0.00 | 0.43 ± 0.00 | 733.81 ± 2.8 | 745.93 ± 4.92 | 741.81 ± 3.91 |
| LSTM | 0.46 ± 0.02 | 0.45 ± 0.03 | 0.49 ± 0.03 | 0.45 ± 0.02 | 0.44 ± 0.02 | 0.48 ± 0.05 | 0.68 ± 0.01 | 0.69 ± 0.02 | 0.67 ± 0.02 | 0.40 ± 0.01 | 0.40 ± 0.01 | 0.40 ± 0.01 | 663.36 ± 19.42 | 680.08 ± 34.83 | 636.66 ± 39.38 |
| GAM | **0.51 ± 0.01** | **0.49 ± 0.02** | **0.53 ± 0.02** | **0.38 ± 0.01** | **0.40 ± 0.01** | **0.40 ± 0.01** | **0.65 ± 0.01** | **0.66 ± 0.01** | **0.64 ± 0.01** | **0.37 ± 0.00** | **0.38 ± 0.01** | **0.37 ± 0.01** | **604.22 ± 15.79** | **633.22 ± 24.17** | **585.36 ± 22.99** |
| Sveavägen SV | | | | | | | | | | | | | | | |
| | $R^2$ | | | MAPE | | | nRMSE | | | nMAE | | | MSE | | |
| | 1-day | 2-day | 3-day | 1-day | 2-day | 3-day | 1-day | 2-day | 3-day | 1-day | 2-day | 3-day | 1-day | 2-day | 3-day |
| Det | -0.04 | 0.02 | 0.03 | 1.11 | 1.18 | 1.00 | 0.92 | 0.89 | 0.88 | 0.66 | 0.65 | 0.62 | 1620.28 | 1525.52 | 1507.64 |
| XGB | **0.49 ± 0.01** | **0.50 ± 0.01** | **0.48 ± 0.00** | **0.50 ± 0.01** | **0.51 ± 0.01** | **0.47 ± 0.01** | **0.64 ± 0.00** | **0.64 ± 0.01** | **0.64 ± 0.00** | **0.37 ± 0.00** | **0.37 ± 0.00** | **0.36 ± 0.00** | **787.94 ± 9.25** | **786.33 ± 17.12** | **804.19 ± 7.64** |
| RF | 0.46 ± 0.00 | 0.45 ± 0.00 | 0.43 ± 0.00 | 0.54 ± 0.00 | 0.55 ± 0.00 | 0.54 ± 0.00 | 0.66 ± 0.00 | 0.66 ± 0.00 | 0.68 ± 0.00 | 0.38 ± 0.00 | 0.38 ± 0.00 | 0.38 ± 0.00 | 847.22 ± 4.06 | 858.18 ± 3.95 | 892.8 ± 4.55 |
| LSTM | 0.47 ± 0.03 | 0.42 ± 0.06 | 0.35 ± 0.08 | 0.63 ± 0.09 | 0.62 ± 0.09 | 0.56 ± 0.07 | 0.65 ± 0.02 | 0.68 ± 0.03 | 0.72 ± 0.04 | 0.40 ± 0.01 | 0.42 ± 0.02 | 0.44 ± 0.03 | 833.4 ± 53.22 | 897.53 ± 93.33 | 1011.5 ± 119.21 |
| GAM | 0.46 ± 0.02 | 0.44 ± 0.02 | 0.42 ± 0.01 | 0.54 ± 0.01 | 0.54 ± 0.01 | 0.53 ± 0.01 | 0.66 ± 0.01 | 0.67 ± 0.01 | 0.68 ± 0.01 | 0.40 ± 0.01 | 0.40 ± 0.01 | 0.40 ± 0.01 | 836.73 ± 29.81 | 873.7 ± 28.96 | 908.67 ± 23.08 |
| Hornsgatan HO | | | | | | | | | | | | | | | |
| | $R^2$ | | | MAPE | | | nRMSE | | | nMAE | | | MSE | | |
| | 1-day | 2-day | 3-day | 1-day | 2-day | 3-day | 1-day | 2-day | 3-day | 1-day | 2-day | 3-day | 1-day | 2-day | 3-day |
| Det | 0.29 | 0.32 | 0.31 | 0.59 | 0.63 | 0.69 | 0.77 | 0.76 | 0.77 | 0.47 | 0.48 | 0.48 | 2431.10 | 2358.07 | 2387.06 |
| XGB | **0.63 ± 0.00** | **0.63 ± 0.01** | 0.65 ± 0.01 | **0.42 ± 0.01** | **0.44 ± 0.01** | 0.44 ± 0.01 | **0.56 ± 0.00** | **0.56 ± 0.01** | 0.55 ± 0.01 | **0.33 ± 0.00** | **0.33 ± 0.00** | **0.33 ± 0.00** | **1285.99 ± 12.61** | 1273.06 ± 33.59 | 1210.59 ± 30.08 |
| RF | **0.63 ± 0.00** | **0.63 ± 0.00** | **0.66 ± 0.00** | 0.45 ± 0.00 | 0.46 ± 0.00 | 0.46 ± 0.00 | **0.56 ± 0.00** | **0.56 ± 0.00** | **0.54 ± 0.00** | 0.34 ± 0.00 | 0.34 ± 0.00 | **0.33 ± 0.00** | 1288.93 ± 6.34 | **1267.42 ± 5.13** | **1176.98 ± 5.07** |
| LSTM | 0.55 ± 0.04 | 0.57 ± 0.03 | 0.61 ± 0.02 | 0.45 ± 0.09 | **0.46 ± 0.07** | **0.38 ± 0.03** | 0.62 ± 0.03 | 0.60 ± 0.02 | 0.58 ± 0.01 | 0.36 ± 0.02 | 0.36 ± 0.01 | **0.34 ± 0.01** | 1565.54 ± 131.48 | 1483.74 ± 97.31 | 1351.91 ± 61.79 |
| GAM | 0.60 ± 0.00 | 0.61 ± 0.01 | 0.64 ± 0.00 | 0.47 ± 0.00 | 0.48 ± 0.01 | 0.49 ± 0.01 | 0.58 ± 0.00 | 0.57 ± 0.00 | 0.55 ± 0.00 | 0.35 ± 0.00 | 0.35 ± 0.00 | 0.35 ± 0.00 | 1376.94 ± 13.02 | 1339.63 ± 20.34 | 1242.28 ± 11.52 |

Comparison between the statistical performance measures of ML models and deterministic forecasts for PM$_{10}$ gives somewhat diverse results, depending on statistical measure, street site and ML model. MSE decrease slightly in most cases and the normalised RMSE and MAE are lower for most ML models and streets, but not always, while MAPE often increases using the ML models (Table 7 and Figure 9).

$R^2$ and Pearson of LSTM prediction are 10% to 40% higher for Folkungagatan and Hornsgatan. However, the prediction results for Sveavägen show little improvement, and tree-based model and GAM give even worse MAPE than the deterministic forecasts. For relative uncertainties represented by MQI, there is no systematic improvement using ML models compared to the deterministic model.

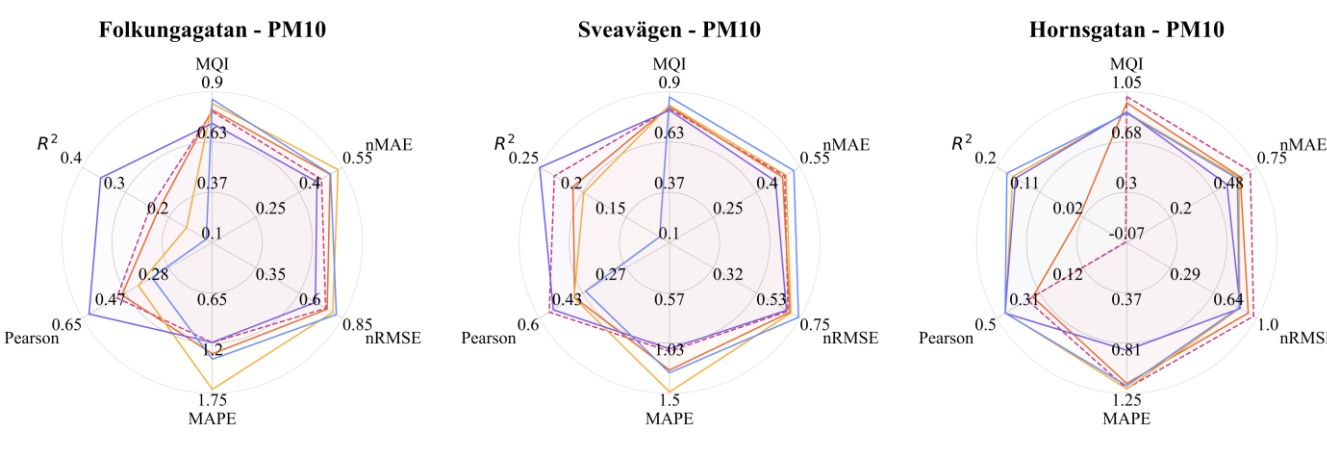

**Figure 9. Statistical performances for ML models versus the deterministic hourly forecasts for PM$_{10}$ at the street canyon sites. Mean of 1-day, 2-day and 3-day forecasts. Note that the ranges are different for different metrics.**

**Table 7. Comparison of 1-, 2-, 3-day deterministic and ML forecasts for $PM_{10}$ for the street canyon sites. The average performances with their 95% confidence interval were computed on the test set from 10 experimental repetitions conducted with different random seeds, and the best performances are bold.**

| Folkungagatan FO | | | | | | | | | | | | | | | |
|---|---|---|---|---|---|---|---|---|---|---|---|---|---|---|---|
| | $R^2$ | | | MAPE | | | nRMSE | | | nMAE | | | MSE | | |
| | 1-day | 2-day | 3-day | 1-day | 2-day | 3-day | 1-day | 2-day | 3-day | 1-day | 2-day | 3-day | 1-day | 2-day | 3-day |
| Det | 0.12 | 0.19 | **0.19** | **1.17** | 1.22 | **1.25** | 0.81 | 0.78 | **0.78** | 0.50 | 0.51 | **0.50** | 83.54 | 77.07 | **76.75** |
| XGB | **0.28** ± **0.01** | 0.15 ± 0.01 | 0.08 ± 0.01 | 1.23 ± 0.02 | 1.37 ± 0.02 | 1.43 ± 0.03 | **0.74** ± **0.00** | 0.80 ± 0.01 | 0.83 ± 0.01 | **0.47** ± **0.00** | 0.54 ± 0.01 | 0.56 ± 0.01 | **69.17** ± **0.94** | 81.23 ± 1.02 | 88.28 ± 1.21 |
| RF | 0.18 ± 0.01 | 0.11 ± 0.01 | 0.04 ± 0.01 | 1.59 ± 0.01 | 1.76 ± 0.02 | 1.79 ± 0.03 | 0.79 ± 0.01 | 0.82 ± 0.01 | 0.85 ± 0.01 | 0.52 ± 0.00 | 0.55 ± 0.01 | 0.57 ± 0.01 | 78.83 ± 1.07 | 84.98 ± 1.43 | 91.70 ± 1.35 |
| LSTM | 0.25 ± 0.22 | **0.26** ± **0.08** | **0.16** ± **0.08** | 1.35 ± 0.51 | **1.16** ± **0.31** | 1.31 ± 0.26 | **0.74** ± **0.09** | **0.74** ± **0.04** | **0.79** ± **0.04** | 0.51 ± 0.1 | **0.50** ± **0.03** | **0.52** ± **0.03** | 71.55 ± 21.24 | **70.96** ± **7.22** | **80.46** ± **8.01** |
| GAM | 0.06 ± 0.04 | 0.01 ± 0.05 | -0.06 ± 0.04 | 1.30 ± 0.07 | 1.40 ± 0.07 | 1.50 ± 0.1 | 0.84 ± 0.02 | 0.86 ± 0.02 | 0.89 ± 0.02 | 0.51 ± 0.01 | 0.53 ± 0.01 | 0.56 ± 0.01 | 90.24 ± 4.01 | 94.42 ± 4.35 | 101.27 ± 4.28 |
| Sveavägen SV | | | | | | | | | | | | | | | |
| | $R^2$ | | | MAPE | | | nRMSE | | | nMAE | | | MSE | | |
| | 1-day | 2-day | 3-day | 1-day | 2-day | 3-day | 1-day | 2-day | 3-day | 1-day | 2-day | 3-day | 1-day | 2-day | 3-day |
| Det | 0.01 | 0.08 | **0.16** | **1.04** | **1.23** | 1.14 | 0.78 | 0.75 | **0.72** | 0.53 | 0.54 | 0.51 | 120.08 | 111.57 | **102.73** |
| XGB | **0.25** ± **0.01** | 0.12 ± 0.02 | 0.15 ± 0.01 | 1.19 ± 0.01 | 1.33 ± 0.02 | 1.36 ± 0.01 | **0.68** ± **0.00** | 0.74 ± 0.01 | 0.73 ± 0.00 | **0.47** ± **0.00** | 0.52 ± 0.01 | 0.51 ± 0.00 | **91.71** ± **1.10** | 107.34 ± 1.85 | 103.44 ± 1.13 |
| RF | 0.21 ± 0.00 | 0.14 ± 0.01 | 0.15 ± 0.01 | 1.40 ± 0.01 | 1.54 ± 0.01 | 1.53 ± 0.01 | 0.70 ± 0.00 | 0.73 ± 0.00 | 0.73 ± 0.00 | 0.48 ± 0.00 | 0.53 ± 0.00 | 0.52 ± 0.00 | 96.62 ± 0.52 | 105.28 ± 0.93 | 103.95 ± 0.69 |
| LSTM | 0.22 ± 0.06 | **0.21** ± **0.05** | **0.11** ± **0.07** | **1.08** ± **0.11** | 1.24 ± 0.19 | **1.03** ± **0.16** | 0.70 ± 0.03 | **0.70** ± **0.02** | 0.74 ± 0.03 | 0.48 ± 0.01 | **0.49** ± **0.02** | 0.50 ± 0.01 | 95.35 ± 7.25 | **96.84** ± **6.32** | 108.79 ± 9.04 |
| GAM | 0.15 ± 0.03 | -0.08 ± 0.04 | 0.02 ± 0.03 | 1.23 ± 0.03 | 1.34 ± 0.05 | 1.41 ± 0.02 | 0.73 ± 0.01 | 0.82 ± 0.02 | 0.78 ± 0.01 | 0.51 ± 0.01 | 0.57 ± 0.01 | 0.56 ± 0.01 | 104.08 ± 3.79 | 131.86 ± 5.05 | 119.97 ± 3.9 |
| Hornsgatan HO | | | | | | | | | | | | | | | |
| | $R^2$ | | | MAPE | | | nRMSE | | | nMAE | | | MSE | | |
| | 1-day | 2-day | 3-day | 1-day | 2-day | 3-day | 1-day | 2-day | 3-day | 1-day | 2-day | 3-day | 1-day | 2-day | 3-day |
| Det | -0.00 | -0.21 | -0.36 | 1.09 | 1.19 | 1.39 | 0.94 | 1.03 | 1.09 | 0.67 | 0.71 | 0.81 | 118.50 | 143.06 | 160.01 |
| XGB | 0.11 ± 0.05 | -0.27 ± 0.12 | -0.09 ± 0.05 | 1.05 ± 0.05 | 1.27 ± 0.06 | **1.21** ± **0.02** | 0.89 ± 0.03 | 1.06 ± 0.05 | 0.98 ± 0.02 | 0.61 ± 0.02 | 0.71 ± 0.03 | 0.67 ± 0.01 | 105.38 ± 6.19 | 151.11 ± 13.92 | 129.35 ± 5.95 |
| RF | **0.18** ± **0.01** | **0.07** ± **0.0** | -0.02 ± 0.01 | 1.06 ± 0.01 | 1.28 ± 0.01 | 1.33 ± 0.01 | **0.85** ± **0.00** | **0.91** ± **0.00** | 0.95 ± 0.00 | 0.60 ± 0.00 | 0.67 ± 0.00 | 0.70 ± 0.00 | **97.74** ± **0.84** | 109.93 ± 0.49 | 121.18 ± 1.15 |
| LSTM | 0.16 ± 0.06 | 0.06 ± 0.09 | **0.05** ± **0.05** | **0.86** ± **0.16** | **0.99** ± **0.18** | **0.89** ± **0.12** | 0.86 ± 0.03 | **0.91** ± **0.04** | **0.92** ± **0.03** | **0.56** ± **0.03** | **0.62** ± **0.03** | **0.59** ± **0.02** | 99.32 ± 7.14 | **111.77** ± **10.36** | **112.97** ± **6.34** |
| GAM | 0.15 ± 0.01 | 0.07 ± 0.01 | -0.02 ± 0.02 | 1.04 ± 0.01 | 1.24 ± 0.02 | 1.32 ± 0.01 | 0.87 ± 0.01 | 0.91 ± 0.00 | 0.95 ± 0.01 | 0.60 ± 0.00 | 0.66 ± 0.00 | 0.69 ± 0.01 | 100.36 ± 1.65 | **109.90** ± **0.89** | 120.50 ± 1.92 |

5    Comparisons between the hourly temporal variations in observations and forecasts of $NO_x$ with the GAM model in October 2022 are shown in Figure 10. Further details for all models are presented in Appendix G. One can see that the deterministic forecast tends to overestimate concentrations of $NO_x$ during daytime especially for Sveavägen, and this is corrected when ML model is being applied. Corresponding plots for $PM_{10}$ are shown in appndix F. In this case, the GAM overestimates concentrations on Hornsgatan during the beginning of October, but performs well otherwise.

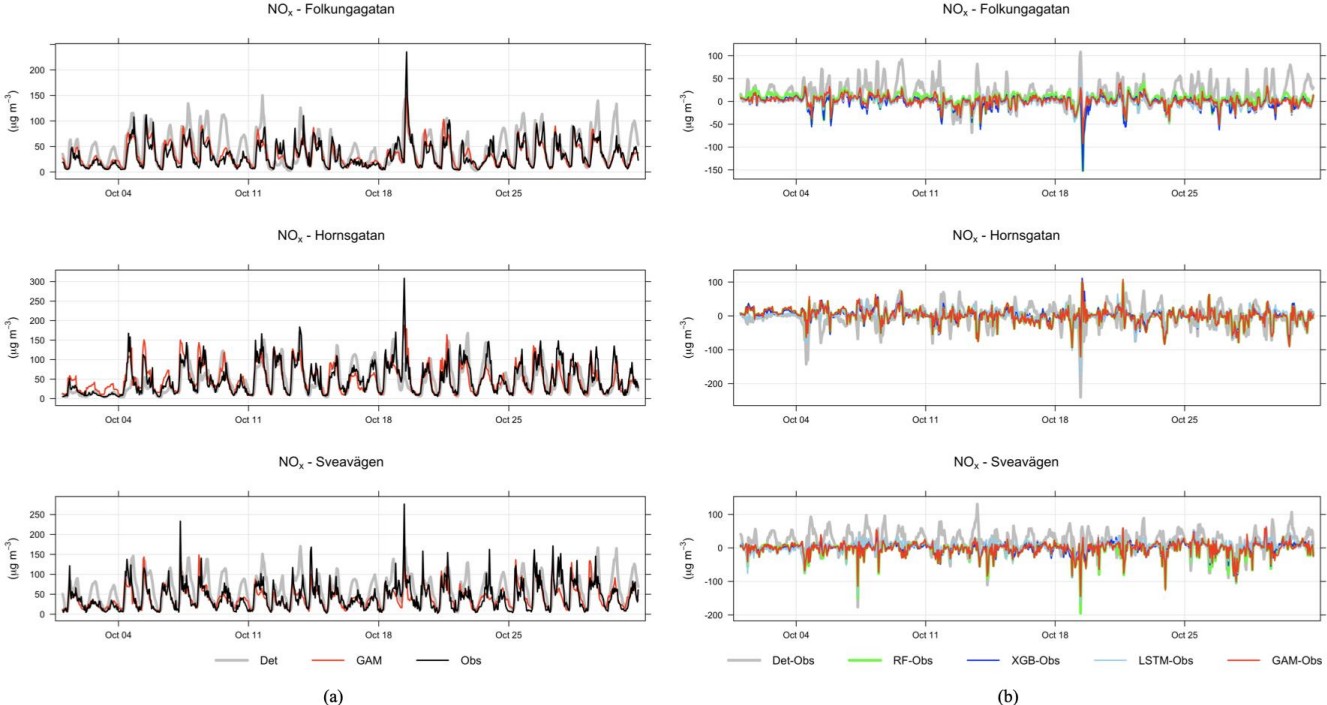

(a)                                                            (b)

**Figure 10. (a) Temporal variations in hourly mean NO$_x$ concentrations at the street canyon site during October 2021 based on mean of 1-, 2- and 3-day forecasts for observations(black), deterministic forecasts(gray) and GAM(red). (b) Absolute deviations of forecasted NO$_x$ concentrations from observed (Obs) concentrations at the street canyon site based on mean of 1-, 2- and 3-day forecasts for October 2021.**

The improvement of the temporal variations of NO$_x$ and PM$_{10}$ is illustrated by comparing the mean diurnal variations in observations with deterministic model and other models in Appendix G. For all street sites, the deterministic forecasts of both NO$_x$ and PM$_{10}$ concentrations show systematic deviations from observations, which are corrected by applying the ML models, especially for NO$_x$. The tendency that the GAM model is not as good at capturing variations in PM$_{10}$ at the urban site is also seen here for the street canyon sites.

As pointed out before it is important to assess statistical performance measures for periods with high concentrations. Similar to what is shwon for the urban site, the statistical performance indexes for all models are much worse for the hourly average concentrations that are higher than the mean values, and the pattern is also similar for the almost street sites, shown in Appendix H. However, the performance of ML models for NO$_X$ maintains the improvement in Hornsgatan, as detailed in Figure 12, suggesting that the model effectively captures the significant variations in high concentration levels.

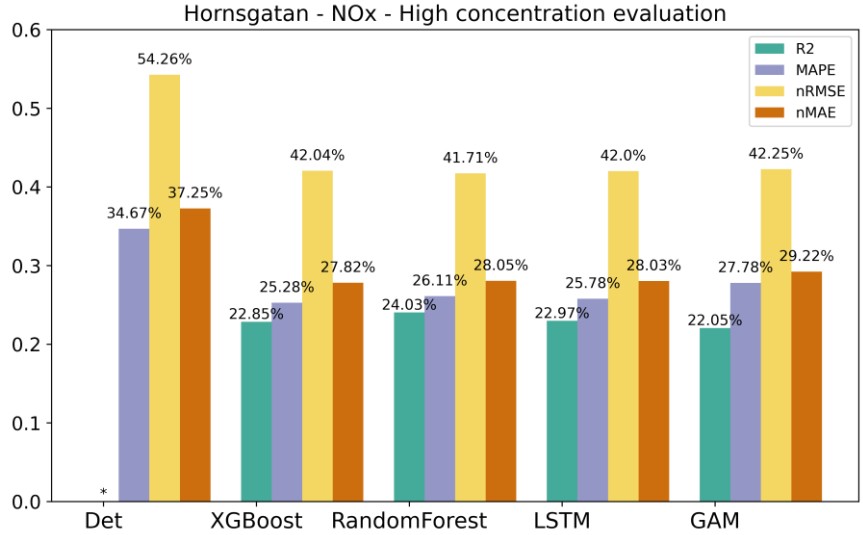

**Figure 11. Statistical performance measures for forecasted NO$_x$ hourly mean concentrations higher than the mean values at Hornsgatan, where * represents a negative R$^2$ value. Mean of 1-, 2- and 3-day forecasts.**

### 4.2.2 Importance of features - street canyon sites

For the street canyon sites, the feature importance rankings are different for PM$_{10}$ and NO$_x$, and also depend on ML models and street sites. Detailed rankings are presented in the figures in Appendix I. There are, however, some typical features that tend to be more important. For PM$_{10}$, Julian day, lagged measurements and deterministic forecasts are, in most cases, among the top 5 most important features for RF and XGB models, whereas precipitation is an important feature for LSTM models. For NO$_x$, deterministic forecasts, hour and weekday are among the most important features, while the features of lagged measurements seem less useful for the ML models. The importance ranking of calendar features of NO$_x$ models indicates the importance of diurnal and weekday variations of traffic emissions not properly captured by the deterministic forecast. The importance of Julian Day reflects the seasonal variation of non-exhaust emission PM$_{10}$, and The importance of precipitation reflects the impacts of street wetness on the suspension of road dust. Even though there are variations, it is difficult to summarise any systematic difference in the features between ML models for the different street sites.

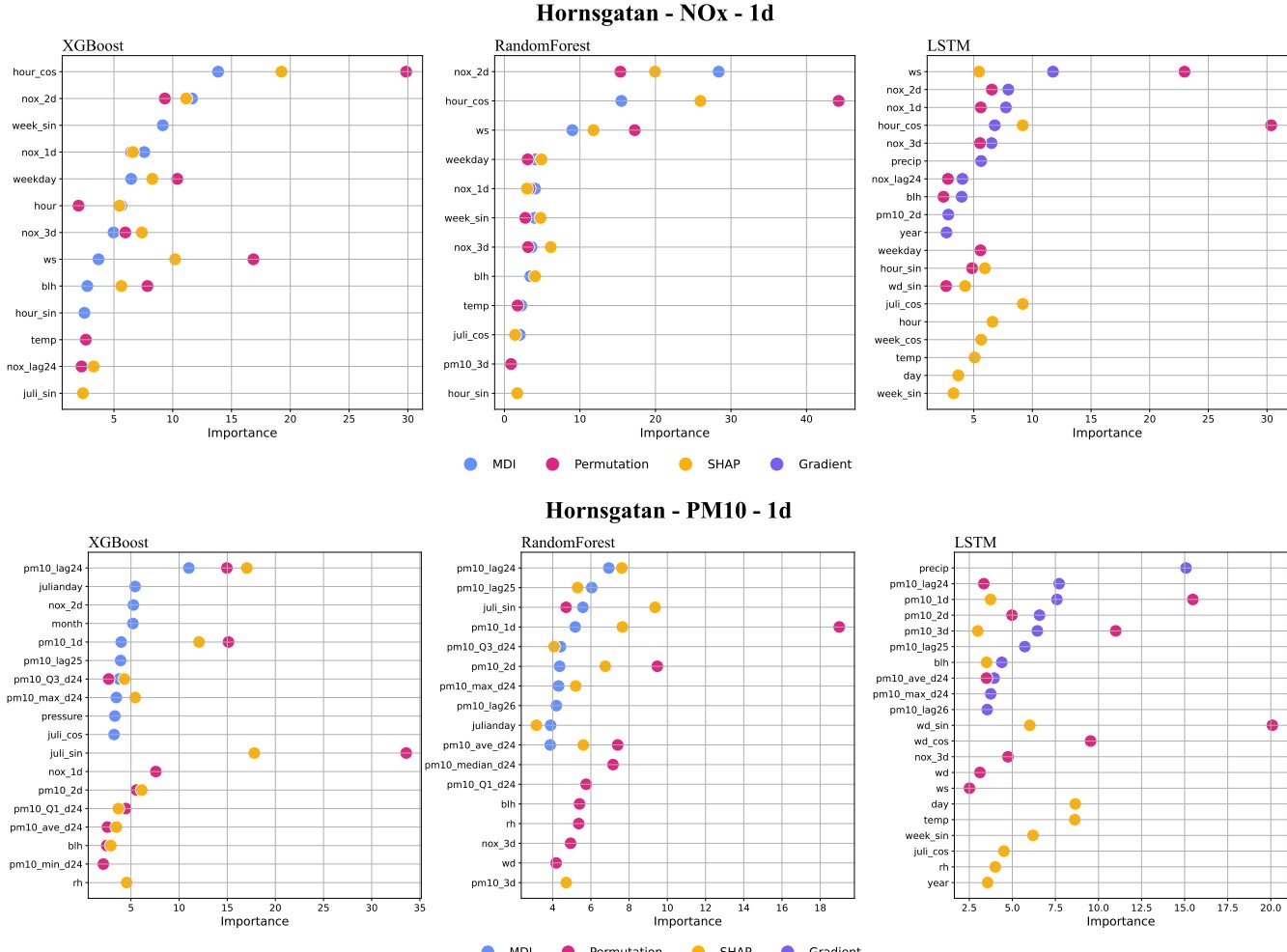

**Figure 12. Top 10 important features (%) for 1-day forecasts using XGB, RF and LSTM at Hornsgatan. All data are hourly mean concentrations.**

## 4.3    Generalisation of street canyon modelling

Until now, the model performance has been evaluated using training and testing data from three single sites respectively. In Stockholm as well as in other cities most of the streets do not have any monitoring station. This is of course due to resource constraints but also associated with the fact that the EU Air Quality Directives regulates the number of monitoring sites required in a city depending on the level of air pollution and number of inhabitants. The monitoring stations should provide information for both areas where the highest concentrations of air pollutants occur and other areas that are representative of the exposure of the general population. Fewer resources are required if this information can be achieved by accurate enough modelling.

We therefore analyze the generalization capacities of the models, with the expectation that we can achieve certain prediction performances of one site without having any measurement data. Computational experiments were carried out through cross-

validation, which combines training and testing data coming from different measurement sites. For the street canyon sites, four combinations of training datasets were applied to evaluate the generalization abilities of different ML models.

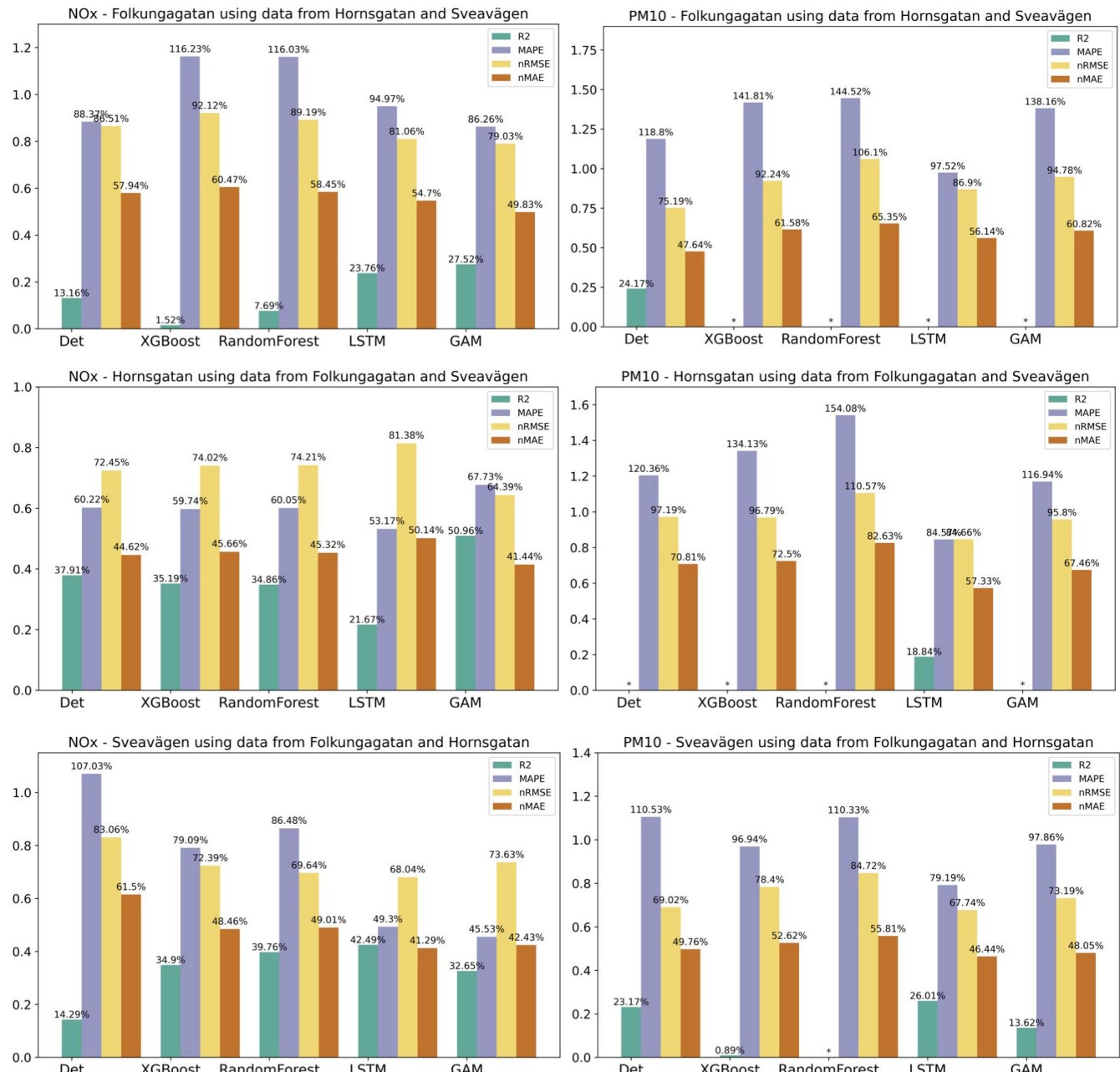

**Figure 13. Statistical performances of NO$_x$ and PM$_{10}$ forecasts for the streets on test set when the ML models are trained using only data from the other streets. Mean of 1-day, 2-day, and 3-day forecasts.**

Figure 13 shows the mean of 1-day, 2-day, and 3-day forecasted NO$_x$ and PM$_{10}$ concentrations on the test set for the three street canyon sites based on training the models on the other streets. It shows that the forecast is improved compared to the

deterministic forecast for Hornsgatan and Sveavägen, but not so much for Folkungagatan. For Sveavägen the $R^2$ is 0.14 using the deterministic forecast whereas the ML models give $R^2$ between 0.62 and 0.63 and here all errors decrease substantially using the ML models. But for Folkungagatan the ML models show different results. $R^2$ is similar or even decreases for tree-based models, whereas errors mostly decrease depending on the ML applied.

The performance of $PM_{10}$ is shown in the right part of Figure 13. It can be seen that it is not possible to find any major improvement in the deterministic forecast for the streets using RF and XGB. But with LSTM $R^2$ increases slightly and errors decrease for Hornsgatan and Sveavägen compared to the deterministic forecasts.

## 5    Discussion

The performance of the ML models is quite similar for the different sites and forecast days. However, there are large differences
in improvements for different pollutants. In general, our results indicate that ML models are more effective in improving $NO_x$ than $PM_{10}$. For $PM_{10}$ the ML models show slight improvement in $R^2$ but not much improvements in relative errors. This difference in improvement is likely associated with the different processes controlling the concentrations, such as different sources: $NO_x$ concentrations being mainly due to vehicle exhaust emissions which show regular variations from one day to the next depending on day of the week and time of day, while $PM_{10}$ is mainly due to road dust emissions controlled by a
combination of variations in vehicle volumes and meteorological conditions that affect suspension of coarse particles from street surfaces (e g Denby et al., 2013a; Johansson et al., 2007; Krecl et al., 2021). Road dust accumulates on the road surfaces during wet road surface conditions and is suspended by vehicle-induced turbulence during dry conditions (Denby et al., 2013a). The improvement of the forecasts of $NO_x$ with ML is partly driven by the calendar, hour, day of the week and to some degree also Julian day, but different features appear as important for RF compared to XGB. For $PM_{10}$ the seasonal variation described
by Julian Day is the most important feature at the street canyon sites, for both RF and XGB. This indicates that the deterministic forecasts are not capable of describing the impacts of meteorology and road dust emissions on $PM_{10}$, even though parameterisations of these processes are included in the deterministic modelling system. The total mass generated by road wear is a key factor for $PM_{10}$ emissions and these emissions are strongly controlled by surface moisture conditions, and this is taken into account by the NORTRIP model. But as pointed out by Denby et al (2013b) there are periods where surface wetness is
not well modelled and it is not known if this is the result of input data, e.g. precipitation, or of the model formulation itself.
It is clear that the deterministic forecast of $O_3$ underestimates concentrations at the urban site due to the fact that the local emissions of $NO_x$ influencing the photochemistry are not properly considered by the CAMS model, but this is corrected using the ML models. Despite the deterministic forecast is the most important feature for both RF and XGB, lagged measured mean and maximum $O_3$ concentrations improve the deterministic forecasts.
Although the fact that the configurations and traffic situations are quite similar for the street canyon sites, the improvements of the deterministic forecasts over ML models differ. For $NO_x$, the forecasts on Hornsgatan are more accurate (lower errors and higher $R^2$) than for the other two sites, while for $PM_{10}$ there is no obvious difference between the sites.

The overall model quality according to the recommendations by the Forum for Air Quality Modeling in the context of the air quality directives, is improved using the ML models resulting in uncertainties that are significantly smaller than the measurement uncertainties for all pollutants. However, the forecasts of the highest concentrations including episodes with high concentrations, are not systematically improved for all pollutants and all performance measures using the ML models.

We have shown that the statistical performances of the deterministic forecasts for concentrations of $NO_x$ at the street canyon sites can be improved using the ML models. However, for $PM_{10}$, LSTM showed systematic improvements at all sites. So again this accentuates the importance of testing the models not only for one pollutant. Further work is needed to improve deterministic forecasts of $PM_{10}$ based on the training of ML models at a few monitoring stations. As discussed above the situation in Stockholm is different from cities in central and southern Europe since the road dust contribution is very large. It

might be that results for $PM_{10}$ are different in other cities, but we have not found any publication on this matter.

## 5.1    Comparison of different ML models

Several studies have compared performance of different machine learners in predicting air quality (Zaini et al., 2021). Assessing forecasts of $PM_{10}$ and $PM_{2.5}$ concentrations, Czernecki et al. (2021) found that XGB performed the best, followed by RF and an artificial neural network model, while stepwise regression performed the worst in four Polish agglomerations.

Likewise, Joharestani et al. (2019) found XGB to perform best of three ML models (XGB, RF, and a deep learning algorithm), in predicting $PM_{2.5}$ in Tehran (Iran). On the contrary, LSTM was shown to outperform XGBoost for forecasting hourly $PM_{2.5}$ concentrations (Qadeer et al., 2020), similar to what was shown by Chuluunsaikhan et al. (2021). Cai et al. (2009) obtained more accurate predictions of CO concentrations using artificial neural network modelling compared to using multiple linear regression and the deterministic California line source dispersion model. On the other hand, Shaban et al. (2015) concluded

that a tree-based algorithm (M5P) outperformed artificial neural network modelling when comparing forecasts of different pollutants in Qatar. There are many reasons for the different results presented in the literature, including model formulation and setup, different types of input data, different atmospheric conditions and source contributions governing the concentrations. Also, different performance metrics have been used. This makes it hard to draw general conclusions regarding which model to use. However, we find that other factors may be more important to consider than the type of model – such as sources of

pollutants and influence of photochemistry, characteristics of the site resulting in different features being of varying importance depending on pollutant type of location. In this context output of feature importance methods can provide useful information to improve models.

Another more practical aspect to consider when comparing the ML models is the complexity and computer resources required for training the models. In AQ literature, deep learning models such as standard LSTM and other Recurrent Neural Networks

(RNNs) have been explored for their prediction capacities. However, most of the studies have adopted complex neural network structures, such as models of multiple outputs that mainly give convenience for data processing and automated feature handling. Nevertheless, training even a simple LSTM model is computationally much more expensive than the two conventional ML models, i.e. the decision tree-based models (RF and XGB) in our case. In fact, we have to resort to the high-

performance machine, The Swedish Berzelius High-performance Computer, to reduce the computational time. For the current practice in our real air quality prediction system, we implemented the two tree-based models, instead of LSTM. But we are also exploring well-designed deep learning models, which may replace the conventional models being adopted in the AQ system in the near future, especially due to the insights to deploy a generic model and handle all the modelling processes automatically.

## 5.2    Temporal dependency of feature importance

The exploration of feature importance is one contribution of the paper for analysing different ML models. In comparison to MDI and Permutation methods, SHAP provides a more comprehensive approach to analysing feature importance. The model can compute the important value of each feature for all data samples but also estimate the feature importance value for each individual sample. This gives us a useful tool to analyse the temporal dependency of feature importance.

Figure 14 illustrates the feature importance analysis using SHAP method for XGBoost model of 1-day NOx prediction. The left graph illustrates the feature importance ranking derived from test dataset, employing red dots to denote samples with higher numerical values of feature and blue dots to represent lower numerical values. Also, the dots on the left side of the x-axis, i.e., SHAP value < 0, reflect a negative impact for predictions, while right-side dots suggest a positive impact. Figure 14(a) revealed a distinct relationship between the feature *hour_cos* and the $NO_X$ predictions. Higher values of *hour_cos*, representing nighttime, exhibit a negative impact on the forecasts. Conversely, lower values of *hour_cos* show a positive correlation with the forecasts. Additionally, the wider distribution of this feature indicates its significant influence on the prediction process, suggesting that the model may capture the diurnal pattern of traffic emissions. In Figure 14(b), a more pronounced diurnal pattern emerges. Here, SHAP values of feature *hour* are positive from 7:00 a.m. to 17:00 p.m., contrasting with the negative values observed at night. Meanwhile, the high concentration of NOx forecasts *nox_2d* from 2-day deterministic model (red dots) show an evident increase during the heavy traffic period, spanning from 8:00 am to 1:00 am. This observation reinforces the substantial effect of traffic emissions on NOx levels.

Figure 15 displays a heatmap of SHAP values, illustrating the temporal variation of feature importance when they are used by a model to forecast. The deterministic forecast *nox_3d* plays an important role in prediction i.e. executes positive influence (red block) for $NO_X$ predictions with higher numeric value and vice versa. Meanwhile, the weekend, e.g., the 2nd, 9th, 16th, and 23rd of October, exhibit negative impacts (blue block), while the weekday factor provides positive support on model forecasts. The impact of the 24-hour lagged values of the NOx, *nox_lag24,* is also evident. For example, the SHAP value at the peak on Oct 19th has a negative impact. Whereas, the SHAP value of the next day shows a positive impact, which explains the delay between the predicted peak and real observation.

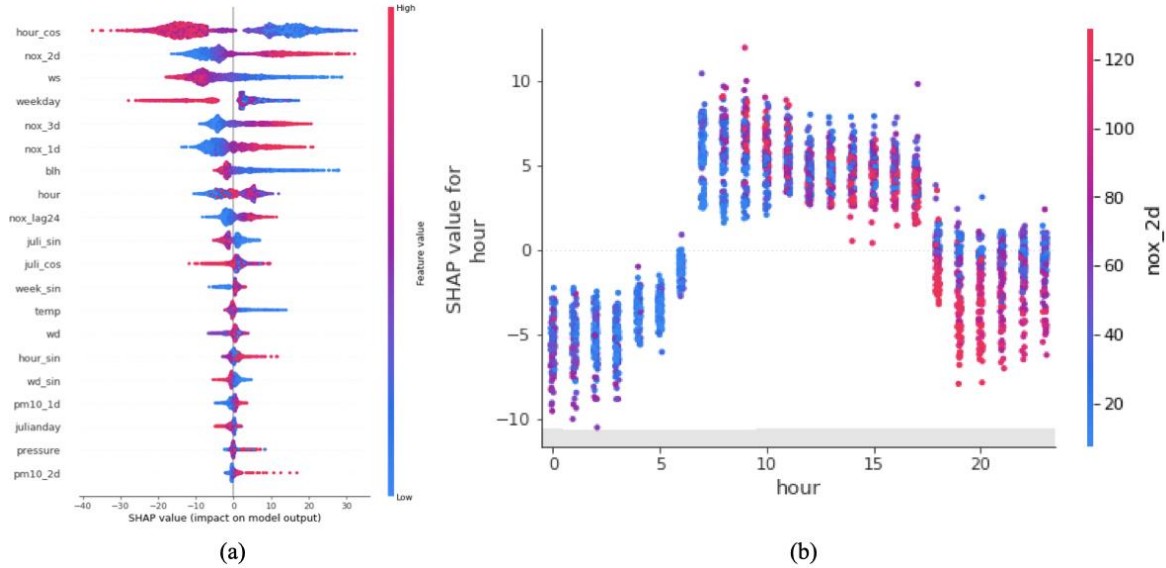

**Figure 14. (a). Feature importance ranking based on SHAP method of XGBoost model for 1-day's NOx prediction on HO site. (b) The relationship between feature *hour* and feature *nox_2d* from the results of SHAP method in (a). All examples belong to test set.**

## NOx- XGBoost - Hornsgatan

5   **Figure 15. SHAP feature importance analysis of XGBoost model for 1-day's prediction of NOx concentrations on HO street. All example is belong to test set. The blue blocks imply a negative impact, while the red blocks are positive.**

## 6    Conclusions

This paper has applied different ML models to improve 1-, 2- and 3-day deterministic forecasts of $NO_x$, $PM_{10}$ and $O_3$ concentrations for multiple locations in Stockholm, Sweden. It is shown that the degree of improvement over deterministic forecasts depends more on pollutant and monitoring site than on what ML algorithm is applied. Also, four feature importance
methods, namely MDI, Permutation, Gradient-based, and SHAP, are utilized to identify significant features that are common and robust across models. Notably, deterministic forecasts of $NO_x$ are significantly improved across all sites, using all models. $R^2$ is increased by up to 80% and prediction errors are reduced by up to 60%. For $PM_{10}$, variable results are achieved, reflecting the more complicated processes controlling the road wear emissions which constitute a large fraction of $PM_{10}$. For $O_3$ at the urban background site, the deviation between deterministically modelled absolute level is corrected by the ML models, and
nRMSE and nMAE are reduced by on average around 20%.

We have shown that it is possible to improve deterministic forecasts of $NO_x$ at street canyon sites, based on training ML models at other sites. When tested for $PM_{10}$, only LSTM shows modest improvements compared to the deterministic forecasts.

One contribution of our study is that we compare forecasts based on several pollutants and base our forecasts on a combination of deterministic models, which are based on the underlying physicochemical mechanisms responsible for the emissions and
dispersion of the pollutants, and three different ML models with additional variables such as measurement data, calendar data and meteorological data. The models are evaluated at different sites and for different pollutants during several months with different meteorological conditions. In addition, by comparing the four feature importance methods, the robust features for associated models are identified, establishing the foundation for model performance analysis and improvement.

There are different aspects that we would like to further improve and extend the models. Investigating the impact of the
COVID-19 pandemic on our model's performance is meaningful, especially considering that our dataset predominantly covers this specific time period during the pandemic. Moreover, we will further explore to transfer the learning approach to more general models, addressing the challenges, posed by the scarcity of monitoring stations in many areas, and to represent spatial correlation of the measurement stations.

**Appendix A. Description of measurement methods and sites.**

All measurement methods are approved for monitoring according to the EU air quality directive for $NO_x$, $O_3$ and $PM_{10}$. $PM_{10}$ was measured either using an optical particle counter (Hornsgatan: OPC, Grimm EDM 180-MC) or Tapered Element Oscillating Microbalance (Sveavägen, Folkungagatan and Urban: TEOM model, 1400AB, Rupprecht & Patashnik, Co). $NO_x$
5   was measured using chemiluminescence (AC32M, Environnement S.A.) and $O_3$ was measured by UV absorption (O342M, Environnement S.A.).

**Table A1. Description of monitoring sites.**

| Site name | Description | Traffic volume | Photo |
|---|---|---|---|
| Hornsgatan | Street canyon site. Measurements of $NO_x$ and $PM_{10}$ on north side of street, 3 m above ground. Street width 24 m and building height 24 m. | 23 000 veh/day (4% heavy duty vehicles). Vehicle composition measured during 4 week campaigns using automatic number plate recognition. |  |
| Sveavägen | Street canyon site. Measurements of $NO_x$, $PM_{10}$ on west side of street, 3 m above ground. Street width 33 m and building height 24 m. | 21 000 veh/day (7% heavy duty vehicles). |  |

| | | | |
|---|---|---|---|
| Folkungagatan | Street canyon site. Measurements $NO_x$, $PM_{10}$ on west side of street, 3 m above ground. Street width 24 m and building height 24 m. | 12 000 veh/day (18% heavy duty vehicles). |  |
| Torkel Knutssongatan | Urban background. Measurements of $NO_x$, $PM_{10}$, ozone and meteorology on top of a 20 m high building. | Ca 13 000 vehicles on Hornsgatan road 250 m N of site. |  |

## Appendix B. Interpolation

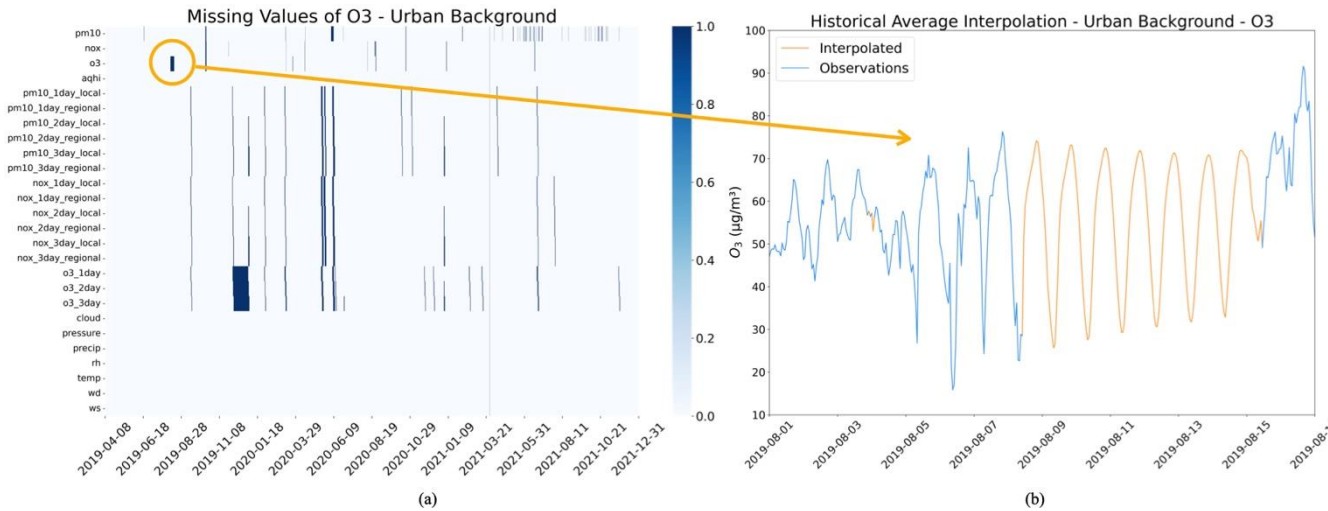

(a)                   (b)

**Figure B1. (a) The missing value of O₃ in the UB dataset, where blue represents missing data and white represents not missing. (b) Interpolation results based on historical averages for O₃ in the UB dataset. The yellow arrows indicate the interpolation results for missing values of O₃ within the yellow circle.**

## Appendix C. Hyperparameter tuning

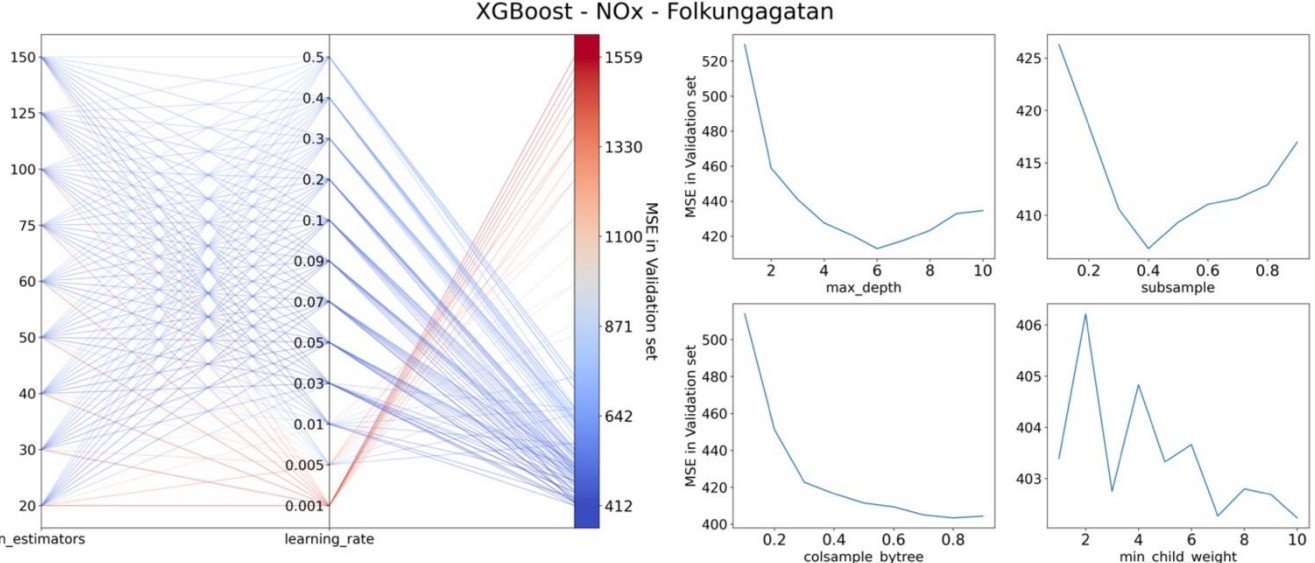

**Figure C1. Illustration of the results of hyperparameter tuning for the XGBoost model of NOx on Folkungagatan.**

5    **Table C1. The result of hyperparameter tuning for all models and all sites.**

| Station | Pollutants | Models | Range of Hyperparameters | Best parameters |
|---------|------------|--------|--------------------------|-----------------|
| FO | NOX | XGBoost | 'n_estimators': [20,30,40,50,60,75,100,125,150],<br>'learning_rate': [0.005,0.01,0.03,0.05,0.07,0.09,0.1, 0.2, 0.3]<br>"max_depth": [1, 2, 3, 4, 5, 6, 7, 8, 9, 10],<br>"subsample": [0.1, 0.2, 0.3, 0.4, 0.5, 0.6, 0.7, 0.8, 0.9],<br>"colsample_bytree": [0.1, 0.2, 0.3, 0.4, 0.5, 0.6, 0.7, 0.8, 0.9],<br>"min_child_weight": [1, 2, 3, 4, 5, 6, 7, 8, 9, 10]. | 'n_estimators': 60,<br>'max_depth': 6,<br>'min_child_weight': 10,<br>'colsample_bytree': 0.8,<br>'learning_rate': 0.03,<br>'subsample': 0.4. |
| FO | NOX | RandomForest | 'n_estimators': [50,100,150,200,250,300,325,350,375,400],<br>'max_features': [None, 'sqrt', 'log2'],<br>'max_depth': [None,1,2,3,4,5,6,7,8,9,10],<br>'min_samples_split': [1,2,3,4,5,6,7,8,9,10],<br>'min_samples_leaf': [1,2,3,4,5,6,7,8,9,10]. | 'max_features': 'sqrt',<br>'n_estimators': 250,<br>'max_depth': 7,<br>'min_samples_split': 10,<br>'min_samples_leaf': 9 |
| FO | NOX | LSTM | 'batch_size': [24,48,72,96,120,144,168],<br>'n_steps_in': [12,24,36,48,60],<br>'hidden_size': [32,64,96,128,160],<br>'learning_rate': [1e-2,5e-2,1e-3,5e-3,1e-4]. | 'batch_size': 168,<br>'n_steps_in': 48,<br>'hidden_size': 160,<br>'learning_rate': 0.001. |
| FO | PM10 | XGBoost | 'n_estimators': [20,30,40,50,60,75,100,125,150],<br>'learning_rate': [0.005,0.01,0.03,0.05,0.07,0.09,0.1, 0.2, 0.3]<br>"max_depth": [1, 2, 3, 4, 5, 6, 7, 8, 9, 10],<br>"subsample": [0.1, 0.2, 0.3, 0.4, 0.5, 0.6, 0.7, 0.8, 0.9],<br>"colsample_bytree": [0.1, 0.2, 0.3, 0.4, 0.5, 0.6, 0.7, 0.8, 0.9],<br>"min_child_weight": [1, 2, 3, 4, 5, 6, 7, 8, 9, 10]. | 'learning_rate': 0.06,<br>'n_estimators': 300,<br>'max_depth': 2,<br>'subsample': 0.5,<br>'colsample_bytree': 0.3,<br>'min_child_weight': 9. |
| FO | PM10 | RandomForest | 'n_estimators': [50,100,150,200,300,400,425,450,475,500,550],<br>'max_features': [None, 'sqrt', 'log2'],<br>'max_depth': [None,1,2,3,4,5,6,7,8,9,10],<br>'min_samples_split': [1,2,3,4,5,6,7,8,9,10],<br>'min_samples_leaf': [1,2,3,4,5,6,7,8,9,10]. | 'max_features': None,<br>'n_estimators': 475,<br>'max_depth': None,<br>'min_samples_split': 1,<br>'min_samples_leaf': 1. |

| Station | Pollutants | Models | Range of Hyperparameters | Best parameters |
|---------|-----------|--------|--------------------------|-----------------|
| FO | $PM_{10}$ | LSTM | 'batch_size': [24,48,72,96,120,144,168],<br>'n_steps_in': [12,24,36,48,60],<br>'hidden_size': [32,64,96,128,160],<br>'learning_rate': [1e-2,5e-2,1e-3,5e-3,1e-4]. | 'batch_size': 168,<br>'n_steps_in': 60,<br>'hidden_size': 128,<br>'learning_rate': 0.001. |
| HO | $NO_X$ | XGBoost | 'n_estimators': [20,30,40,50,60,75,100,125,150],<br>'learning_rate': [0.08,0.085,0.09,0.095,0.1, 0.2, 0.3, 0.4, 0.5],<br>"max_depth": [1, 2, 3, 4, 5, 6, 7, 8, 9, 10],<br>"subsample": [0.1, 0.2, 0.3, 0.4, 0.5, 0.6, 0.7, 0.8, 0.9],<br>"colsample_bytree": [0.1, 0.2, 0.3, 0.4, 0.5, 0.6, 0.7, 0.8, 0.9],<br>"min_child_weight": [1, 2, 3, 4, 5, 6, 7, 8, 9, 10]. | 'learning_rate': 0.095,<br>'n_estimators': 40,<br>'max_depth': 6,<br>'subsample': 0.8,<br>'colsample_bytree': 0.7,<br>'min_child_weight': 6. |
| HO | $NO_X$ | RandomForest | 'n_estimators': [50,100,150,200,250,300,325,350,375,400],<br>'max_features': [None, 'sqrt', 'log2'],<br>'max_depth': [None,1,2,3,4,5,6,7,8,9,10],<br>'min_samples_split': [1,2,3,4,5,6,7,8,9,10],<br>'min_samples_leaf': [1,2,3,4,5,6,7,8,9,10]. | 'max_features': None,<br>'n_estimators': 375,<br>'max_depth': None,<br>'min_samples_split': 1,<br>'min_samples_leaf': 2. |
| HO | $NO_X$ | LSTM | 'batch_size': [24,48,72,96,120,144,168],<br>'n_steps_in': [12,24,36,48,60],<br>'hidden_size': [32,64,96,128,160],<br>'learning_rate': [1e-2,5e-2,1e-3,5e-3,1e-4]. | 'batch_size': 168,<br>'n_steps_in': 60,<br>'hidden_size': 160,<br>'learning_rate':0.005. |
| HO | $PM_{10}$ | XGBoost | 'n_estimators': [20,30,40,50,60,75,100,125,150],<br>'learning_rate': [0.08,0.085,0.09,0.095,0.1, 0.2, 0.3, 0.4, 0.5],<br>"max_depth": [1, 2, 3, 4, 5, 6, 7, 8, 9, 10],<br>"subsample": [0.1, 0.2, 0.3, 0.4, 0.5, 0.6, 0.7, 0.8, 0.9],<br>"colsample_bytree": [0.1, 0.2, 0.3, 0.4, 0.5, 0.6, 0.7, 0.8, 0.9],<br>"min_child_weight": [1, 2, 3, 4, 5, 6, 7, 8, 9, 10]. | 'learning_rate': 0.085,<br>'n_estimators': 30,<br>'max_depth': 4,<br>'subsample': 0.6,<br>'colsample_bytree': 0.8,<br>'min_child_weight': 1. |
| HO | $PM_{10}$ | RandomForest | 'n_estimators': [50,100,150,200,300,400,425,450,475,500,550],<br>'max_features': [None, 'sqrt', 'log2'],<br>'max_depth': [None,1,2,3,4,5,6,7,8,9,10],<br>'min_samples_split': [1,2,3,4,5,6,7,8,9,10],<br>'min_samples_leaf': [1,2,3,4,5,6,7,8,9,10]. | 'max_features': 'sqrt',<br>'n_estimators': 450,<br>'max_depth': None,<br>'min_samples_split': 4,<br>'min_samples_leaf': 1. |
| HO | $PM_{10}$ | LSTM | 'batch_size': [24,48,72,96,120,144,168],<br>'n_steps_in': [12,24,36,48,60],<br>'hidden_size': [32,64,96,128,160],<br>'learning_rate': [1e-2,5e-2,1e-3,5e-3,1e-4]. | 'batch_size': 168,<br>'n_steps_in': 60,<br>'hidden_size': 32,<br>'learning_rate': 0.001. |
| SV | $NO_X$ | XGBoost | 'n_estimators': [20,30,40,50,60,75,100,125,150],<br>'learning_rate': [0.001,0.005,0.01,0.03,0.05,0.07,0.09,0.1, 0.2, 0.3, 0.4, 0.5],<br>"max_depth": [1, 2, 3, 4, 5, 6, 7, 8, 9, 10],<br>"subsample": [0.1, 0.2, 0.3, 0.4, 0.5, 0.6, 0.7, 0.8, 0.9],<br>"colsample_bytree": [0.1, 0.2, 0.3, 0.4, 0.5, 0.6, 0.7, 0.8, 0.9],<br>"min_child_weight": [1, 2, 3, 4, 5, 6, 7, 8, 9, 10]. | 'learning_rate': 0.09,<br>'n_estimators': 60,<br>'max_depth': 6,<br>'subsample': 0.8,<br>'colsample_bytree': 0.6,<br>'min_child_weight': 10. |
| SV | $NO_X$ | RandomForest | 'n_estimators': [50,100,150,200,250,300,325,350,375,400],<br>'max_features': [None, 'sqrt', 'log2'],<br>'max_depth': [None,1,2,3,4,5,6,7,8,9,10],<br>'min_samples_split': [1,2,3,4,5,6,7,8,9,10],<br>'min_samples_leaf': [1,2,3,4,5,6,7,8,9,10]. | 'max_features': 'log2',<br>'n_estimators': 375,<br>'max_depth': None,<br>'min_samples_split': 8,<br>'min_samples_leaf': 5 |
| SV | $NO_X$ | LSTM | 'batch_size': [24,48,72,96,120,144,168],<br>'n_steps_in': [12,24,36,48,60],<br>'hidden_size': [32,64,96,128,160],<br>'learning_rate': [1e-2,5e-2,1e-3,5e-3,1e-4]. | 'batch_size': 168,<br>'n_steps_in': 12,<br>'hidden_size': 64,<br>'learning_rate':0.001. |

| Station | Pollutants | Models | Range of Hyperparameters | Best parameters |
|---|---|---|---|---|
| SV | PM$_{10}$ | XGBoost | 'n_estimators': [30,40,50,100,150,200,250,300,350,400,450,500],<br>'learning_rate': [0.001,0.005,0.01,0.02,0.03,0.04,0.05,0.06,0.07,0.08,0.09,0.1, 0.2, 0.3, 0.4],<br>"max_depth": [1, 2, 3, 4, 5, 6, 7, 8, 9, 10],<br>"subsample": [0.1, 0.2, 0.3, 0.4, 0.5, 0.6, 0.7, 0.8, 0.9],<br>"colsample_bytree": [0.1, 0.2, 0.3, 0.4, 0.5, 0.6, 0.7, 0.8, 0.9],<br>"min_child_weight": [1, 2, 3, 4, 5, 6, 7, 8, 9, 10]. | 'learning_rate': 0.02,<br>'n_estimators': 50,<br>'max_depth': 3,<br>'subsample': 0.2,<br>'colsample_bytree': 0.9,<br>'min_child_weight': 1 |
| SV | PM$_{10}$ | RandomForest | 'n_estimators': [50,100,150,200,300,400,425,450,475,500,550],<br>'max_features': [None, 'sqrt', 'log2'],<br>'max_depth': [None,1,2,3,4,5,6,7,8,9,10],<br>'min_samples_split': [1,2,3,4,5,6,7,8,9,10],<br>'min_samples_leaf': [1,2,3,4,5,6,7,8,9,10]. | 'max_features': 'log2',<br>'n_estimators': 500,<br>'max_depth': 8,<br>'min_samples_split': 3,<br>'min_samples_leaf': 1 |
| SV | PM$_{10}$ | LSTM | 'batch_size': [24,48,72,96,120,144,168],<br>'n_steps_in': [12,24,36,48,60],<br>'hidden_size': [32,64,96,128,160],<br>'learning_rate': [1e-2,5e-2,1e-3,5e-3,1e-4]. | 'batch_size': 168,<br>'n_steps_in': 48,<br>'hidden_size': 96,<br>'learning_rate':0.01. |
| UB | NO$_X$ | XGBoost | 'n_estimators': [20,30,40,50,60,75,100,125,150],<br>'learning_rate': [0.001,0.005,0.01,0.02,0.03,0.04,0.05,0.07,0.09,0.1, 0.2, 0.3, 0.4],<br>"max_depth": [1, 2, 3, 4, 5, 6, 7, 8, 9, 10],<br>"subsample": [0.1, 0.2, 0.3, 0.4, 0.5, 0.6, 0.7, 0.8, 0.9],<br>"colsample_bytree": [0.1, 0.2, 0.3, 0.4, 0.5, 0.6, 0.7, 0.8, 0.9],<br>"min_child_weight": [1, 2, 3, 4, 5, 6, 7, 8, 9, 10]. | 'learning_rate': 0.02,<br>'n_estimators': 150,<br>'max_depth': 6,<br>'subsample': 0.8,<br>'colsample_bytree': 0.6,<br>'min_child_weight': 3. |
| UB | NO$_X$ | RandomForest | 'n_estimators': [50,100,150,200,225,250,275,300,325,350,375,400],<br>'max_features': [None, 'sqrt', 'log2'],<br>'max_depth': [None,1,2,3,4,5,6,7,8,9,10],<br>'min_samples_split': [1,2,3,4,5,6,7,8,9,10],<br>'min_samples_leaf': [1,2,3,4,5,6,7,8,9,10]. | 'max_features': 'sqrt',<br>'n_estimators': 275,<br>'max_depth': 10,<br>'min_samples_split': 1,<br>'min_samples_leaf': 7. |
| UB | NO$_X$ | LSTM | 'batch_size': [24,48,72,96,120,144,168],<br>'n_steps_in': [12,24,36,48,60],<br>'hidden_size': [32,64,96,128,160],<br>'learning_rate': [1e-2,5e-2,1e-3,5e-3,1e-4]. | 'batch_size': 168,<br>'n_steps_in': 60,<br>'hidden_size': 160,<br>'learning_rate': 0.001. |
| UB | PM$_{10}$ | XGBoost | 'n_estimators': [50,75,100,200,300,400,500,600],<br>'learning_rate': [0.01,0.03,0.04,0.05,0.06,0.07,0.09,0.1, 0.2, 0.3, 0.4],<br>"max_depth": [1, 2, 3, 4, 5, 6, 7, 8, 9, 10],<br>"subsample": [0.1, 0.2, 0.3, 0.4, 0.5, 0.6, 0.7, 0.8, 0.9],<br>"colsample_bytree": [0.1, 0.2, 0.3, 0.4, 0.5, 0.6, 0.7, 0.8, 0.9],<br>"min_child_weight": [1, 2, 3, 4, 5, 6, 7, 8, 9, 10]. | 'learning_rate': 0.04,<br>'n_estimators': 600,<br>'max_depth': 6,<br>'subsample': 0.4,<br>'colsample_bytree': 0.8,<br>'min_child_weight': 1. |
| UB | PM$_{10}$ | RandomForest | 'n_estimators': [50,100,150,200,250,300,325,350,375,400],<br>'max_features': [None, 'sqrt', 'log2'],<br>'max_depth': [None,1,2,3,4,5,6,7,8,9,10],<br>'min_samples_split': [1,2,3,4,5,6,7,8,9,10],<br>'min_samples_leaf': [1,2,3,4,5,6,7,8,9,10]. | 'max_features': None,<br>'n_estimators': 250,<br>'max_depth': None,<br>'min_samples_split': 6,<br>'min_samples_leaf': 5. |
| UB | PM$_{10}$ | LSTM | 'batch_size': [24,48,72,96,120,144,168],<br>'n_steps_in': [12,24,36,48,60],<br>'hidden_size': [32,64,96,128,160],<br>'learning_rate': [1e-2,5e-2,1e-3,5e-3,1e-4]. | 'batch_size': 168,<br>'n_steps_in': 24,<br>'hidden_size': 96,<br>'learning_rate': 0.001. |
| UB | O$_3$ | XGBoost | 'n_estimators': [50,100,150,200,250,275,300,325,350,400],<br>'learning_rate': [0.02,0.03,0.04,0.05,0.06,0.08, 0.2, 0.3, 0.4],<br>"max_depth": [1, 2, 3, 4, 5, 6, 7, 8, 9, 10],<br>"subsample": [0.1, 0.2, 0.3, 0.4, 0.5, 0.6, 0.7, 0.8, 0.9],<br>"colsample_bytree": [0.1, 0.2, 0.3, 0.4, 0.5, 0.6, 0.7, 0.8, 0.9],<br>"min_child_weight": [1, 2, 3, 4, 5, 6, 7, 8, 9, 10]. | 'learning_rate': 0.04,<br>'n_estimators': 300,<br>'max_depth': 4,<br>'subsample': 0.7,<br>'colsample_bytree': 0.7,<br>'min_child_weight': 10. |

| Station | Pollutants | Models | Range of Hyperparameters | Best parameters |
|---------|-----------|--------|--------------------------|-----------------|
| UB | $O_3$ | RandomForest | 'n_estimators': [50,100,200,300,350,375,400,425,450,500,550,600], 'max_features': [None, 'sqrt', 'log2'], 'max_depth': [None,1,2,3,4,5,6,7,8,9,10], 'min_samples_split': [1,2,3,4,5,6,7,8,9,10], 'min_samples_leaf': [1,2,3,4,5,6,7,8,9,10]. | 'max_features': None, 'n_estimators': 400, 'max_depth': None, 'min_samples_split': 1, 'min_samples_leaf': 7. |
| UB | $O_3$ | LSTM | 'batch_size': [24,48,72,96,120,144,168], 'n_steps_in': [12,24,36,48,60], 'hidden_size': [32,64,96,128,160], 'learning_rate': [1e-2,5e-2,1e-3,5e-3,1e-4]. | 'batch_size': 168, 'n_steps_in': 24, 'hidden_size': 128, 'learning_rate': 0.0001. |

**Appendix D. Temporal variations in hourly mean NOx, PM10, and O3 concentrations at the urban background**

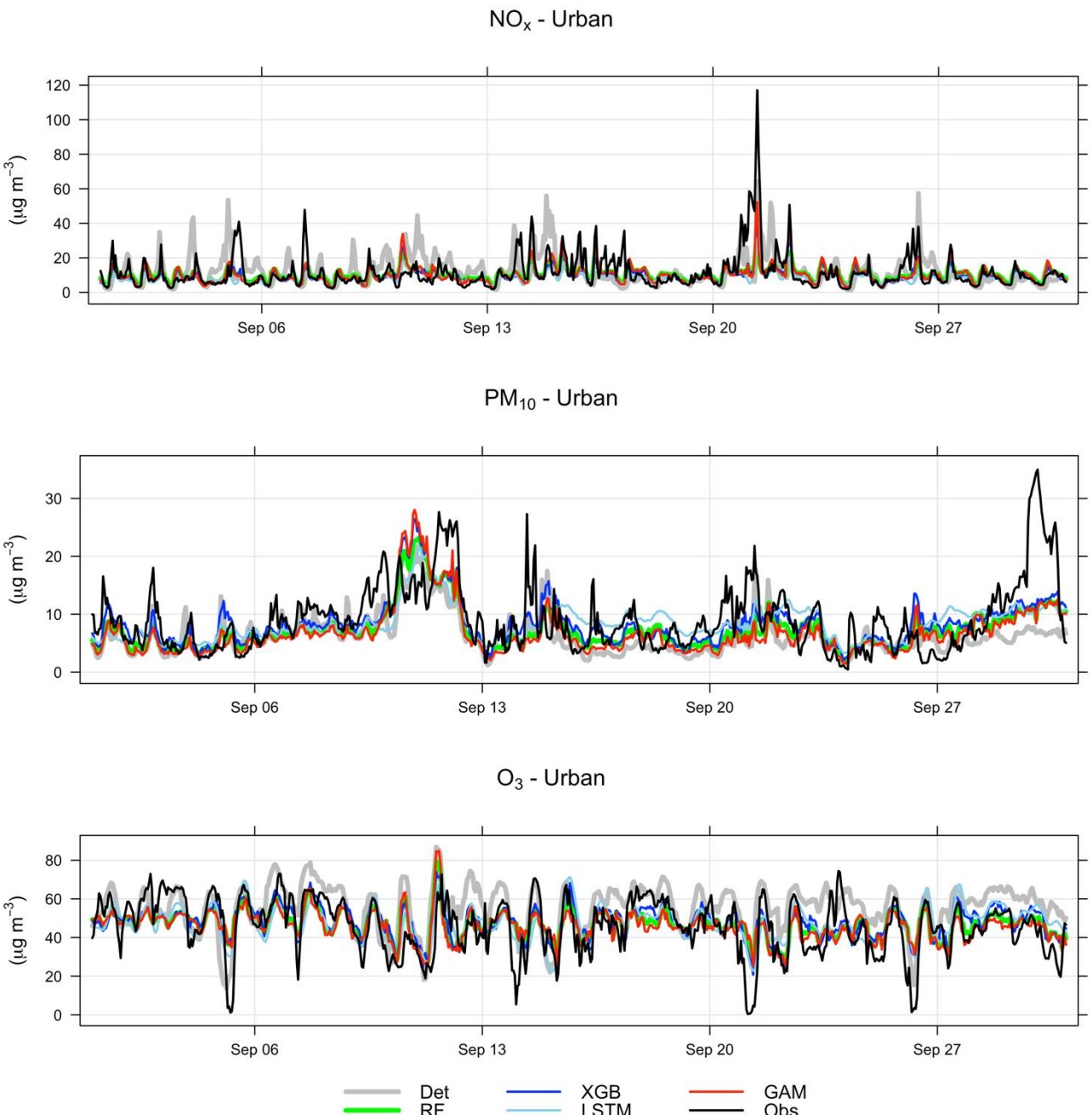

Figure D1. Temporal variations of deterministic and ML forecasted NOx, PM10 and O3 concentrations together with corresponding measured concentrations at the urban background site for September 2021. Mean of 1-, 2- and 3-day forecasts.

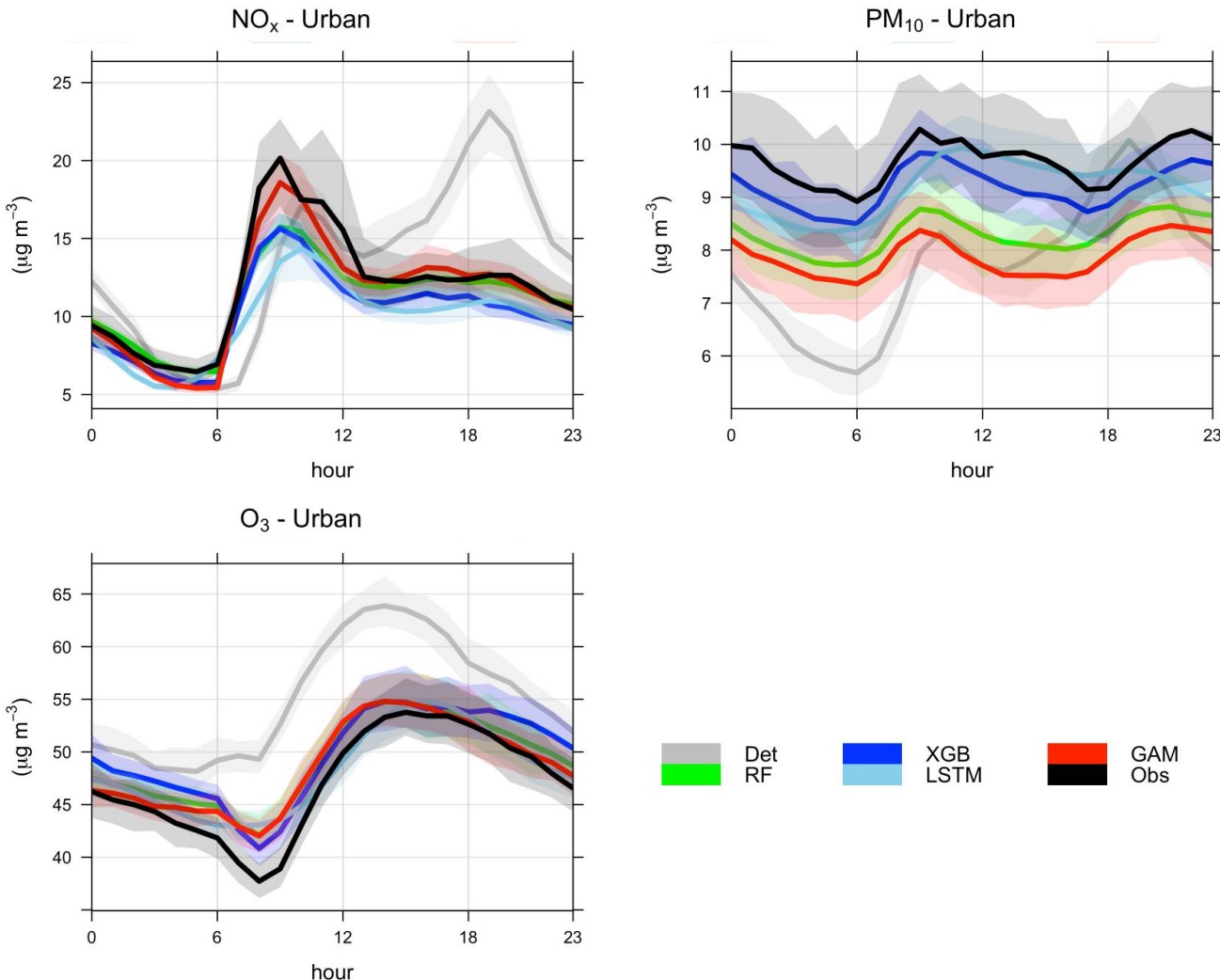

Figure D2. Mean diurnal variations in measured and forecasted concentrations of $NO_x$, $PM_{10}$ and $O_3$ at the urban site. Mean of 1-, 2- and 3-day forecasts for June – December 2021.

**Appendix E. Statistical performance measures for forecasts higher than the hourly mean concentrations at the urban site.**

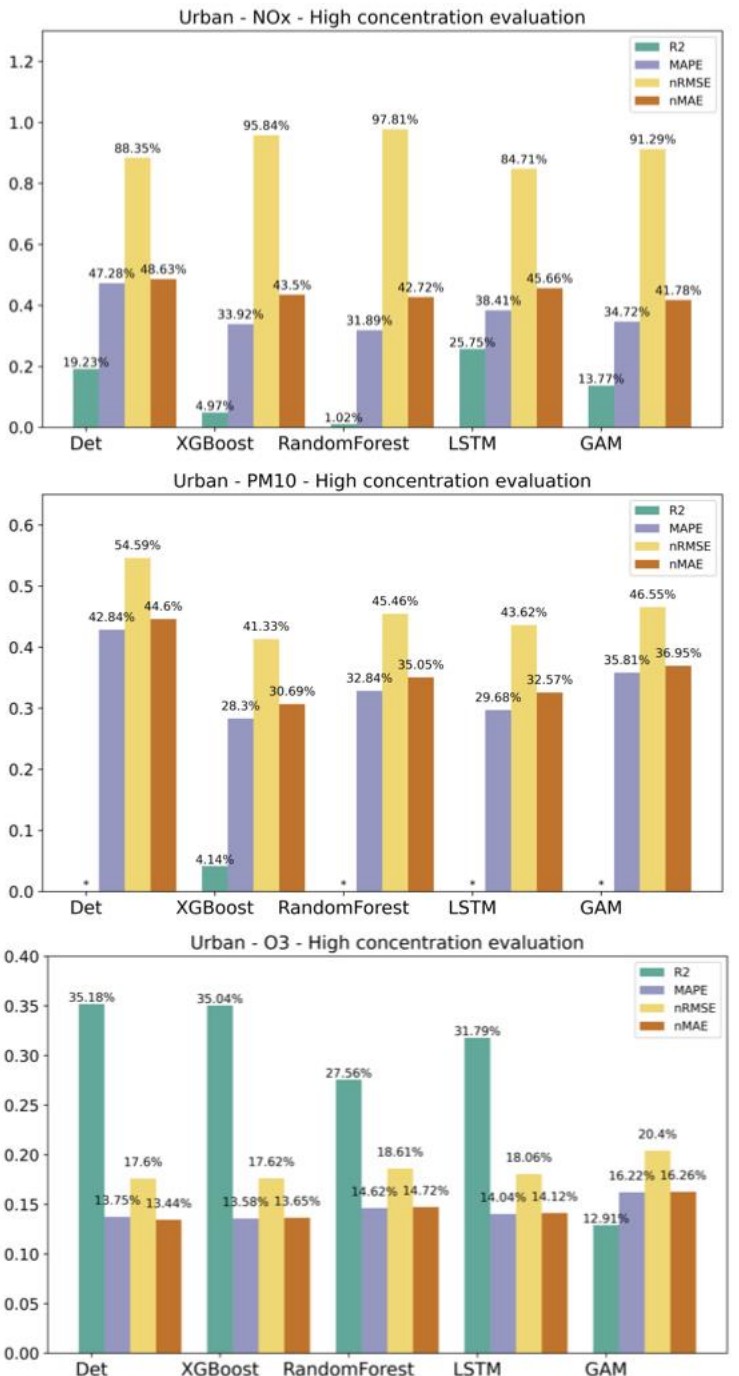

Figure E1. Statistical performance measures for concentrations of NOx, PM10 and O3 higher than the hourly mean value at

5   the urban site, where * represents a negative R2 value. Mean of 1-, 2- and 3-day forecasts.

## Appendix F Importance of features – Urban

**Urban Background - NOx - 1d**

**Urban Background - NOx - 2d**

**Urban Background - NOx - 3d**

5    Figure F1. Top 10 important features (%) for NO$_x$ forecasts using XGB, RF and LSTM at the urban site.

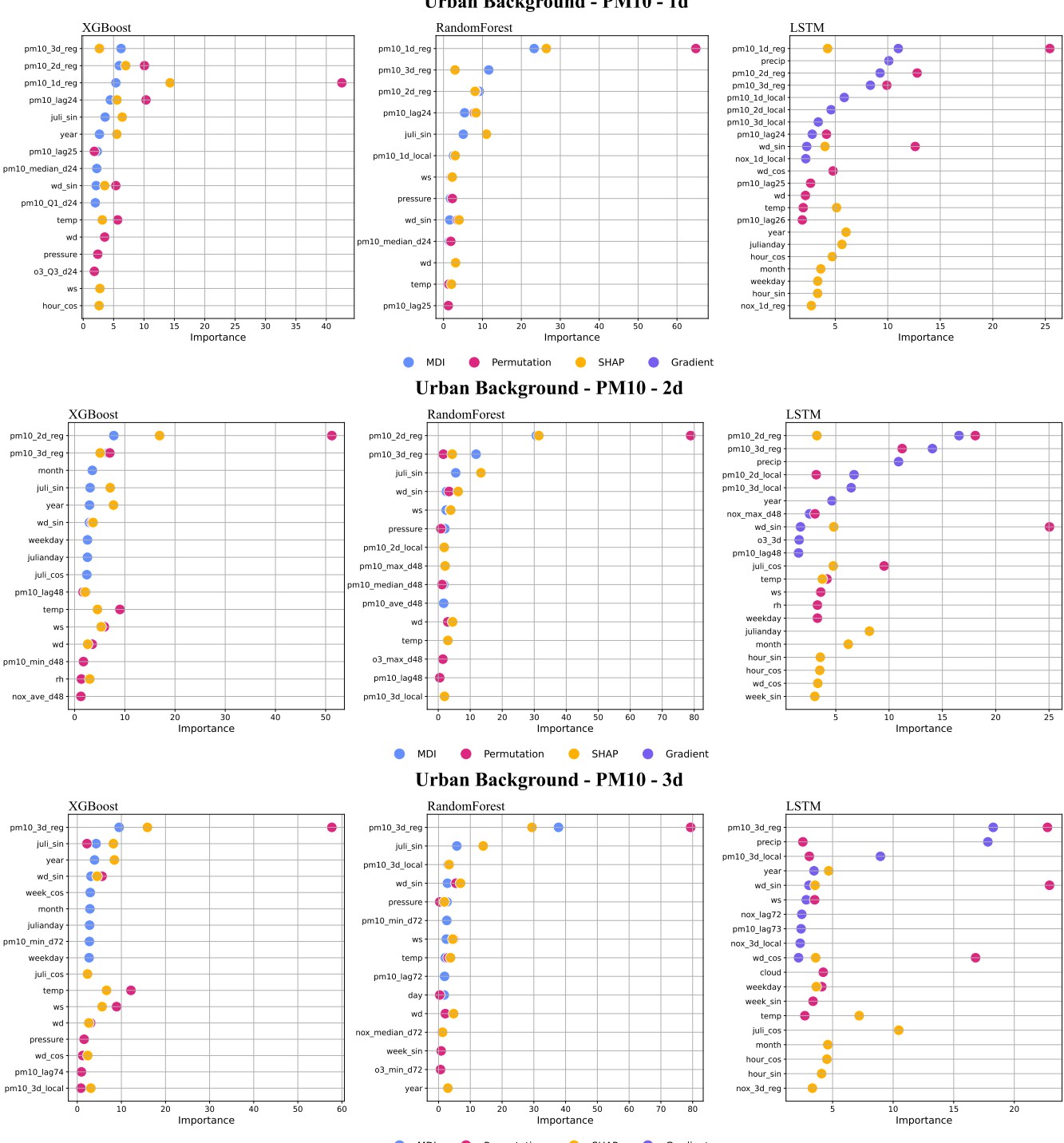

Figure F2. Top 10 important features (%) for $PM_{10}$ forecasts using XGB, RF and LSTM at the urban site.

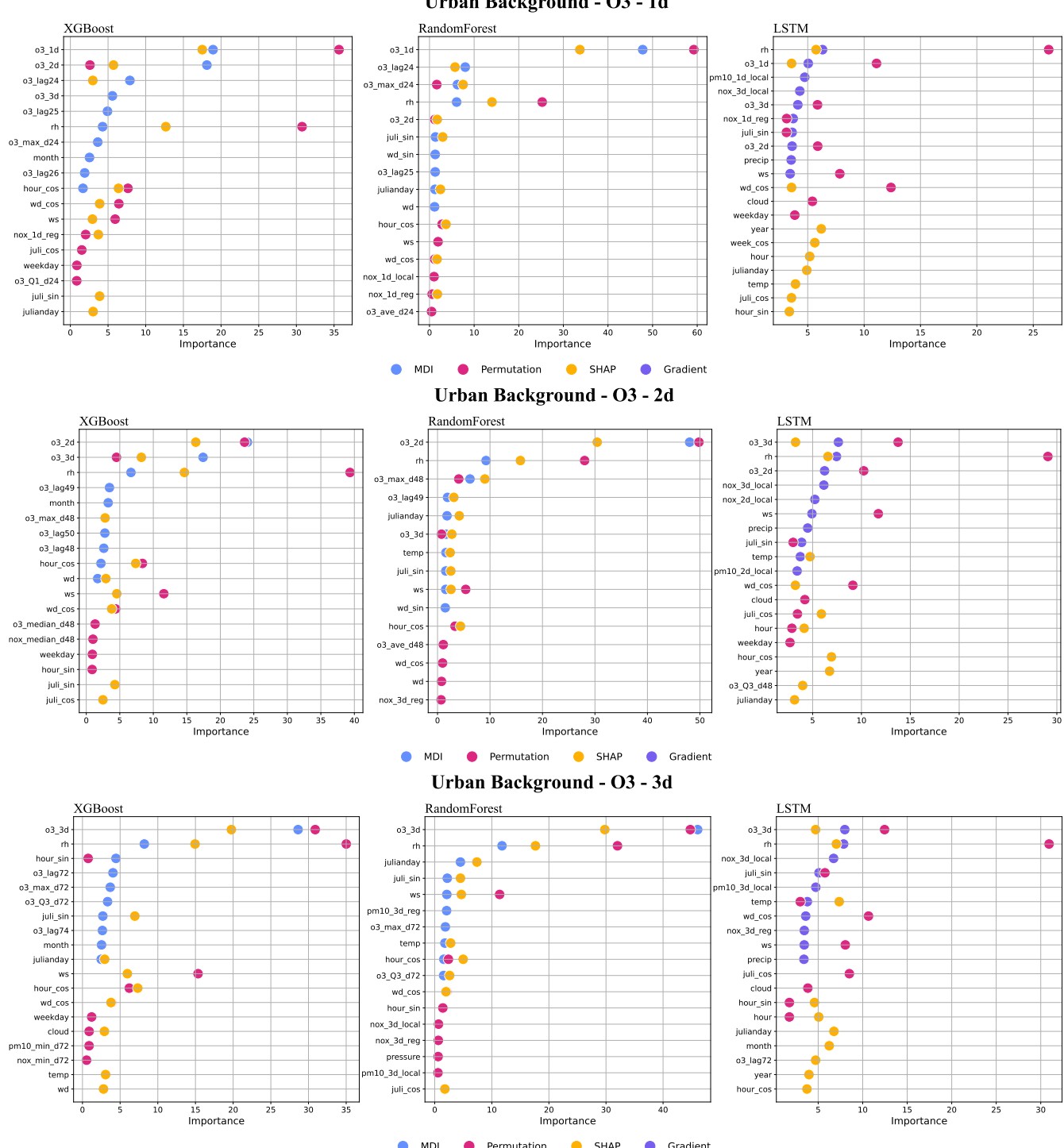

Figure F3. Top 10 important features (%) for $O_3$ forecasts using XGB, RF and LSTM at the urban site.

**Appendix G. Temporal variations in hourly mean NO$_x$, PM$_{10}$ and O$_3$ concentrations at the street canyon sites**

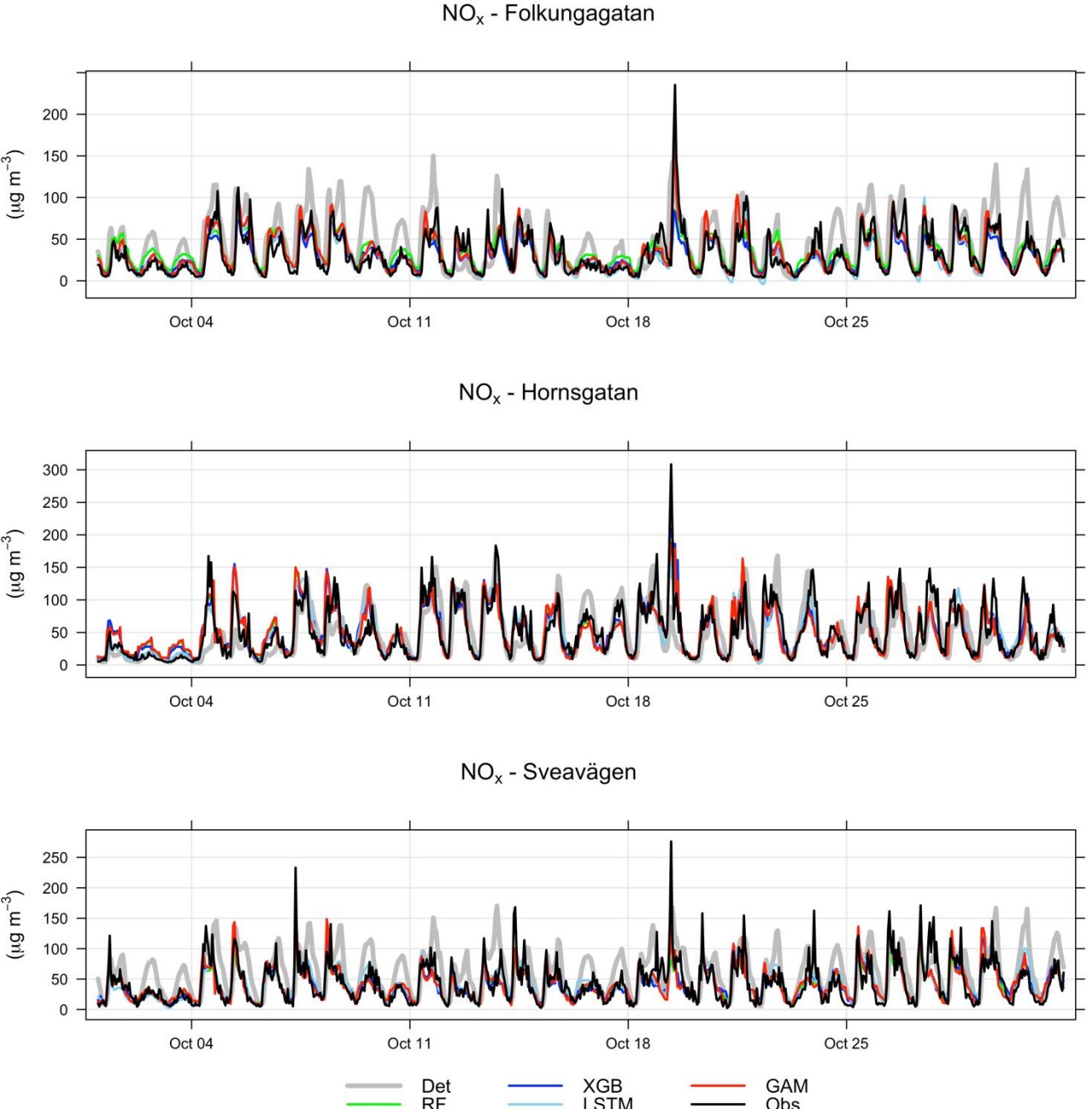

Figure G1. Temporal variations of hourly deterministic and ML forecasted NO$_x$ concentrations together with corresponding measured concentrations at street canyon sites for October 2021. Mean of 1-, 2- and 3-day forecasts.

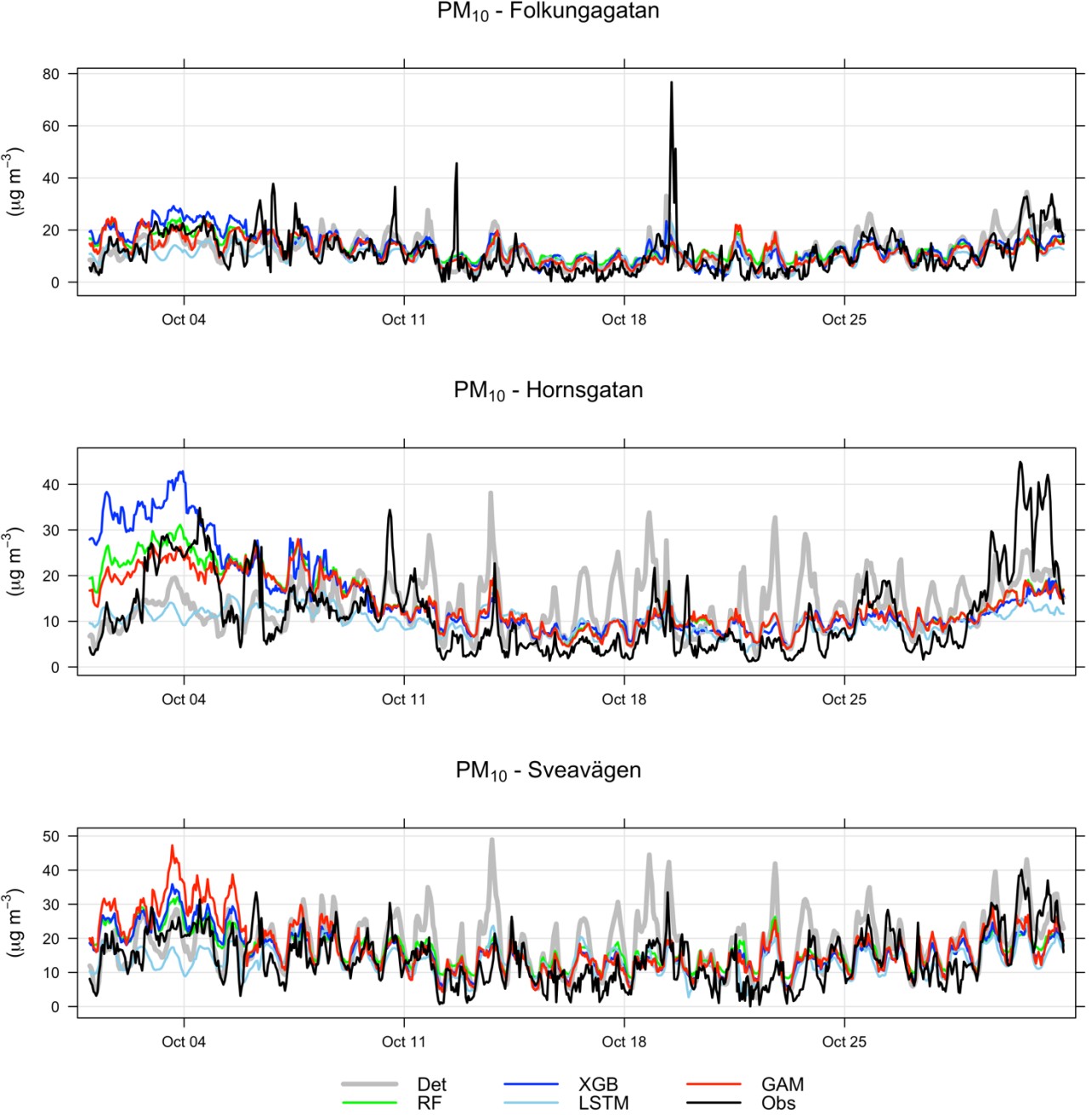

Figure G2. Temporal variations of hourly deterministic and ML forecasted PM$_{10}$ concentrations together with corresponding measured concentrations at the street canyon sites for October 2021. Mean of 1-, 2- and 3-day forecasts.

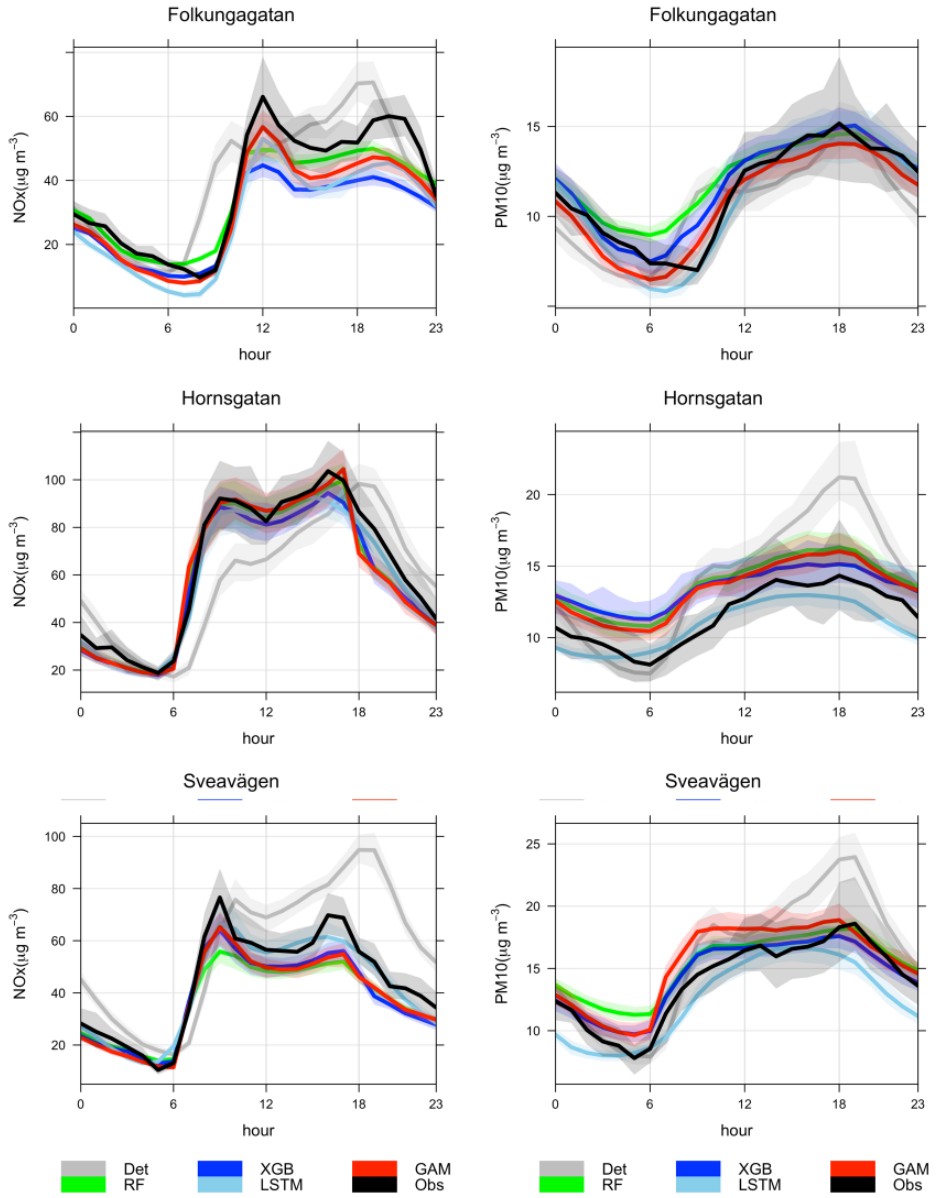

Figure G3. Mean diurnal variations in measured and forecasted concentrations of $NO_x$ and $PM_{10}$ at the street canyon sites. Mean of 1-, 2- and 3-day forecasts for September – December 2021.Shaded areas are 95% confidence intervals.

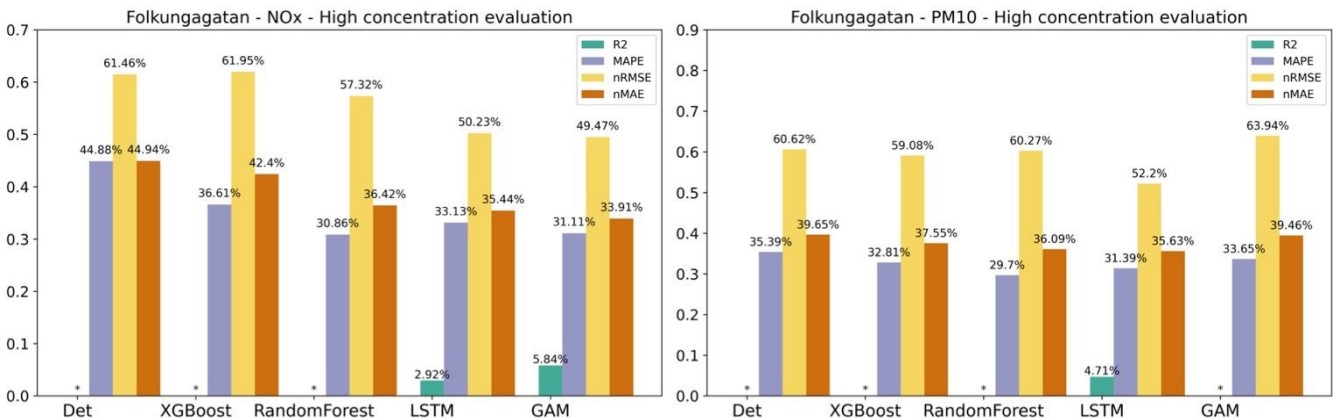

5    Figure H1. Statistical performance measures for forecasted $NO_x$ and $PM_{10}$ hourly mean concentrations higher than the mean values at Folkungagatan, where * represents a negative $R^2$ value. Mean of 1-, 2- and 3-day forecasts.

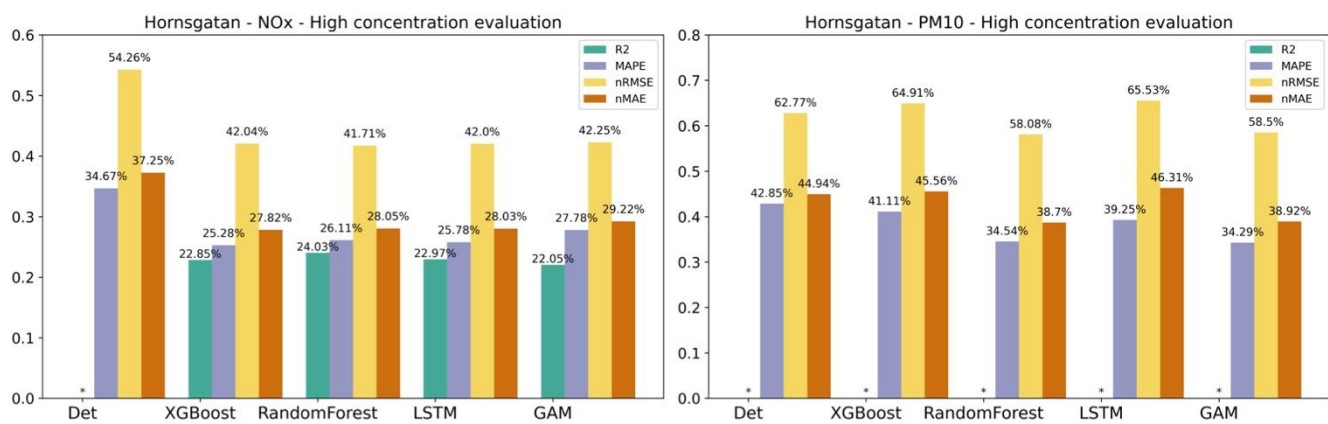

Figure H2. Statistical performance measures for forecasted $NO_x$ and $PM_{10}$ hourly mean concentrations higher than the mean values at Hornsgatan, where * represents a negative $R^2$ value. Mean of 1-, 2- and 3-day forecasts.

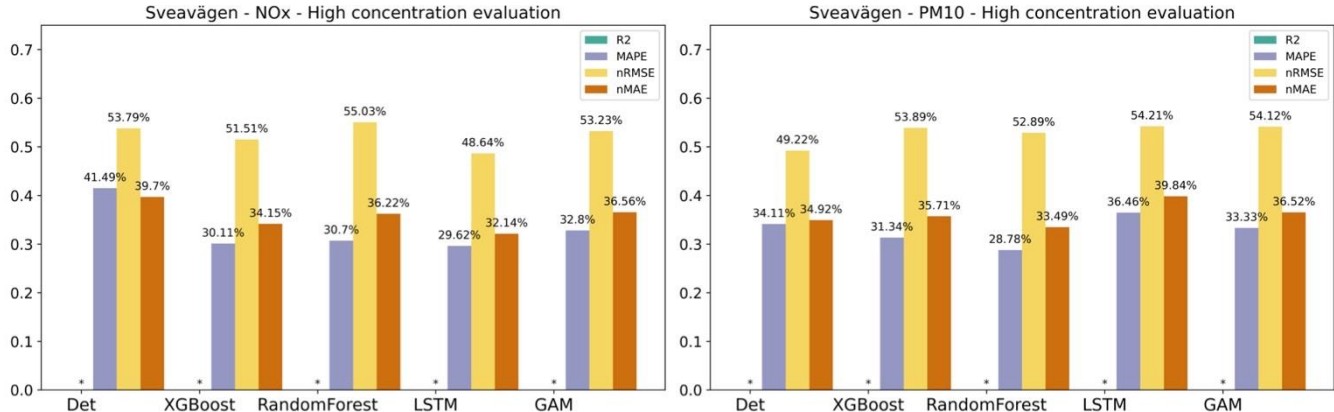

Figure H3. Statistical performance measures for forecasted $NO_x$ and $PM_{10}$ hourly mean concentrations higher than the mean values at Sveavägen, where * represents a negative $R^2$ value. Mean of 1-, 2- and 3-day forecasts.

## Appendix I. Importance of features – Street Canyon sites

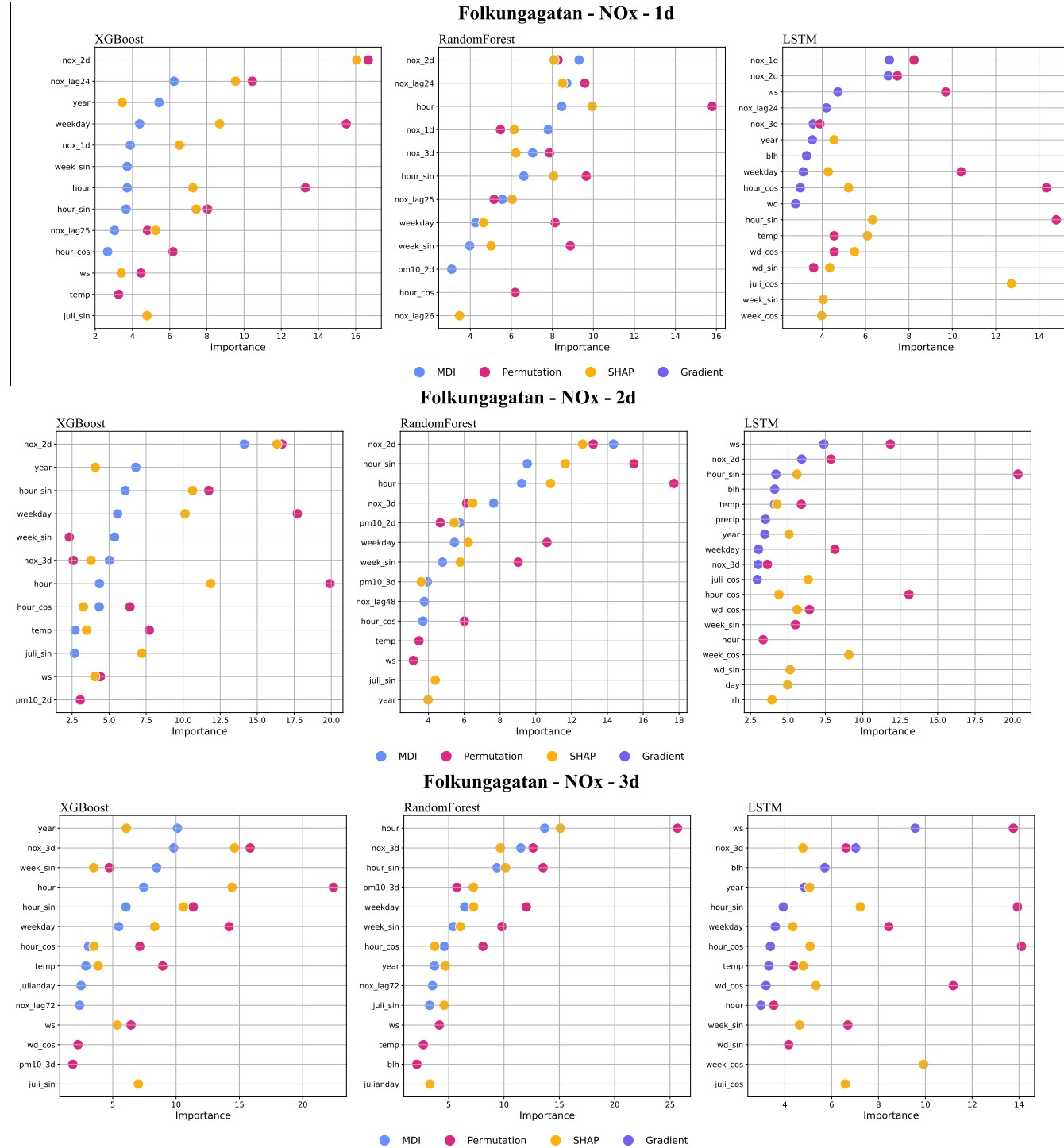

5    Figure I1. Top 10 important features (%) for NO$_x$ forecasts using RF, XGB and LSTM at Folkungagatan.

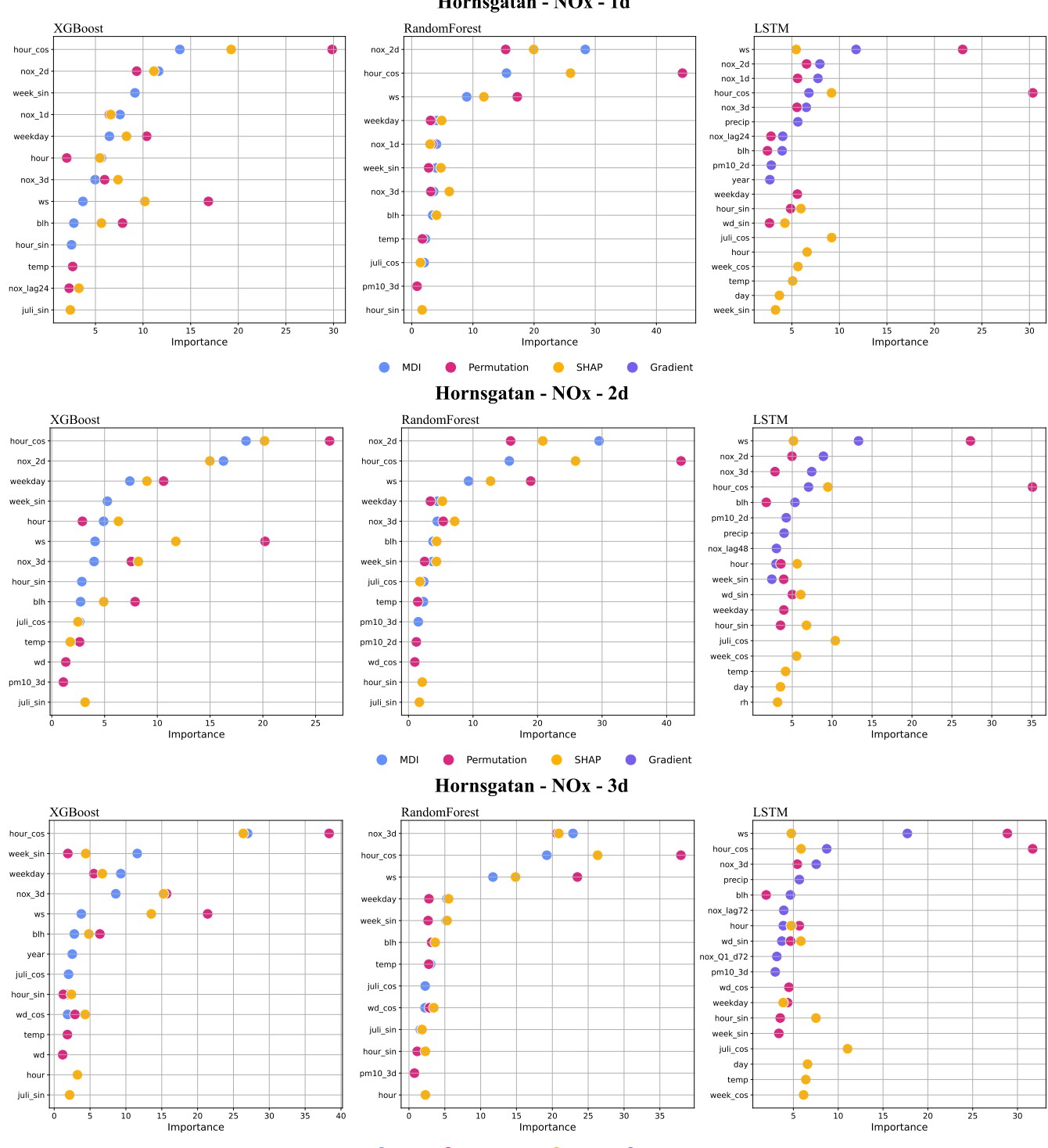

Figure I2. Top 10 important features (%) for NO$_x$ forecasts using RF, XGB and LSTM at Hornsgatan.

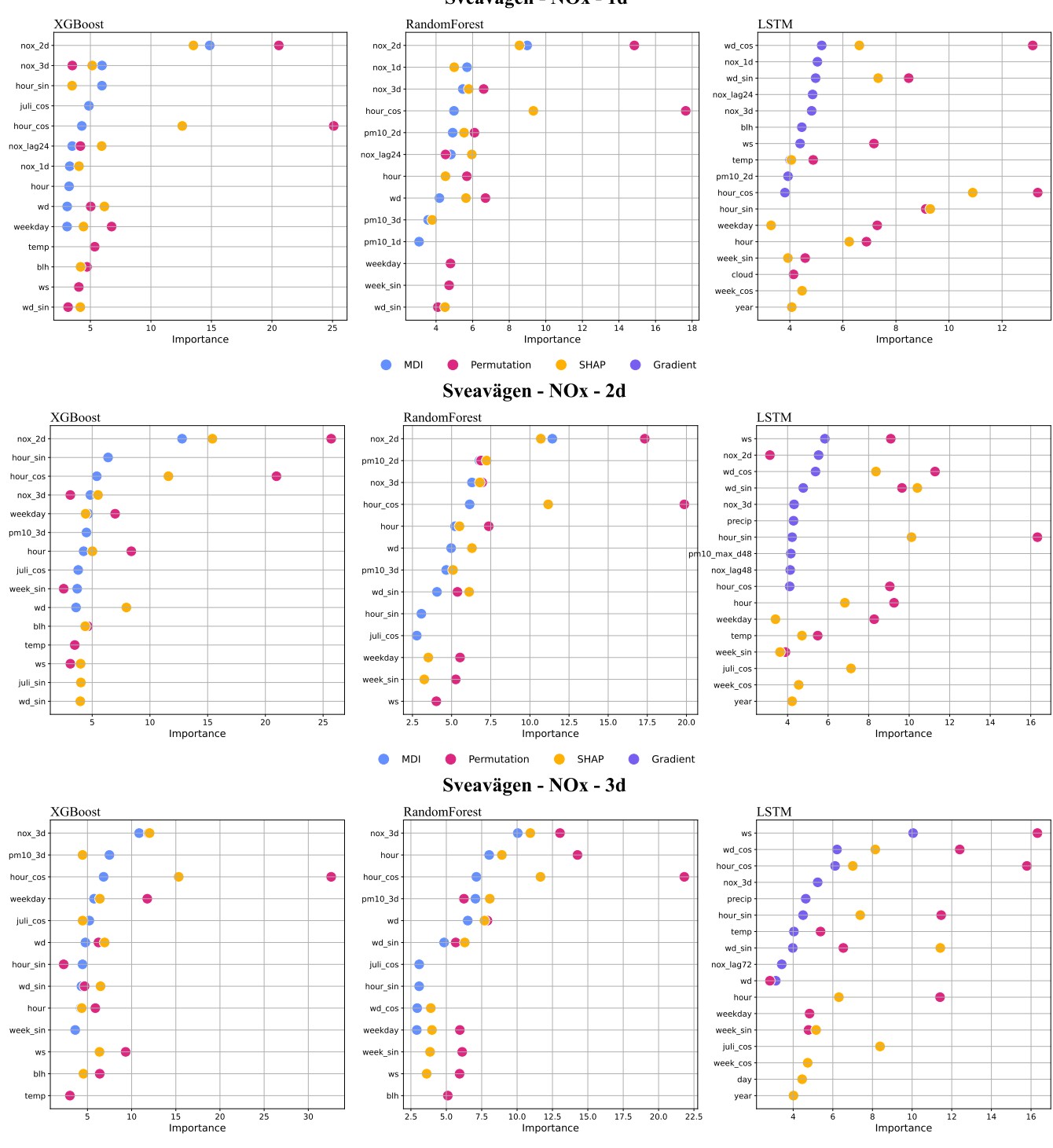

5    Figure I3. Top 10 important features (%) for NO$_x$ forecasts using RF, XGB and LSTM at Sveavägen.

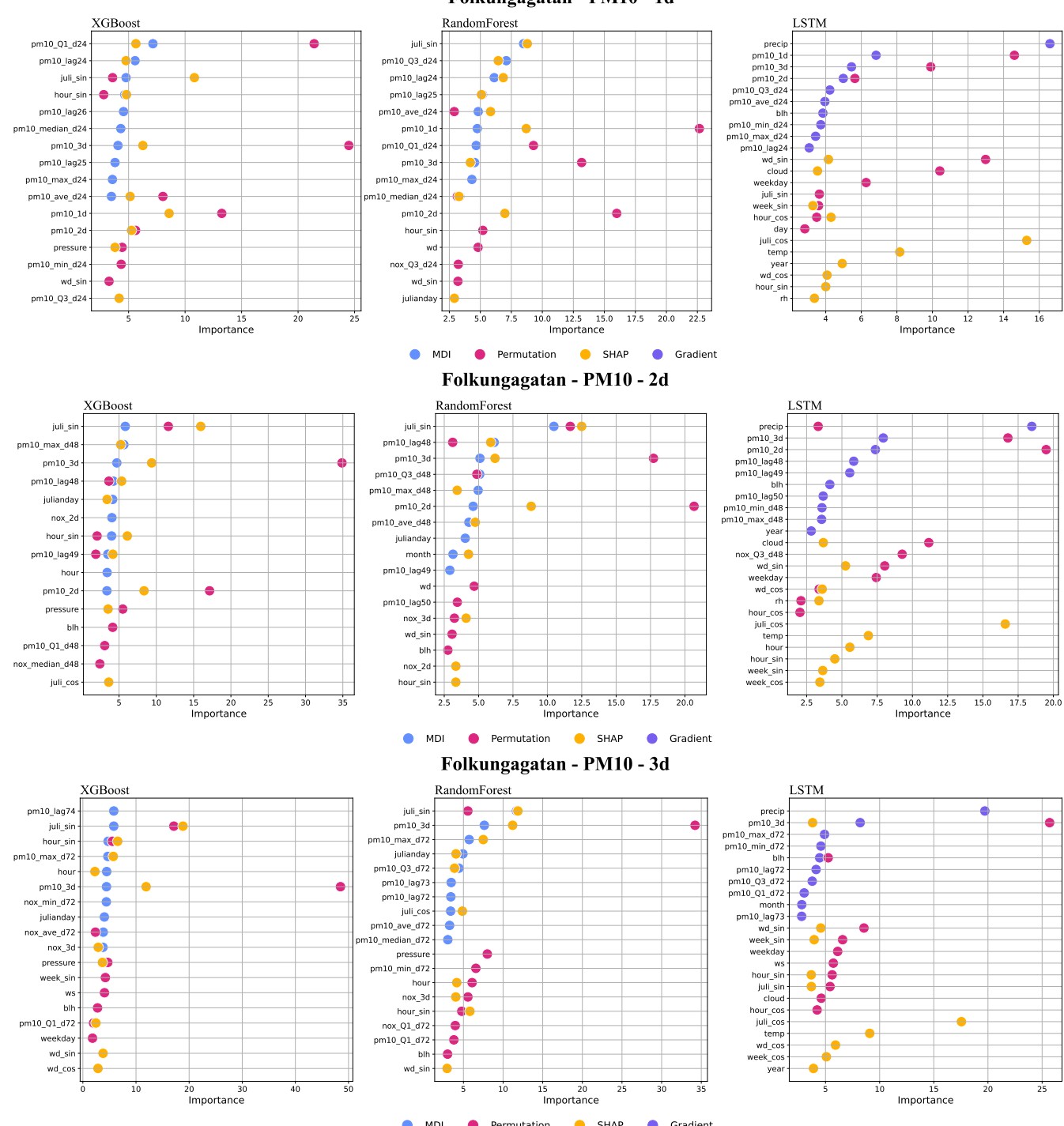

5    Figure I4. Top 10 important features (%) for PM$_{10}$ forecasts using RF, XGB and LSTM at Folkungagatan.

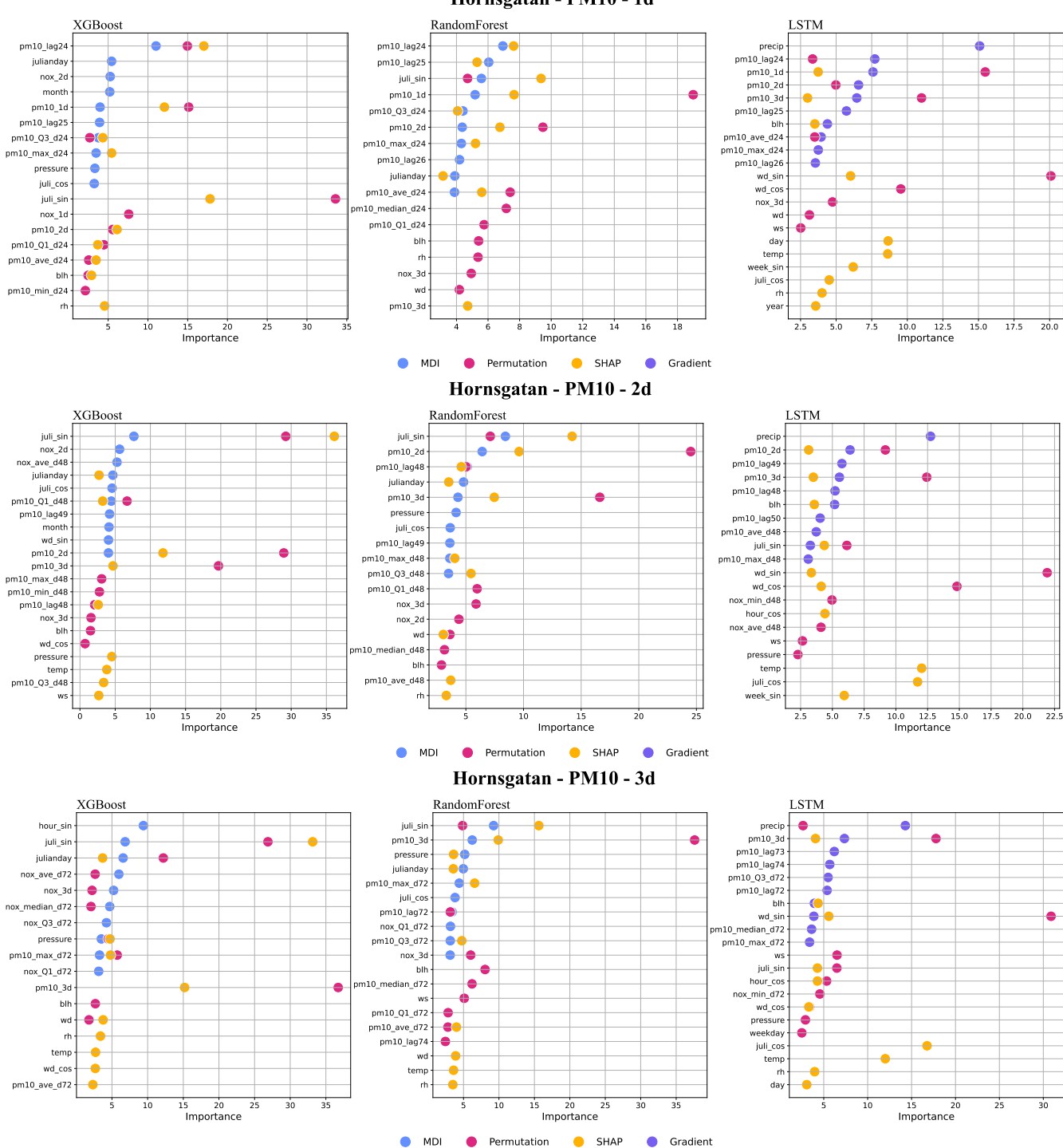

5    Figure I5. Top 10 important features (%) for PM$_{10}$ forecasts using RF, XGB and LSTM at Hornsgatan.

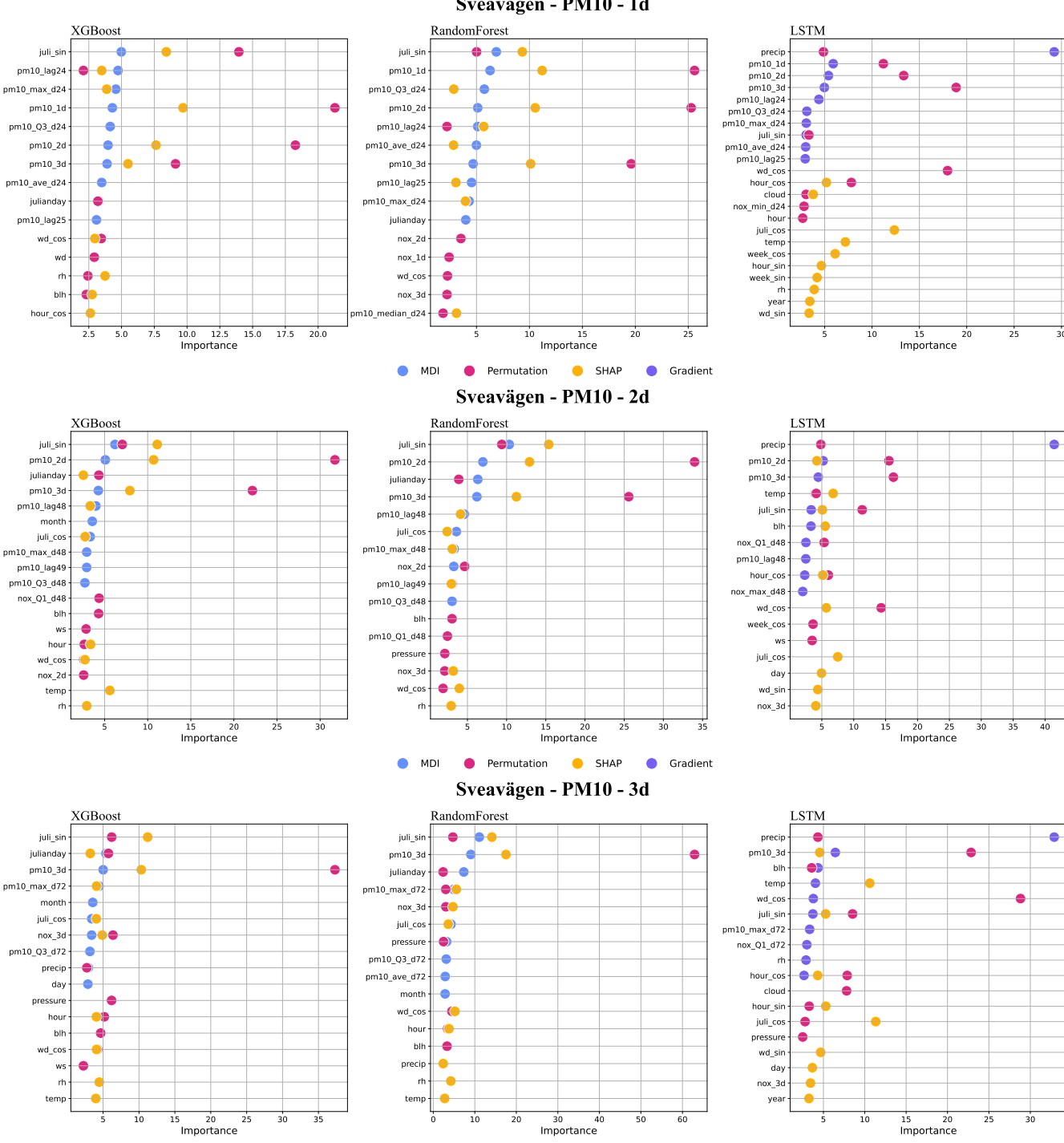

Figure I6. Top 10 important features (%) for PM$_{10}$ forecasts using RF, XGB and LSTM at Sveavägen.

**Code/Data availability:** Python codes and data are available here: https://zenodo.org/records/8433033.

**Author contribution:** CJ and MS have provided initial inputs on the AQ prediction modelling approaches. ME has been responsible for deterministic modelling and providing monitoring data and meteorological forecasts. ZZ and XM have been responsible for the ML modelling and optimization, feature importance analysis and statistical calculations. CJ, XM and ME initiated and planned the project that supported this study. All authors have contributed to analysing data and writing the manuscript.

**Competing interests**: The authors declare that they have no conflict of interest.

**Acknowledgements:** Financial support: The project was funded by ICT – The Next Generation and Digital Future at KTH Royal Institute of Technology (contract VF 2021-0082).

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
