# Peer review of "Improving 3-day deterministic air pollution forecasts using machine learning algorithms"

_Atmospheric Chemistry and Physics, 2023_

## Author Comment (AC1)

Response letter for reviews of "Improving 3-day deterministic air pollution forecasts using machine learning algorithms"

We thank both referees for valuable comments.

Referee comments in black and replies in *blue, italics*.

Replies to Referee #1

The authors present several ML models for air quality forecasting focusing on PM10, NOx and compare those results against deterministic forecasts. The dataset used to train ML models focuses on Stockholm. Overall, the authors show that the ML investigated seem to outperform the deterministic forecasts over the periods that the models were tested on, which were 1-, 2-, and 3-days in length periods. The ML models considered are very standard methods used widely in the literature, and are probably the appropriate starting points given the structure and size of training data. The authors additionally show some feature importance results.

In my opinion the length of this paper is too long and the results section needs to be shortened, the overall results summarized, and the key findings should be emphasized more. Both Table 2 and 4 could go to the Appendix as well as most of the figures in the results section. It might be easier to present/condense the text results in the tables as figures. I think the feature importance plots should be in the main text.

*Reply:*
*It is true that we used standard machine learning models for longer-term air quality prediction. The contribution is not applying the most state of the art ML models but demonstrating the improvement of data-driven ML models on such kind of prediction problem over conventional deterministic models, which fits the scope of this journal. Even without very advanced ML modelling structure, the complexity and amount of the work is large concerning the setup of the longer-term forecasting scheme, empirical modelling work with different sites, analysis of the prediction results and so on. Meanwhile, these algorithms have been implemented in a real AQ prediction system for Stockholm city. Therefore, the paper has a lot of material, leading to current length. We have tried to shorten the paper but it ends up with some extension to answer referees comments.*

*The reason Table 2 and 4 are kept in the main text is that they show the differences in 1-, 2- and 3-day forecasts for the different models and pollutants, which is not shown in the Figures. It is difficult to condense these in a Figure.*
*In fact, adding all feature importance plots to main text makes the presentation of the results even more difficult to follow. In that case, we have to add more illustrations, making the paper even longer than current version.*

In the abstract it is wrongly stated that one cannot subject LSTMs to feature importance methods. A google search provides examples of how this can be performed. I think the authors need to investigate a gradient-based feature importance method for the LSTM and compare that model against the other models investigated, since the application of feature importance alongside the usage of ML models for longer-term horizon prediction is a main focus of their paper.

*Reply:*
*Thank you for the comments. We agree with the referee that the statement is not correct. The statement in the abstract is due to some misunderstanding in our communication. We have now revised the texts and included feature importance also for LSTM.*

*However, how to interpret feature importance has been a side-line topic for understanding the RNN model. On the other side, one essential idea of RNN is to automate the feature engineering process because the importance of features can be trained by adjusting the weights of the connections. For RNN, there are some methods in recent studies for ranking the feature importance e.g. gradient-based, perturbation-based, or Shaley value sampling approaches etc(Ismail et al., 2020). But the interpretation of the results could be different for the same model. Also, there are issues such that the gradient-based method calculates temporally varying rankings making the ranking of featrues depend on the testing set..*

*But, even so, we have added gradient-based feature importance results for the LSTM model in which the results are the average of the gradients of all samples for each testing set. The results are shown in Appendix B and Appendix E.*

Furthermore, the authors did not mention which importance method was used for the tree-based models. There are now a variety of methods available for these models, which often do not agree on the importance ranking. For the XGB model, how do other feature importance metrics (potentially such as the permutation and SHAP importance) compare to what was used?

*Reply:*
*We have added some details at the end of subsection 2.4. For the RandomForest and XGBoost models, feature importance is ranked based on the mean decrease in impurity,which also serves for feature selection of the models. There are other approaches to measure feature importance such as (Zhou et al., 2021) but this is beyond the scope of our current study.*

It is not clear in the paper how the data sources were combined and then split into training, and validation splits.
Furthermore, it was not stated if any preprocessing of the data was performed (which probably needed to be carried out given the different ranges the input quantities cover). How large was the data set?
*Reply:*

*We have added a Table to illustrate the basics of four datasets at the end of subsection 2.1, as well as making the datasets publicly available for easy viewing and further investigation.*
*Regarding data splitting, we split the dataset along the time axis into non-overlapping training, validation and test data in a ratio of 16:4:5. That is, the validation set is the latter 20% of the total training set and the test set is the latter 20% of the total data set.*

*In the pre-processing process, outliers, such as negative pollutant measurements, are identified and removed and furthermore, standard methods, such as interpolation, are applied to handle missing values in the data. Given the current length of the paper, adding such details will not benefit the paper's readability. But we have added a description in subsection 2.1 as follows:*

*"The measurement data with a missing rate of less than 5% and missing values are replaced with mean values of available data in the neighbourhood according to the respective autocorrelation properties."*

The authors state that the ML models were trained on the same data; were the tree models trained on randomized (tabular) data, or was it split some other way (was the time-dependence preserved)?
*Reply:*
*We have added an explanation in subsection 2.4 below.*

*"Due to the temporal correlation of the air pollutant concentrations, the principal assumption of cross-validation is not satisfied. To preserve the time-dependent property, "TimeSeriesSplit" was chosen as the cross-validation strategy. In the $k_{th}$ split, it turns the first k folds as the training set, and the $(k+1)_{th}$ fold as the test set. The value of parameter k is set as 5."*

Figure 3 makes sense for the LSTM; however I think the authors could extend the figure to include some more schematic details pertaining to the tree-based models studied (and a schematic LSTM could be helpful as well).

*Reply:*
*Figure 3 describes a rolling prediction scheme, not only for LSTM but also for the other machine learning methods. The graph emphasizes that we have a prediction horizon of 24, 48, 36 hours. Note that this is not sequence to sequence model that is typical for RNN, and we only predict a single value at 24, 48 or 36. The values in the delayed horizon are used to calculate features e.g. statistics of the inputs.*

Models
It becomes clear that there are in fact many models being used. How many relative to (presumably) Figure 4 (it currently says "Fel! Hittar inte referenskalla")? The authors should probably add some comments about how this considerably complexifies the overall model pipeline, relative to one single model being used, and limits model(s) generalizability. Is this really a better solution relative to one single LSTM in terms of complexity?

*Reply:*
*The erroneous reference for Figure 4 has been corrected.*

*The idea of using several models is to compare the performance of different machine learners in improving the deterministic forecasts. There are advantages and disadvantages associated with different models and we show that the performance of the models depend on pollutant and site. RNN has been widely applied for AQ prediction in literature but not so much for longer term prediction like we did. A limitation applying RNN in environmental science is the difficulty in interpreting the result. The conventional ML methods have some advantages in this aspect. In this study, one main concern is to deploy feasible algorithms for a real AQ prediction system. It updates the training process regularly once new data is accumulated. Relatively simpler algorithms are appreciated at the moment, as the current system does not have extensive GPU resource for training. Nevertheless, for research we go beyond the current ML algorithms and use our national HPC resources for deep learning models.*

There is no detail provided on model and training hyperparameters, and whether hyper parameter optimization was performed, for any of the models. How large was the LSTM (how many layers, layer size, what activation functions were used, etc.). Similarly, what were the XGB parameters used? Were all of the parameters guessed (or defaults used)? Overall, the models mostly look comparable.

*Reply:*
*Thank you for the comments. We have added the detailed model parameters. These are also summarized below.*

*The two tree-based models use default parameters of the library "scikit-learn" since a rough grid search didn't enhance the model performance significantly. We also recognize that hyperparameter tuning is needed for each dataset to get the optimal models. But the improvement is not so significant in comparison to what we achieved over the deterministic model.*

*The LSTM model consists of two layers, each with 100 neurons, and passed through a fully connected layer before the output. The activation function was a "Tanh".The LSTM model was trained by Adam optimizer. The batch size is set as 72. The initial learning rate is 0.01 and is automatically adjusted using "ReduceLROnPlateau" with the parameter patience set to 10, i.e., training is stopped when the loss of the validation set is detected as not decreasing for 10 consecutive epochs."*

Given the results as they are in the current manuscript, it seems the obvious choice is RF, but I think the authors also need to compare linear regression to the other models, which should be considered the baseline ML model that the others need to beat. I don't see much value in including the GAM (other than ruling it out).

*Reply: For some statistical performance, especially bias measures, GAM gives better results. Other studies have also shown that ensemble models based on GAM can further improve predictions of concentration for some pollutants. We believe it contains useful information and prefer to keeping it in the model list.*

Finally, it was not clear if cross-validation was performed and if the presented results show ensemble averages or something else. I think by now this is a very standard procedure, and it  also provides an estimate of the uncertainty present in the models prediction capabilities given the training data set (for example, the tables should be presenting mean and variances for the ensemble). Given that there are so many models at play, it might be more useful to understand when models are more or less certain in their predictions.

*Reply:*

*As mentioned in the previous response, we used the cross-validation method of TimeSeriesSplit to preserve the time-dependent properties. In addition, we trained by setting different random number seeds, after several iterations, to obtain the optimal results. In the face of so many models, how to combine multivariate data and construct a unified model framework will be the next step of our research.*

Other

Most of the results in later figures show the mean of 1-,2-,3- day forecasts. How does the performance depend on day?

*Reply: A comparison of the performance for 1-, 2- and 3-day forecasts is show in Table 2.*

I'd rather the authors show the coefficient of determination (e.g. R2) alongside/rather than the Pearson coefficient.

*Reply: It is possible of course, but questionable if the extra work is worth it as the conclusions will be the same.*

Line 10: The deterministic predictions are used as models' inputs but at which time? Is it the current prediction (and the models' job is to correct CAMS?).

*Reply: The deterministic forecasts starts from 01:00 (mean value for 00:00 to 01:00) and are provided for all coming 24, 48 or 72 hours. The MLs job is to improve the deterministic forecasts, which are based on both CAMS and the local Gaussian and Street canyon model.*

Throughout the paper the authors use "MLs" but this should probably read ML models for grammatical consistency.

*Reply: Yes, we have changed this.*

In Figure 5 when there is a large drop just after September 20, what did the XAI method claim was the important feature(s). Is there anything different about the inputs to the model on that date?

*Reply: We have not evaluated the importance of features for specific short time periods. Such analysis might be something to include in a future study.*

*References*

*Ismail, A. A., Gunady, M., Corrada Bravo, H., & Feizi, S. Benchmarking deep learning interpretability in time series predictions. Advances in neural information processing systems, 33, 6441-6452, 2020*

*Zhou, Z., & Hooker, G. Unbiased measurement of feature importance in tree-based methods. ACM Transactions on Knowledge Discovery from Data (TKDD), 15(2), 1-21, 2021*

Replies to referee 2

This paper describes and evaluates the use of different machine learning algorithms to make short-term forecasts of ground-level air pollution at 4 observational stations in Stockholm, Sweden. While the paper certainly adds to the current literature, and fits within the scope of the journal, the paper lacks some important details and is sometimes difficult to follow. My major comments relate to the need for more clarity on the exact methodology used, particularly with reference to the data processing. Clarity on these details is necessary before publication.

*Reply:*
*Thank the reviewer for taking time to go through our manuscript and for the constructive comments. We have now carefully modified our manuscript based on the reviewer's comments.*

Specific major comments:
1. It is unclear how the data is split for model training, validation and testing. What is the percentage split of this data? Crucially, how was the data split? Was it split randomly, or was it split temporally (if so, what were the dates for the training/test data)? Some details are necessary here, as improper data splitting can lead to over-inflation of model skill results due to data leakage.

*Reply:*
*Data splitting is not random to prevent data leakage. We split the dataset by the TimeSeriesSplit strategy, that is, splitting along the time axis into non-overlapping training, validation, and test data in a ratio of 16:4:5. The validation dataset is the latter 20% of the total training data and the test set is the latter 20% of the total data. This is explained in section 2.4.*

2. Did the authors carry out hyperparameter tuning for their models? If so, did they use a validation set to do this tuning, and then evaluate on a held-out test dataset? Currently there are no details on hyperparameters in the manuscript. It is stated in the conclusion that fine-tuning is possible future work - does this mean that defaults were used for the hyperparameters?

*Reply:*
*The two tree-based models use the default parameters of "scikit-learn". We tune the models using grid search but the performance improvement is not so significant. Especially, the hyper parameters shall be optimized for each dataset. The LSTM model consists of two layers of LSTM, each with 100 neurons, and passed through a fully connected layer before the output. The activation function was a "Tanh". The LSTM model was trained by Adam optimizer. The batch size is set as 72. The initial learning rate is 1e-2 and is automatically adjusted using "ReduceLROnPlateau" with*

*the parameter patience set to 10, i.e., training is stopped when the loss of the validation set is detected as not decreasing for 10 consecutive epochs.*
*. Specific parameter information has been added to subsection 2.4.*

3. The authors state that the LSTM model is not used to its full potential. I am unclear what this means. As I understand it, the LSTM was not trained to make autoregressive predictions of air pollution, as would typically be the case when forecasting with an LSTM? If this is the case, I imagine this to allow comparison between the LSTM and the tree-based methods. However, this is not really a fair test of the skill of the LSTM, and the possible advantages it provides against tree-based methods. Some clarity on this would be appreciated.

*Reply:*

*Thank you for the comment. A simple LSTM model was deployed in our study for the prototype air quality prediction system. While this paper does not focus on innovative machine learning algorithms, we believe that the deep learning model can be further improved for its prediction accuracy. Indeed, we have been investigating more complex models such as multi-layer LSTM network, and advanced variants of LSTM, such as CNN-LSTM and Bi-LSTM models. But this is beyond the scope of this study.*

4. It is unclear how the feature importances were extracted from the tree-based methods. There are a number of different methods for this. Was this based on permutation-based feature importance, for example? Why was the particular importance method chosen, and what are the drawbacks of using this method? In addition, there are methods to extract feature importances from LSTM models that could be used, to compare against importances from the tree-based models.

*Reply:*

*We have added some details at the end of subsection 2.4 to answer the issue raised by both reviewers.*
*For the RandomForest and XGBoost models, feature importance is ranked based on the mean decrease in impurity (MDI), and it enables feature selections for the models. The low computational cost of this algorithm and the relative accuracy of the results achieved are important reasons for our choice. There are recent studies for unbiased*

*feature importance e.g. (Zhou et al., 2021) but it is beyond the scope of the current study..*

*In addition, we have added gradient-based feature importance results for the LSTM model in which the results are the average of the gradients of all samples in the test set for each dataset, shown in Appendix B and Appendix E.*

5. It is stated that data from the UB site covers around 1000 days, while the street canyon data extends over 500 days. Were the percentage splits of the training and testing data similar for both cases? Given there were fewer data for the street canyon sites, might this affect model performance? Finally, given that the data comes from 2019-2021, might the impact of coronavirus restrictions affect model skill, or model generalisation to future data?

*Reply:*

*UB data and street data use the same data split ratio.*

*The size of the dataset will affect the performance of the model, especially the LSTM model, but we think that the current data set size is sufficient and does not significantly affect the result. The real system will always be trained with the latest incremental data.*

*During the COVID-19 pandemic, there have been obvious mutations in the operation of the city, which also affects the changes in pollutants. We have deployed the model to the forecasting system in Stockholm and retrained the model regularly to ensure the model's ability to generalize to future data.*

Other specific comments:
1. Line 5: some citations for reduced lung function etc. would be welcome.

*Reply:Done.*

2. Line 26-28: 'Although… the challenges of forecasting air pollution concentrations in a longer-term horizon such as a day or even several days have not been investigated'. If this is referring to multi-day daily forecasts of pollutants, this is not true. There is a significant body of work looking at

forecasting air pollution on the time horizon of several days e.g., Kleinert et al, 2022 for ozone.

3. Line 28-29: 'very few studies have combined deterministic models and ML in forecasting air pollution levels of a few hours/days in the future'. There are some studies that do this. They should be cited here.

*Reply to both comments above:*

*We have modified this scentence:*

*"Forecasting air pollution concentrations in a longer-term horizon such as a day or several days have been investigated by e g Kleinert et al. (2022) for $O_3$. Some studies have also combined deterministic models and ML in forecasting air pollution levels of a few hours/days in the future (e g Hong et al., 2022), but mostly for one single pollutant at the time. "*

4. Page 4, line 15: 'whereas the alternative approach substitutes the missing values with mean values of available data in the neighbourhood'. Substituting with the mean value from the other 3 stations? Or the nearest station?

*Reply:*

*In the pre-processing process, outliers, such as negative pollutant measurements, are identified and removed and furthermore, interpolation are also applied to handle missing values in the data. We have added a short description in subsection 2.1 as follows:*

*"The measurement data with a missing rate of less than 5% and missing values are replaced with mean values of available data in the neighbourhood according to the respective autocorrelation properties."*

*Specifically, the adjacent data refers to a sample without a missing value near the autoregressive period. For example, the autoregressive period of $NO_X$ is 24 hours, determined by the autocorrelation diagram and partial autocorrelation diagram, that is, the average value of 24 hours ago and its adjacent data (23 hours ago and 25 hours ago) is calculated and interpolation into the missing value.*

5. Page 5, line 9: 'are extracted from a location outside the greater Stockholm domain' - what location?

*Reply: 59.50N, 18.35E. This point is chosen to be close to the boundaries of our greater Stockholm model domain north of Stockholm. There are no real observations at this site.*

6. Page 8, line 15 – missing reference.

*Reply: Corrected.*

7. The metrics used are generally clear and well reported, however, for clarity, it would be good to include bold highlighting of the best performing model for each case in Table 2 and Table 4.

*Reply:*

*Thank the reviewer for this comment. We have updated the Tables and the differences in performance can also be (more easily) seen in the Figures 7 and 12.*

8. Thank you for including the analysis of model performance and high pollutant concentrations. This is important.

Technical comments:
1. Line 26: 'For O3 at the urban background site the local photochemistry is? not properly accounted for by the relatively coarse Copernicus Atmosphere Monitoring Service ensemble model (CAMS) used here for forecasting O3, but is compensated for using the MLs (ML models?) by taking lagged measurements into account.' Perhaps?

*Reply: Thank, we have changed accordingly.*

2. Throughout: MLs or machine learning models? MLs is not a common term.
3. The Figure on page 9 has no caption or label.

*Reply:*

*We have revised accordingly.*

4. Conclusion: reflection -> reflecting

*Reply: We have corrected it.*

*References*
*Hong, H.; Choi, I.; Jeon, H.; Kim, Y.; Lee, J.-B.; Park, C.H.; Kim, H.S. An Air Pollutants Prediction Method Integrating Numerical Models and Artificial Intelligence Models Targeting the Area around Busan Port in Korea. Atmosphere 2022, 13, 1462. https://doi.org/10.3390/atmos13091462.*

*Kleinert, F., Leufen, L. H., Lupascu, A., Butler, T., and Schultz, M. G.: Representing chemical history in ozone time-series predictions – a model experiment study building on the MLAir (v1.5) deep learning framework, Geosci. Model Dev., 15, 8913–8930, https://doi.org/10.5194/gmd-15-8913-2022, 2022.*

---

## Author Response (AR2)

**Response letter for reviews of "Improving 3-day deterministic air pollution forecasts using machine learning algorithms"**
Referee comments in black and replies in *blue, italics.*

**Replies to Editor**
Thank you to the authors and reviewers for their comments and revisions. However, both reviewers have indicated a continued need for substantial revisions and clarifications in order to make the manuscript acceptable for publication in ACP. In particular, an acceptable version would require:
shortening and restructuring of a few sections (potential for where this could be achieved was highlighted by both reviewers in the two rounds of reviews).

*Reply:*
*Thanks for your comments and suggestions! We have conducted a comprehensive revision of the paper and carried out additional computational experiments in order to answer all the questions being raised.*

*We have revised the paper structure, making lots of effort to shorten the paper into a neat form. In summary, the conducted revisions are outlined as follows:*

*Structure changes:*
- *We have smoothed and shortened the description of Section I.*
- *The Air Quality System of Stockholm and meteorological forecast subsections are moved from the section of "methods" to a new section called "Background".*
- *Some results are moved to appendices to shorten the length of the main manuscript, e.g. some of Figures 7 – 14.*
- *We have merged the figures and created radar plots to simplify the presentation of the final results.*
- *Discussion section is revised to reduce the paper length.*

*Additional content*
- *We added two new subsections of data preprocessing and feature importance assessment in the methodology presentation.*
- *We added some explanations and results for hyperparameter tuning for each model.*
- *We added two new feature ranking methods (Permutation and SHAP) to satisfy the requirements of the reviewers*
- *We added some detailed analysis of feature importance in the result and discussion sections.*

*Meanwhile, we complement necessary clarifications in the text in response to the concerns and highlighted comments of both reviewers. Please refer to the individual answer to each question of the reviewers.*

additional analyses (e.g. addition to existing tables) contrasting different methods for measuring feature importances.

*Reply:*
*As mentioned in the outline before, two additional methods were implemented into both the tree-based model (XGBoost and RandomForest) and LSTM model to thoroughly assess the stability of the feature's importance ranking and the temporal dependence of crucial features.*

*Subsection 3.5 was added to explain the methods of feature importance ranking. Tree-based models commonly employ three methods: MDI, permutation (Breiman, 2001), and SHAP (Lundberg et al., 2017). Also, LSTM models typically utilize gradient-based method, permutation, and SHAP (Shrikumar et al., 2017) for measuring importance.*

*Correspondingly, we add subsection 5.2 to discuss the results of feature analysis.*

a test of how the selection of the COVID-affected year might influence estimated model performance

*Reply:*
*Indeed, the COVID-19 pandemic had an impact on road traffic and pollutant emissions as a result of some restrictive regulations implemented (Sokhi et al., 2021; Torkmahalleh et al., 2021). However, the COVID restrictions in Sweden continued until February 2022, but our data in this study only encompasses observations up to December 31, 2021. So, the dataset is within entire duration of the COVID period, which makes it impossible to assess the effect of COVID-19 in this study. Nevertheless, since we will continue the model development in the longer run, the effect of COVID will be investigated when new data after 2022 is included in the model training and prediction. We expect this will be presented in our future work.*

more details on the data pre-processing (and phrasing concerning its importance); if too long for the main text at least in the appendix
*Reply:*
*We have added data preprocessing method in subsection 3.1, which describes the preprocessing method, and the effect of data interpolation.*

a correct context and comparison of skill metrics with similar symbolic representations (r2-score is not the same as Pearson correlation)

*Reply:*

*Thank you for the comment. There is a notation in the header of tables 3/4/5 in the previous version that "r" represents Pearson correlation, sorry for the misunderstanding!*

*We currently replace Pearson correlation with R-squared in the result tables, and the calculation formula can be found in subsection 3.6.*

it is still not clear enough how/if/where hyperparameter tuning was conducted. The reviewers note that "This needs to be mentioned in the main text alongside the added list of actual parameters used. The LSTM was not subject to hyperparameter optimization. It is important for readers to understand that model and training choices are not arbitrary guess-work; it is a critical component for any model with the intention of deployment." and "they mention that they trained by setting different random number seeds to obtain optimal results. Searching over random seeds to find optimal results is likely to lead to overstating the performance of the model when it comes to using the model operationally or on other data?" Good answers to these points raised are critical to ensure the robustness of the results presented in the manuscript.
Overall, it would be necessary to comprehensively and convincingly address these concerns before the manuscript can be considered for publication. If necessary, analyses might need to be repeated/revised if e.g. the current hyperparameter tuning cannot be demonstrated to be robust (rotation of test vs. training years?). In addition, further minor comments by the reviewers would need to be addressed point-by-point.

*Reply:*
*Thank you for the comments! We agree that hyperparameter tuning is an important step in machine learning model construction. To clarify our work, we created subsection 3.4 on hyperparameter tuning. The optimal parameters and searching ranges for each model are shown in Appendix B.*

*To demonstrate the robustness and generalization ability of the models, we actually trained the model repeatedly 10 times by setting different random seeds. The results are evaluated using independent data and the mean value along with its 95% confidence interval is presented in Table 5/6/7. The findings reveal a quite small variation for the two tree-based models(XGBoost and RandomForest), whereas the LSTM model shows some fluctuation but less than 5% in most cases. This variance may be attributed to the stochastic initialization of weights, which influences the*

*subsequent trajectory of gradient descents and model adaptation in the training process.*

*We have addressed the concerns of the reviewers, point by point. Please refer to our answers to the reviewer's comments.*

**Replies to Referee #1**

This paper has piece-meal improved – the track changes show that the most prevalent change was replacing MLs with ML models; I was expecting more significant changes or comments to be incorporated. There are several comments highlighted below made by the authors where it seems they misunderstand the importance of some of the ML methods used.

*Reply:*
*Thanks for your suggestions and comments on this paper! We agree that the improvement in the first round is not comprehensive, and some of the points raised are missed. Furthermore, we have carried out a comprehensive revision and lots of additional computation experiments in the past few months to improve the manuscript and science behind it. The change we made can be summarized as follows:*

*Structure changes:*
- *We have smoothed and shortened the description of Section I.*
- *The Air Quality System of Stockholm and meteorological forecast subsections are moved from the section of "methods" to a new section called "Background".*
- *Some results are moved to appendices to shorten the length of the main manuscript, e.g. some of Figures 7 – 14.*
- *We have merged the figures and created radar plots to simplify the presentation of the final results.*
- *Discussion section is revised to reduce the paper length.*

*Additional content*
- *We added two new subsections of data preprocessing and feature importance assessment in the methodology presentation.*
- *We added some explanations and results for hyperparameter tuning for each model.*
- *We added two new feature ranking methods (Permutation and SHAP) to satisfy the requirements of the reviewers*
- *We added some detailed analysis of feature importance in the result and discussion sections.*

[1] The explanation for Tables 2 and 4 being kept in the main text is fair, and the addition of bold-face makes it a lot easier to see the top-1 in these tables. However, the feature importance plots are not shown in the main text. If the authors misunderstood last time: The feature importance results are some of the main ML results of the paper and they need to be in the main text at the expense of some of figures 7 through 14. For example, show only Figure 8 and not both Figures 7 and 8, in addition to tables, because Figure 8 shows the skill-score metric. that you actually care about. You can move the other less important results showing many columns of bake-off results to the appendix / supplementary materials and reference them in the main text. That alone would represent a major improvement to the manuscript.

*Reply:*
*Thank you for constructive suggestion! We have made new efforts to improve the presentation. We introduce radar charts to make it easier to present and compare performance metrics calculated from different models. In addition, for Figures 7-Figure 14, we merged some of the graphs to reduce space and moved the remaining results to the Appendix to shorten this manuscript.*

[2] The authors state that "how to interpret feature importance has been a side-line topic for understanding the RNN model" and then "there are other approaches to measure feature importance such as (Zhou et al), but this is beyond the scope of our current study.
I do not understand these comments since the authors correctly note that different methods (and model parameterizations) can lead to different results. The same is also true for tree-based models, which is why I asked in the first round for the authors to provide at least one other method for those models, which they did not do. The application of these methods are not beyond the scope of this paper, these are key details that the authors missed.

*Reply:*
*We didn't understand the comments precisely in the first round. Nevertheless, we had some internal discussions and finally implemented different feature ranking methods by following the suggestions of the reviewers.*
*We add subsection 3.5 in the main text to illustrate the feature ranking methods. We did thorough experiments to analyse the results, which are added in the sections of results and discussion.*

[3] Since reviewer #2 also noted the permutation importance method, this needs to be shown alongside MDI for the tree models (why even the choice of MDI, why not fANOVA? These two alone commonly give different rankings). The permutation method can be applied to both the tree models and the RNN (as can SHAP). You can stack the results from MDI along with permutation results without having to add another figure in the paper. The conclusion from doing such an analysis will allow the authors to identify which top features the different methods have in common, as those are the robust ones that should be selected.

*Reply:*
*Thank you for these detailed comments and good idea! Two additional methods were implemented into both the tree-based models (XGBoost and RandomForest) and LSTM model to thoroughly assess the robustness of the feature's importance ranking and the temporal dependence of crucial features.*

*Subsection 3.5 was added to explain the methods of feature importance. Tree-based models commonly employ three feature ranking methods: MDI, permutation(Breiman, 2001), and SHAP(Lundberg et al., 2017). Also, LSTM models typically utilize gradient-based method, permutation, and SHAP(Shrikumar et al., 2017) for measuring importance.*

*The results of four feature importance methods were moved to the main text, as shown in Figures 9 and 14, and more results are displayed in Appendices E and H. In addition, we use a model as an example and discuss the correlation between features and time-dependent properties based on TreeSHAP in subsection 5.2.*

[4] The authors stated "there are issues such that the gradient-based method calculates temporally varying rankings making the rankings of features dependent on the testing set". Correct. It would be a nice addition for the authors to investigate the September 20th case, again, why the large drop? Do the feature importance methods tell us what's going on?

*Reply:*
*Thank you for the suggestion! We have tried the idea to explain why the peaks cannot be captured such as the case of 20th Sept mentioned by the reviewer, and to see if we can explain the gaps by top-ranked features. Unfortunately, we didn't find any solid explanation. Nevertheless, the top-ranked features do help understand the models better, and some features show interesting temporal patterns. We have added some analysis in subsection 4.1.2 / 4.2.2 and a discussion about the temporal dependency of the top-ranked features in subsections 5.2.*

[5] The authors made the comment "In the pre-processing process, outliers, such as negative pollutant measurements, are identified and removed and furthermore, standard methods, such as interpolation, are applied to handle missing values in the data. Given the current length of the paper, adding such details will not benefit the paper's readability. But we have added a description in subsection 2.1 as follows: …" Imputation / managing missing data are critical details regarding the data preparation where the authors seem to diminish their importance with these comments.

*Reply:*
*We added a more detailed subsection in the methodology part, subsection 3.1, which describes the missing data case, the preprocessing method, and the effect of data interpolation.*

[6] The authors now state in the manuscript: "Due to the temporal correlation of the air pollutant concentrations, the principal assumption of cross-validation is not

satisfied. Therefore, to preserve the time-dependent property, "TimeSeriesSplit" was chosen as the cross-validation strategy. … The value of parameter k is set as at 5." I think what the authors mean to say is that cross-validation via random sampling is not satisfied and that a time-ordered strategy is required (cross-validation is being applied along the time coordinate). You need to say that you used sklearn when "TimeSeriesSplit" is mentioned for the first time, readers who are not familiar with sklearn will have no idea what it means. That an ensemble of size 5 was created means you can put the error-bars on the tables as I originally asked in the first round. You also noted that an ensemble was created via different seed choices. RNNs are usually very sensitive to initial weight choices. Whichever ensembling method you choose to report the error bars, state it clearly in the table captions.

*Reply:*
*Thank you for the comments! We have revised the paper accordingly. To demonstrate the robustness and generalization ability of the models, the training process repeats 10 times with different random seeds. The results are evaluated on independent test set, and the mean values along with its 95% confidence interval are shown in Table 5/6/7. The results reveal quite small variation in the two tree-based models(XGBoost and RandomForest), whereas the LSTM model presents a bigger variance but less than 5% in most cases.*

[7] The coefficient of determination, R2, is defined as $1 - \sum_i (y_i - f(x_i))^2 / \sum (y_i - \langle y \rangle)^2$. When used as a metric, R2=0 is the baseline model that predicts `<y>`; R2>0 is more skill-full relative to that baseline, <0 less skill-full than the baseline. Pearson correlation is not a measurement relative to a baseline, there can be high-correlation while at the same time poor skill relative to the simple baseline. Table 2 has "r = Pearson correlation" – in general the coefficient of determination is not the square of the Pearson score.

*Reply:*
*Thank you for the comment! There is a notation in the header of tables 3/4/5 in the previous version that "r" represents Pearson correlation, sorry for the misunderstanding!*

*The formula for all metrics is summarised in subsection 3.6. We currently replace Pearson correlation with R-squared value in the result tables but for comparison, it is still shown in the radar plot.*

[8] The authors mention that they performed a grid-search of the hyper-parameters. This needs to be mentioned in the main text alongside the added list of actual

parameters used. The LSTM was not subject to hyper-parameter optimization. It is important for readers to understand that model and training choices are not arbitrary guess-work; it is a critical component for any model with the intention of deployment.

*Reply:*
*Yes, we agree that hyperparameter tuning is an important procedure for machine learning model construction. We add a subsection, 3.4, on hyperparameter tuning in the main text to meet the requirements of both reviewers. We show that we did the hyperparameter optimization for all models(XGBoost, RandomForest and LSTM). To shorten the presentation, the parameter selection range and optimal parameters are added in Appendix B.*

**Replies to Referee #2**

Thank you to the authors for responding to reviewer comments.

Further comments:

1. Additional details on the data splitting is now provided, and temporal splitting has been used to train, validate and test the models. Thank you to the authors for adding this.

*Reply:*
*Thank you for notifying this!*

2. Thank you also for including the details on hyperparameters in 2.4. Currently however, it simply says that the 'default parameters' were used for the tree-based methods. The actual values of these parameters should be included, perhaps in the appendix.

*Reply:*
*Thank you for the comments! We added a subsection, 3.4, on hyperparameter tuning in the main text to meet the requirements of both reviewers. We show that we did the hyperparameter optimization for all models (XGBoost, RandomForest and LSTM). To shorten the presentation, the parameter selection range and optimal parameters are added in Appendix B.*

3. I think that the fact that the study uses data from coronavirus years should be highlighted. It may be the models are able to perform well on these years, but not on other years, even with further training? This is mentioned is the author response, but is not in the manuscript currently.

*Reply:*
*Yes, unfortunately, our current study is based on the data until the end of 2021. We added a short discussion in subsection 3.1 as follows:*

*"It should be noted that there are several studies showing the impact of the COVID-19 pandemic on pollutant emissions as a result of some restrictive regulations (Sokhi et al., 2021; Torkmahalleh et al., 2021). The COVID-19 pandemic in Sweden*

*commenced in January 2020 and continued until February 2022, so the majority of the data is collected during this pandemic period.''*

*Since we will continue the model development in the longer run, the effect of COVID will be investigated when new data after 2022 is included in the model training and prediction. We expect this will be presented in our future work.*

4. Thank you for including feature importance details. It should be mentioned that different feature importance methods can provide different importances, and there are many options available for this.

*Reply:*
*We have implemented two additional methods for both tree-based models (XGBoost and RandomForest) and LSTM model to thoroughly assess the stability of the feature's importance values and the temporal dependence of crucial features.*

*Subsection 3.5 was added to explain the methods of feature importance. Tree-based models commonly employ three feature ranking methods: MDI, permutation(Breiman, 2001), and SHAP(Lundberg et al., 2017). Also, LSTM models typically utilize gradient-based method, permutation, and SHAP(Shrikumar et al., 2017) for measuring importance.*

*The results of four feature importance methods were moved to the main text, as shown in Figures 9 and 14, and more results are displayed in Appendices E and H. In addition, we use a model as an example and discuss the correlation between features and time-dependent properties based on TreeSHAP in subsection 5.2.*

5. In the authors' response, they mention that they trained by setting different random number seeds to obtain optimal results. Searching over random seeds to find optimal results is likely to lead to overstating the performance of the model when it comes to using the model operationally or on other data? This is also not mentioned in the manuscript.

*Reply:*
*Thank you for the comment! To demonstrate the robustness and generalization ability of the models, the models are trained 10 times with different random seeds. The*

*results are evaluated on independent test data, and the mean values along with its 95% confidence interval are shown in Table 5/6/7.*

**References**

*Breiman, L.: "Random forests." Machine learning 45, 5-32, 2001.*
*Lundberg, S.M. and Lee, S.I.: A unified approach to interpreting model predictions. \*Advances in neural information processing systems\*, \*30\*, 2017.*
*Shrikumar, A., Greenside, P. and Kundaje, A.: Learning important features through propagating activation differences. In \*International conference on machine learning\* (pp. 3145-3153). PMLR, 2017.*
*Sokhi, R.S., Singh, V., Querol, X., Finardi, S., Targino, A.C., de Fatima Andrade, M., Pavlovic, R., Garland, R.M., Massagué, J., Kong, S. and Baklanov, A.: A global observational analysis to understand changes in air quality during exceptionally low anthropogenic emission conditions. \*Environment international\*, \*157\*, p.106818, 2021.*
*Torkmahalleh, M.A., Akhmetvaliyeva, Z., Darvishi Omran, A., Darvish Omran, F., Kazemitabar, M., Naseri, M., Naseri, M., Sharifi, H., Malekipirbazari, M., Kwasi Adotey, E. and Gorjinezhad, S.: Global air quality and COVID-19 pandemic: do we breathe cleaner air?, 2021.*

---

## Author Response (AR3)

**Response letter for reviews of "Improving 3-day deterministic air pollution forecasts using machine learning algorithms"**
Referee comments in black and replies in *blue, italics.*

**Replies to Editor**
The authors have implemented substantial changes to address the reviewer comments so that the manuscript is now ready for publication subject to technical corrections. In particular, the authors are asked to:
- check for effective ways to shorten the main part of their manuscript, e.g. through re-wording or by moving sections to the Supplementary Material. While it might be important to demonstrate certain points, especially in response to valid comments made by the the reviewers, the authors could re-consider whether a pointer to the Supplementary might suffice in a few cases. Further efforts to tighten the presentation where possible would enhance accessibility. Candidates could be the interpolation/missing data section (Figures 3 and 4), which could be moved to the appendix, as could be the part on hyperparameter tuning.

*Reply:*
*Thank you so much for your decision on accepting the paper subject to minor revisions! We have conducted some revisions of the paper to shorten the main text according to your suggestion.*
*Specifically, we moved the example of the interpolation result, Figure 3, and the example of the hyperparameter tuning result, Figure 6, to Appendix B and C, respectively.*

- re-read the manuscript and check for grammatical errors and typos. While there will be copyediting, this will reduce the risk of carrying over errors into the published version of the manuscript, especially those that are hard to spot for copyeditors. For example, I noticed several misspellings such as 'learin_rate'. In addition, there should always be spaces separating words from abbreviations (e.g. line 24 "exPlanations(SHAP)")

*Reply:*
*Thank you! We did a comprehensive copyediting and corrected the grammatical errors and typos we found.*

Please also consider brief explanations (maybe a couple of sentences each) on the following points mentioned in the letters to the editor:
- Being more specific on the exact imputation techniques used would add helpful details.

*Reply:*
*The essence of interpolation is to find the samples that are most correlated with the missing sample and replace the missing value with these correlated samples. In the context of time series data, the samples at time "t" are strongly correlated with time "t ± 1". At the same time, they are highly correlated with samples at time "t-p" and "t-2p", where "p" is the data's periodicity.*

*Considering the above two points and our prediction scheme, detailed in section 3.2, we adopted the historical average interpolation based on the periodicity. Subsequently, the missing value at time t is substituted by the average of the available data from two preceding periods (i.e., "t – p" and "t – 2p") as well as their adjacent values (i.e., "t - p ± 1" and "t - 2p ± 1").*

- Discussing limitations around still struggling to capture high concentrations would add useful context.

*Reply: Thank you for the comments! Our models indeed face challenges when predicting high-concentration values. Below, we outline the key limitations and potential reasons:*

*1. Sample Imbalance:*
*High-concentration peaks have minor occurrence in the sample, and they are not as well-represented in our training dataset as low or normal values. Such imbalance may introduce bias in the prediction models, leading to reduced performance in predicting high-concentration values.*

*2. Model Limitations:*
*While the machine learning models have effectively improved the predictions by the deterministic model, they are still not explicitly designed to capture extreme events. The model's architecture, feature selection, or training process may have limitations in handling high pollutant concentrations.*

*3. External Factors:*
*High pollutant concentrations can be influenced by external factors, such as sudden changes in meteorological conditions, industrial incidents, or instantaneous emission sources. These factors either do not belong to our feature space or are not adequately captured by our models due to their unpredictability, leading to reduced prediction accuracy for high-concentration values.*

*In our future work, we therefore plan to explore methodologies for refining our model architecture and enhance the model capability of predicting high pollutant concentrations.*

Finally:
- Figures 9 and 14 should be updated to have different colours for MDI/gradient methods.

*Reply:*
*We have modified all figures for feature ranking, including Figure 7, Figure 12 and the ones in the Appendix, also MDI for blue points and Gradient for purple points, respectively.*

- The radar plots are difficult to read. For example, the bottom of Figure 11 is cut off, but more importantly the axis in the title is said to be e.g. 0 - 1, but the axis on the radar plot is labelled in %s. It would be simpler if these were labelled with the numbers, rather than percentages. This would also remove the need for declaring the axis range in the title.

*Reply:*
*We updated all radar plots, including Figure 5, Figure 8 and Figure 9. Numbers are labelled on each axis to make it easier to read.*

**Replies to Referee #1**
Thank you for submitting the revised manuscript, which addresses most of my comments, including the values of hyperparameters and the details of the tuning, the inclusion and explanation of different feature importance methods and a clear explanation of how the model was trained re different seeds. I am happy to accept that the study is limited to coronavirus years, and looking at different years will be a part of future work. I still have a couple of minor comments on the figures
1. Figures 9 and 14 should be updated to have different colours for MDI/gradient methods.

*Reply:*
*Thank you for your valuable suggestions and efforts in previous rounds of reviewing.*

*We have modified all figures for feature ranking, including Figure 7, Figure 12 and the ones in the Appendix, also MDI for blue points and Gradient for purple points, respectively.*

2. The radar plots are difficult to read. For example, the bottom of Figure 11 is cut off, but more importantly the axis in the title is said to be e.g. 0 - 1, but the axis on the radar plot is labelled in %s. It would be simpler if these were labelled with the numbers, rather than percentages. This would also remove the need for declaring the axis range in the title.

*Reply:*
*We updated all radar plots, including Figure 5, Figure 8 and Figure 9. Values are now labelled on each axis to make it easier to read and understand.*

**Replies to Referee #2**
N/A